# Test-Time Training Provably Improves
# Transformers as In-context Learners

**Halil Alperen Gozeten** [* 1]  **Emrullah Ildiz** [* 1]  **Xuechen Zhang** [1]  **Mahdi Soltanolkotabi** [2]  **Marco Mondelli** [3]
**Samet Oymak** [1]

## Abstract

Test-time training (TTT) methods explicitly update the weights of a model to adapt to the specific test instance, and they have found success in a variety of settings, including most recently language modeling and reasoning. To demystify this success, we investigate a gradient-based TTT algorithm for in-context learning, where we train a transformer model on the in-context demonstrations provided in the test prompt. Specifically, we provide a comprehensive theoretical characterization of linear transformers when the update rule is a single gradient step. Our theory *(i)* delineates the role of alignment between pretraining distribution and target task, *(ii)* demystifies how TTT can alleviate distribution shift, and *(iii)* quantifies the sample complexity of TTT including how it can significantly reduce the eventual sample size required for in-context learning. As our empirical contribution, we study the benefits of TTT for TabPFN, a tabular foundation model. In line with our theory, we demonstrate that TTT significantly reduces the required sample size for tabular classification (3 to 5 times fewer) unlocking substantial inference efficiency with a negligible training cost.

## 1. Introduction

Modern language models can be viewed as general-purpose, addressing a diverse range of user queries in a zero-shot fashion (Kojima et al., 2022; Wei et al., 2022). However, as the complexity or novelty of the query increases, such as in complex multi-step reasoning scenarios, the pretrained model may falter. This has motivated two popular approaches: in-context learning and test-time computation. In-context learning (ICL) incorporates demonstrations related to the query as part of the prompt facilitating better inference by the model (Brown et al., 2020; Min et al., 2022). Test-time computation methods explicitly increase the inference-time compute to elicit higher quality responses (Snell et al., 2024; Jaech et al., 2024). An important instance of test-time computation is test-time training (TTT) where the weights of a model are explicitly updated to adapt to specific test instances (Sun et al., 2020; Liu et al., 2021). A gradient-based TTT approach can be described as follows: Given a test prompt and a pretrained sequence model, we update the pretrained weights by performing a few gradient iterations on a suitable self-supervised objective (e.g. next-token prediction objective). We can then use the resulting model to perform the inference on the test prompt. This procedure is applicable beyond language models, and it is conceptually similar to meta-learning approaches such as model-agnostic meta-learning (Finn et al., 2017).

Recently, TTT has found significant success in the context of language modeling by boosting the accuracy and reasoning capability of state-of-the-art models (Akyürek et al., 2024; Sun et al., 2024). This success is in part due to the fact that TTT can be naturally integrated within in-context learning: We can fine-tune the transformer model to fit to the labels or chain-of-thought rationales provided as part of the in-context demonstrations, and then re-apply ICL on the adapted model. In fact, recent work (Akyürek et al., 2024) utilizes a variation of this procedure and additional data augmentation to obtain remarkable improvements in the ARC reasoning benchmark. This motivates the central question of our work:

> *What are the provable benefits of test-time training of transformers, specifically, for enhancing in-context learning?*

**Contributions.** As our central contribution, we address this question by providing a comprehensive theoretical study of test-time training for one-layer linear transformers. Focusing on prompts following a linear dataset model, we provide

---

[*]Equal contribution   [1]University of Michigan, Ann Arbor [2]University of Southern California [3]Institute of Science and Technology Austria. Correspondence to: Halil Alperen Gozeten <alperen@umich.edu>.

*Proceedings of the 42nd International Conference on Machine Learning*, Vancouver, Canada. PMLR 267, 2025. Copyright 2025 by the author(s).

a *precise risk characterization of TTT with one gradient step update rule*. This characterization is established in terms of three ingredients: (i) *context length (number of in-context examples during inference)*, (ii) *target sample size available for TTT* in Section 4, and (iii) *alignment between the pre-trained model and target task* in Section 5. We show that, as sample size increases, TTT can alleviate the distribution shift bottlenecks that arise in standard ICL. This also reveals regimes where TTT with zero or small initialization is preferable to TTT with the pre-trained model (i.e. cold vs. warm start). Interestingly, our experiments with the GPT2 architecture (Radford et al., 2019) show that multi-layer transformers exhibit behavior in line with our theory. Our theory and experiments demonstrate that one step of TTT yields significant performance gains with less computation, consistent with recent empirical observations (Akyürek et al., 2024) that only a few gradient steps offer substantial test-time improvements. Additionally, while standard ICL requires $\Omega(d)$ context length under an isotropic task prior, with $d$ denoting feature dimension, we prove that TTT can succeed with $o(d)$ context length by effectively memorizing the target task. Our technical novelty stems from accurately capturing the statistical benefit of TTT during in-context learning. Specifically, while the transformer is trained on a single prompt, we characterize how the sample complexity benefit of TTT is proportional to the number of target examples within the prompt.

As empirical corroboration of our theory, we explore tabular learning and TabPFN (Hollmann et al., 2023; 2025) – a state-of-the-art tabular foundation model pretrained with structural causal model priors. TabPFN is well-aligned with our theoretical setting with similar token encodings but different prior distributions. An important drawback of TabPFN lies in its inference cost, as it uses a full tabular dataset as context during inference. In line with our theory, we demonstrate that TTT can convert TabPFN into a task-specific tabular model that works equally well with significantly less data (up to 5 times fewer). This in turn implies substantial inference gains given that the complexity of softmax-attention is quadratic in the sequence length.

## 2. Related Work

We organize our discussion of related work into two main areas: in-context learning and test-time training.

**In-context learning.** In-context learning has received significant interest during the past few years (Brown et al., 2020; Liu et al., 2023; Agarwal et al., 2024). This has also motivated works toward a stronger theoretical understanding of ICL (Zhang et al., 2024; Mahankali et al., 2024; Ahn et al., 2023b; Xie et al., 2022; Garg et al., 2022; Li et al., 2023). Closer to us, Mahankali et al. (2024); Ahn et al. (2023b); Zhang et al. (2024) study the optimization

landscape of a one-layer linear attention model and show that the optimized model implements a single step of projected gradient descent over the in-context demonstrations. More recent works (Lu et al., 2024; Wu et al., 2023) extend these to characterize the pretraining task sample complexity of ICL. While these works focus on pretraining capabilities, we focus on the adaptation of the pretrained model to the target task. Additionally, rather than empirical risk minimization, we use gradient descent for adaptation and characterize its risk when the sample size is determined by the target prompt length. In our theoretical model, each token represents a data point containing input features and the target label. Remarkably, TabPFN (Hollmann et al., 2023; 2025) shows that, using this simple encoding and pretraining the model with sufficiently rich data priors, transformers can accomplish state-of-the-art classification on tabular datasets via in-context learning. There have been also efforts (Thomas et al., 2024) to fine-tune in-context tabular models like TabPFN at the dataset level using retrieval-based local context selection.

**Test-time training.** Test-time training (Sun et al., 2020; Liu et al., 2021; Gandelsman et al., 2022) and related test-time adaptation (Wang et al., 2021; Niu et al., 2022; Yuan et al., 2023) methods aim to overcome distribution shift bottlenecks. This is typically accomplished by adapting the model with the test example using self-supervised or unsupervised objectives. These methods can work with just a single text example or admit a minibatch of examples. For sequence/language modeling tasks, one can utilize TTT on the test sequence via the next-token prediction objective (Sun et al., 2024; Hardt & Sun, 2024; Hübotter et al., 2025). Specifically, during in-context learning, the query is unlabeled (e.g. math problem to solve), but we are provided with related examples and associated labels (e.g. through retrieval). Thus, we can fit to these labels as the source of supervision (essentially fine-tuning the model on the dataset of in-context examples). For instance, the training objective in Akyürek et al. (2024) utilizes this approach boosted by additional data augmentation. Finally, the computational efficiency of TTT is an important consideration which motivates our investigation of TTT with a single gradient update.

## 3. Problem Setup

**Notation.** Let $[n]$ denote the set $\{1, \cdots, n\}$ for an integer $n \geq 1$. We denote vectors and matrices using bold lower-case and upper-case letters, respectively, such as $\boldsymbol{x}$ for vectors and $\boldsymbol{X}$ for matrices. The element $x_i$ refers to the $i$-th entry of the vector $\boldsymbol{x}$. We represent the zero vector of size $n$ by $\mathbf{0}_n$ and the zero matrix of size $m \times n$ by $\mathbf{0}_{m \times n}$. The operator $\mathtt{tr}(\boldsymbol{X})$ represents the trace of $\boldsymbol{X}$, $\boldsymbol{X}^\dagger$ is its Moore–Penrose pseudoinverse, and $\|\boldsymbol{X}\|_2$ denotes the spectral norm of $\boldsymbol{X}$. Given $\boldsymbol{x}, \boldsymbol{y} \in \mathbb{R}^d$, the notation $[\boldsymbol{x}\ \boldsymbol{y}] \in \mathbb{R}^{d \times 2}$ represents the

row-wise concatenation, while $[\boldsymbol{x};\boldsymbol{y}] \in \mathbb{R}^{2d}$ represents the column-wise concatenation.

**In-context learning and test-time training.** In-context learning is the ability of the model to learn from demonstrations in the prompt, i.e. *in-context*. Specifically, given a sequence of demonstrations of desired input/output pairs $(\boldsymbol{x}_1, y_1), (\boldsymbol{x}_2, y_2), \ldots, (\boldsymbol{x}_n, y_n) \in \mathbb{R}^d \times \mathbb{R}$ followed by a query input $\boldsymbol{x}$ in the prompt, the model can guess the corresponding output query $y$. Concretely, we can define the context tokens $\boldsymbol{z}_i = [\boldsymbol{x}_i; y_i] \in \mathbb{R}^{d+1}$ for $i \in [n]$ and the query token $\boldsymbol{z} = [\boldsymbol{x}; 0] \in \mathbb{R}^{d+1}$. Then the input prompt can be written in the form

$$\boldsymbol{Z} = [\boldsymbol{z}_1 \ \cdots \ \boldsymbol{z}_n \ \boldsymbol{z}]^\top = \begin{bmatrix} \boldsymbol{x}_1 \ \boldsymbol{x}_2 \ \cdots \ \boldsymbol{x}_n \ \boldsymbol{x} \\ y_1 \ y_2 \ \cdots \ y_n \ 0 \end{bmatrix}^\top \in \mathbb{R}^{(n+1)\times(d+1)}.$$

To estimate the output query $y$, we focus on a sequence model $\text{SM}(\boldsymbol{Z}, \boldsymbol{W})$ with $\boldsymbol{Z}$ the input prompt and $\boldsymbol{W}$ the model parameters. We assume that the sequence model is *pre-trained* with a token distribution $\mathcal{D}_{\text{z}}^{\text{PT}}$ where $(\boldsymbol{z}_i)_{i=1}^n, [\boldsymbol{x};\boldsymbol{y}] \overset{\text{i.i.d.}}{\sim} \mathcal{D}_{\text{z}}^{\text{PT}}$ with the corresponding optimal parameters given by

$$\boldsymbol{W}^* = \arg\min_{\boldsymbol{W}} \mathbb{E}_{(\boldsymbol{z}_i)_{i=1}^n, [\boldsymbol{x};\boldsymbol{y}] \sim \mathcal{D}_{\text{z}}^{\text{PT}}} \left[ (y - \text{SM}(\boldsymbol{Z}, \boldsymbol{W}))^2 \right]. \quad (1)$$

During inference, we test the sequence model on another distribution $\mathcal{D}_{\text{z}}^{\text{TT}}$ and observe $k$ samples of this test distribution $\mathcal{S}_{\text{TT}} = \{(\boldsymbol{Z}_j, y_j)\}_{j=1}^k$ where $y_j$ is the label of the query token inside $\boldsymbol{Z}_j$. The main idea behind *Test-Time Training* (TTT) is to refine the model's parameters using the test data $\mathcal{S}_{\text{TT}}$ before performing inference. Concretely, the empirical loss of the sequence model on the test set $\mathcal{S}_{\text{TT}}$ with an arbitrary model parameter $\boldsymbol{W}$ is given by

$$\hat{\mathcal{L}}_{\mathcal{S}_{\text{TT}}}(\boldsymbol{W}) = \sum_{j=1}^k (y_j - \text{SM}(\boldsymbol{Z}_j, \boldsymbol{W}))^2. \quad (2)$$

One can thus refine $\boldsymbol{W}^*$ by optimizing the above test-time empirical loss. In this paper, we focus on a TTT strategy involving a single gradient descent step over $\hat{\mathcal{L}}_{\mathcal{S}_{\text{TT}}}$, i.e. $\boldsymbol{W}_{\text{TT}} := \boldsymbol{W}^* - \eta \nabla \hat{\mathcal{L}}_{\mathcal{S}_{\text{TT}}}(\boldsymbol{W}^*)$. The error incurred by the model can then be calculated using the population loss via

$$\mathcal{L}(\boldsymbol{W}) = \mathbb{E}_{(\boldsymbol{z}_i)_{i=1}^n, [\boldsymbol{x};\boldsymbol{y}] \sim \mathcal{D}_{\text{z}}^{\text{TT}}} \left[ (y - \text{SM}(\boldsymbol{Z}, \boldsymbol{W}))^2 \right]. \quad (3)$$

Now, we will define the expected loss of the weights $\boldsymbol{W}_{\text{TT}}$ we achieve after test-time training over all test-time training sets $\mathcal{S}_{\text{TT}}$:

$$\mathcal{L}_{TT}(\boldsymbol{W}_{\text{TT}}) := \mathbb{E}_{\mathcal{S}_{\text{TT}}} [\mathcal{L}(\boldsymbol{W}_{\text{TT}})]. \quad (4)$$

Our goal is to characterize the loss $\mathcal{L}_{TT}(\boldsymbol{W}_{\text{TT}})$ obtained after test-time training as a function of the number of context samples $n$, the number of data points in test-time training

$k$, the distributions $(\mathcal{D}_{\text{z}}^{\text{PT}}, \mathcal{D}_{\text{z}}^{\text{TT}})$, and the pretrained starting point $\boldsymbol{W}^*$.

**Architecture.** We study the one-layer linear attention model as the sequence model:

$$\text{SM}(\boldsymbol{Z}, \boldsymbol{W}) = \left[ \boldsymbol{Z} \boldsymbol{W}_Q \boldsymbol{W}_K^\top \boldsymbol{Z}^\top \boldsymbol{Z} \boldsymbol{W}_V \right]_{n+1,d+1} = \boldsymbol{x}^\top \boldsymbol{W} \boldsymbol{X}^\top \boldsymbol{y}, \quad (5)$$

where $[\cdot]_{n+1,d+1}$ denotes the entry $(n+1, d+1)$ of the corresponding matrix. Here, the query, key, and value matrices $\boldsymbol{W}_Q, \boldsymbol{W}_K, \boldsymbol{W}_V \in \mathbb{R}^{(d+1)\times(d+1)}$ are defined as

$$\boldsymbol{W}_Q \boldsymbol{W}_K^\top = \begin{bmatrix} \boldsymbol{W} & \boldsymbol{0}_{d\times 1} \\ \boldsymbol{0}_{1\times d} & 0 \end{bmatrix} \qquad \boldsymbol{W}_V = \begin{bmatrix} \boldsymbol{0}_{d\times d} & \boldsymbol{0}_{d\times 1} \\ \boldsymbol{0}_{1\times d} & 1 \end{bmatrix},$$

and we set the data matrix $\boldsymbol{X} \in \mathbb{R}^{n\times d}$ and the corresponding labels $\boldsymbol{y} \in \mathbb{R}^n$ to be:

$$\boldsymbol{X} = \begin{bmatrix} \boldsymbol{x}_1 & \ldots & \boldsymbol{x}_n \end{bmatrix}^\top \qquad \boldsymbol{y} = \begin{bmatrix} y_1 & \ldots & y_n \end{bmatrix}^\top.$$

Here, we collapse the query and key matrices by identifying the top-left $d\times d$ block of $\boldsymbol{W}_Q \boldsymbol{W}_K^\top \in \mathbb{R}^{(d+1)\times(d+1)}$ with a single matrix $\boldsymbol{W} \in \mathbb{R}^{d\times d}$, and choose the value matrix to retrieve the output of the query token with one-layer linear attention. We note that similar models have been used in prior work (Zhang et al., 2023; Mahankali et al., 2023; Ahn et al., 2023a; Li et al., 2024; 2025) for the theoretical analysis of a variety of phenomena, albeit not for characterizing the benefits of TTT.

**Data model.** We consider a linear model with Gaussian data. Specifically, during pre-training, we assume the context tokens $\boldsymbol{z}_i = [\boldsymbol{x}_i; y_i]$ and the query input/output vector $[\boldsymbol{x}; y]$ to be sampled i.i.d. from the distribution $\mathcal{D}_{\text{z}}^{\text{PT}}(\boldsymbol{\Sigma}_{\boldsymbol{\beta}})$. Concretely, the outputs are generated according to the linear model $y_i = \boldsymbol{x}_i^\top \boldsymbol{\beta}_{\text{PT}} + \xi_i$, with task parameter $\boldsymbol{\beta}_{\text{PT}} \sim \mathcal{N}(\boldsymbol{0}, \boldsymbol{\Sigma}_{\boldsymbol{\beta}})$, input features $\boldsymbol{x}_i \sim \mathcal{N}(\boldsymbol{0}, \boldsymbol{\Sigma}_{\boldsymbol{x}})$, and noise terms $\xi_i \sim \mathcal{N}(0, \sigma^2)$ for $i \in [n]$.

During inference, we test the sequence model on a new task parameter $\boldsymbol{\beta}_{\text{TT}}$ with the input prompts generated following the test-time distribution $\mathcal{D}_{\text{z}}^{\text{TT}}(\boldsymbol{\beta}_{\text{TT}})$. This test-time distribution is governed by a similar linear setting. Concretely, the outputs are generated according to the linear model $y_i = \boldsymbol{x}_i^\top \boldsymbol{\beta}_{\text{TT}} + \xi_i$, where the input features $\boldsymbol{x}_i$ are sampled i.i.d. from the same feature distribution $\mathcal{N}(\boldsymbol{0}, \boldsymbol{\Sigma}_{\boldsymbol{x}})$ and the noise terms are sampled i.i.d. from the same distribution $\mathcal{N}(0, \sigma^2)$.

We construct the test-time set $\mathcal{S}_{\text{TT}} = \{(\boldsymbol{Z}_j, y_j)\}_{j=1}^k$ by following the test-time distribution $\mathcal{D}_{\text{z}}^{\text{TT}}(\boldsymbol{\beta}_{\text{TT}})$. We first sample $(n + k)$ query input/output pairs $\{(\bar{\boldsymbol{x}}_i, \bar{y}_i)\}_{i=1}^{(n+k)} \overset{\text{i.i.d.}}{\sim} \mathcal{D}_{\text{z}}^{\text{TT}}(\boldsymbol{\beta}_{\text{TT}})$. Then, we designate the first $n$ samples as fixed context tokens across the set, whereas we use the latter $k$ samples as queries in the set. In formulas,

$$\boldsymbol{Z}_j = \begin{bmatrix} \bar{\boldsymbol{x}}_1 \ \bar{\boldsymbol{x}}_2 \ \cdots \ \bar{\boldsymbol{x}}_n \ \bar{\boldsymbol{x}}_{j+n} \\ \bar{y}_1 \ \bar{y}_2 \ \cdots \ \bar{y}_n \ 0 \end{bmatrix}^\top, \quad y_j = \bar{y}_{j+n} \quad \forall j \in [k]. \quad (6)$$

**Remark:** Our procedure can be implemented using a single forward-backward pass by putting all examples within the prompt and using a suitable attention mask to ensure that only the first $n$ examples (context examples with labels) attend the last $k$ examples (query examples without labels) and vice versa. This allows for efficient parallel training. This also means that we use a fixed context-query split where the same $n$ examples are used as context during TTT. This can be generalized to a K-fold procedure where all examples are used as both context and query during TTT (as in Akyürek et al. (2024)); however, we opted for the 1-fold option, which is more amenable to a precise statistical analysis.

**Single-step GD for TTT.** We now turn our attention to deriving the single-step gradient update over the test set $\mathcal{S}_{\text{TT}}$. To this aim, we denote by $X_{\text{context}}$ and $y_{\text{context}}$ the context inputs and their outputs, whereas we denote by $X_{\text{train}}$ and $y_{\text{train}}$ the training query inputs and their outputs:

$$X_{\text{context}} = \begin{bmatrix} \bar{x}_1 & \dots & \bar{x}_n \end{bmatrix}^\top, \qquad y_{\text{context}} = \begin{bmatrix} \bar{y}_1 & \dots & \bar{y}_n \end{bmatrix}^\top,$$

$$X_{\text{train}} = \begin{bmatrix} \bar{x}_{n+1} & \dots & \bar{x}_{n+k} \end{bmatrix}^\top, \qquad y_{\text{train}} = \begin{bmatrix} \bar{y}_{n+1} & \dots & \bar{y}_{n+k} \end{bmatrix}^\top.$$

The following proposition (proved in Appendix A) characterizes the single-step GD TTT update.

**Proposition 3.1.** *Consider the linear attention model with parameters $W \in \mathbb{R}^{d \times d}$. Suppose the test-time training loss function is defined as in (2) and define $u_{\text{context}} := X_{\text{context}}^\top y_{\text{context}} \in \mathbb{R}^d$. Then, for any step size $\eta > 0$, the new parameter $W_{TT}$ after one gradient-descent step from $W$ is given by the rank-1 update*

$$W_{TT} = W + 2\eta X_{\text{train}}^\top (y_{\text{train}} - X_{\text{train}} W u_{\text{context}}) u_{\text{context}}^\top.$$

In the coming sections, we will start our analysis when the covariance matrices $(\Sigma_\beta, \Sigma_x)$ are isotropic (Section 4), and then we provide an extension to more general covariance matrices (Section 5). Finally, we corroborate our theoretical results with empirical evidence (Section 6).

## 4. Analysis for Isotropic Covariances

In this section, we study the loss $\mathcal{L}_{TT}(W_{\text{TT}})$ induced by a single-step gradient update in the isotropic scenario $(\Sigma_\beta, \Sigma_x) = (I, I)$, for an arbitrary test-time parameter $\beta_{\text{TT}}$. The assumption of isotropic covariance matrices enables us to explicitly characterize the behavior of the loss as a function of $n$, $d$, and $k$. We first analyze how $\mathcal{L}_{TT}(W_{\text{TT}})$ varies with the number of in-context examples ($n$) and the embedding dimension ($d$). We then compare two distinct choices of $W^*$, as a function of the test-time training set size $k$: *(i)* $W^*$ is obtained from (1), and *(ii)* $W^* = \mathbf{0}_{d \times d}$. The former represents a test-time training approach, whereas the latter corresponds to training from scratch with the single-step gradient update.

We start with the characterization of $W^*$ and its loss $\mathcal{L}(W^*)$.

**Proposition 4.1.** *Let $\Sigma_\beta, \Sigma_x = I, \sigma^2 = 0$ and let $\beta_{TT}$ be an arbitrary new task parameter. Then, the optimal pre-trained model's parameter $W^*$ and its population loss are given by*

$$W^* = \frac{1}{n+d+1} I, \qquad \mathcal{L}(W^*) = \|\beta_{TT}\|^2 \frac{d+1}{n+d+1}.$$

The characterization of $W^*$ is obtained from Li et al. (2024, Corollary 1) and its loss with respect to the new task parameter $\beta_{\text{TT}}$ is provided in Appendix B for the more general setting with noise term $\sigma^2$. Due to the technical difficulty of analyzing the single-step gradient in the noisy setting and to have manageable expressions, we will focus on the noiseless setting.

Given $k, n$, and $d$, the loss of the pre-trained parameters, $\hat{\mathcal{L}}_{\mathcal{S}_{\text{TT}}}(W^*)$, is a random variable with respect to the test-time set $\mathcal{S}_{\text{TT}}$. Consequently, the loss $\mathcal{L}(W_{\text{TT}})$ is also a random variable dependent on $\mathcal{S}_{\text{TT}}$. When applying a single-step update to the pre-trained model's parameters, we define the expectation of the loss $\mathcal{L}(W_{\text{TT}})$ with respect to $\mathcal{S}_{\text{TT}}$ in (4), treating it as a function of the step size $\eta$. The optimal step size is then selected to minimize the expected test-time training loss. In the following theorem (proved in Appendix B), we present the optimal learning rate and the corresponding improvement in the loss induced by test-time training.

**Theorem 4.2.** *Let $n/d = \Theta(1)$. Recall the definition of the expected loss $\mathcal{L}_{TT}(W_{TT})$ with respect to $\mathcal{S}_{TT}$ in (4). In the isotropic covariance and noiseless setting ($\sigma^2 = 0$), the optimal step-size that minimizes $\mathcal{L}_{TT}(W_{TT})$ is*

$$\eta^* \approx \frac{d}{2(k+d)\,n^2\,(n+d)\,\|\beta_{TT}\|_2^2}.$$

*With this optimal step-size $\eta^*$, the improvement in the loss due to the test-time training is*

$$\mathcal{L}(W^*) - \mathcal{L}_{TT}(W_{TT}) \approx \frac{k}{k+d} \frac{d^3}{(n+d)^3} \|\beta_{TT}\|_2^2.$$

*Remark* 4.3. We note that in the proportional $n, d$ regime, the Gaussian approximation in Lemma B.1, which is utilized in the proof of Theorem 4.2, introduces an error of $O(n^{-1})$. Additionally, during the derivation, we omit lower-order terms (such as $2n$ in $n^2 + 2n$), which introduces errors of one degree lower relative to the leading terms. Consequently, after carrying these approximations through the entire derivation, the overall approximation error remains at most $O(n^{-1})$, since the final loss expression is zeroth-order in $n, d$. Hence, our result is robust to these errors as $n, d \to \infty$ while maintaining the ratio $\alpha = n/d$. The same reasoning also applies to the non-isotropic covariance case, which will be discussed in Section 5.

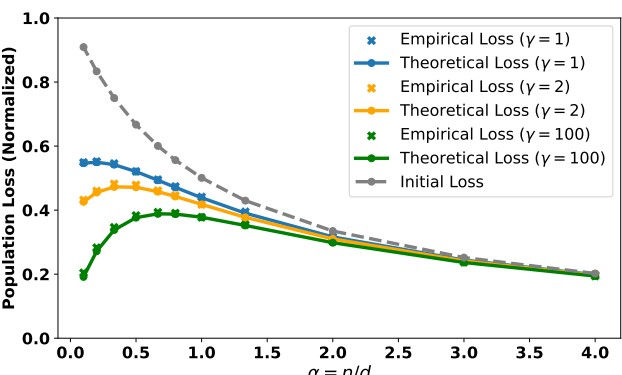

(a) Non-monotonic behavior of the loss $\mathcal{L}(W_{\text{TT}})$ after the test-time-training update with respect to $\alpha = n/d$ for various $\gamma = k/d$ values.

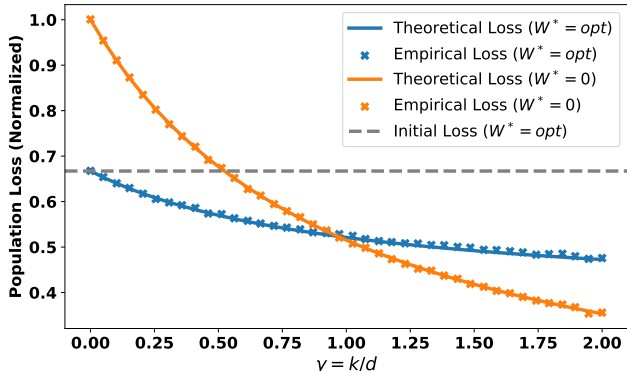

(b) Losses $\mathcal{L}(W_{\text{TT}})$ after the test-time-training update as a function of $\gamma = k/d$ when using optimal pre-trained weights vs. $\mathbf{0}_{d \times d}$ (null).

*Figure 1.* Plot of the normalized population losses after test-time training when $\Sigma_x = \Sigma_\beta = I$, $\sigma^2 = 0$, and the new task is $\beta_{\text{TT}} = \mathbf{1}_d$. Solid lines denote theoretical predictions which match the empirical (markers) results. **(a):** The figure illustrates a non-monotonic trend as the ratio $\frac{n}{d}$ shifts, see Corollary 4.4. **Setting:** $n = 300$; the ratio $\frac{n}{d}$ changes between 0.1 and 4; the three lines depict $\gamma = 1$, $\gamma = 2$, and $\gamma = 100$ where $\gamma = k/d$; the step size $\eta$ is selected optimally as per Theorem 4.2. **(b):** The figure reveals a threshold in $k$ at which the preferable initialization switches from the optimal pre-trained $W^*$ to the null model $\mathbf{0}_{d \times d}$ as $k$ increases. The transition threshold $k$ aligns well with Corollary 4.6. **Setting:** $n = 200$; $d = 400$; $k$ changing between 0 and $4n$ with equal increments.

Theorem 4.2 establishes the improvement produced by the test-time training as a function of $k, n$, and $d$. When $n$ and $d$ are fixed, such improvement is proportional to $\frac{k}{k+d}$. Additionally, if $k/d = \gamma$ is fixed, the loss $\mathcal{L}_{TT}(W_{TT})$ exhibits an intriguing behavior as a function of $n/d = \alpha$, as described by the next corollary (proved in Appendix B).

**Corollary 4.4.** *Recall the definitions of $\gamma = k/d$, $\alpha = n/d$, and consider the loss $\mathcal{L}_{TT}(W_{TT})$ as a function of $\alpha$ in the isotropic covariance and noiseless setting. If $\gamma > \frac{1}{2}$, then the loss $\mathcal{L}_{TT}(W_{TT})$ is non-monotonic in $\alpha$. Specifically, the loss $\mathcal{L}_{TT}(W_{TT})$ is increasing for $\alpha < \sqrt{\frac{3\gamma}{\gamma+1}} - 1$ and decreasing for $\alpha > \sqrt{\frac{3\gamma}{\gamma+1}} - 1$. Conversely, if $\gamma < \frac{1}{2}$, then $\mathcal{L}_{TT}(W_{TT})$ is monotonic in $\alpha$.*

The phase transition points described in Corollary 4.4 are approximate but become exact as $k, n$, and $d$ approach infinity while maintaining the ratios $\gamma = k/d$ and $\alpha = n/d$. The non-monotonic behavior identified in Corollary 4.4 is observed as a result of two opposing effects, which are the initial loss and the improvement by TTT. As $n$ grows against $d$ (i.e. $\alpha$ increases), the pre-trained model does better initially and already has a lower loss before TTT, which makes it harder to be further reduced by TTT as the rank-1 update is unable to correct all directions. On the other hand, when $d$ grows against $n$ ($\alpha$ decreases), the initial loss is high, providing more room (error) to be corrected by rank-1 update, and thus, the improvement by TTT is larger. This intuition aligns with Theorem 4.2, which establishes that the TTT improvement scales as $(\frac{d}{n+d})^3$. Together, these two trends result in the non-monotonic behavior.

We plot the behavior of the loss $\mathcal{L}_{TT}(W_{TT})$ as a function of $\alpha = n/d$ for various values of $\gamma = k/d$ in Figure 1a, making the following three observations: *(i)* As stated in Theorem 4.2, the improvement achieved through test-time training is approximately proportional to $1/(\alpha + 1)^3$ for large $n$ and $d$. This behavior is evident in Figure 1a, which shows that for every value of $\gamma$, the improvement diminishes as $\alpha = n/d$ increases. *(ii)* The non-monotonic behavior described Corollary 4.4 is clearly observed in Figure 1a. The increasing/decreasing regions in the loss as a function of $\alpha$ for different values of $\gamma$ are consistent with the theoretical results. *(iii)* As $\gamma$ increases, the improvement from the gray curve by test-time training is proportional to the ratio $\gamma/(\gamma + 1)$ for a fixed $\alpha = n/d$. This observation aligns with Theorem 4.2.

In addition to the scenario where we begin from an optimally pre-trained model $W^*$, a natural question is how much improvement can be achieved when the initial weight matrix $W^* = \mathbf{0}_{d \times d}$ (zero initialization). This initialization corresponds to training the model from scratch using a single-step gradient descent. In the following theorem, we quantify the improvement gained by applying a single-step gradient descent from this initialization.

**Theorem 4.5.** *Consider the isotropic covariance and noiseless setting ($\sigma^2 = 0$). Suppose the initial weight matrix is $W^* = \mathbf{0}_{d \times d}$. Then, the optimal step-size that minimizes $\mathcal{L}_{TT}(W_{TT})$ is*

$$\eta^* = \frac{1}{2(k + d + 1)(n^2 + 4n + 3 + d)\|\beta_{TT}\|_2^2}.$$

*With this optimal step-size $\eta^*$, the improvement in the loss*

*due to test-time training is*

$$\mathcal{L}(\mathbf{W}^*) - \mathcal{L}_{TT}(\mathbf{W}_{TT}) = \frac{k}{k+d+1} \frac{n^2}{n^2+4n+3+d} \|\boldsymbol{\beta}_{TT}\|_2^2,$$

*where $\mathcal{L}(\mathbf{W}^*) = \|\boldsymbol{\beta}_{TT}\|_2^2$.*

We provide the proof of Theorem 4.5 in Appendix B, where we also solve the more general noisy setting with initial weight $\mathbf{W}^* = \mathbf{0}_{d \times d}$. If $\sigma^2 = O\left(\|\boldsymbol{\beta}_{TT}\|_2^2\right)$ and $k$ grows sufficiently faster than $d$ (e.g., $k/d \to \infty$), then the test-time training update can reduce the loss to near 0 as $k, n, d \to \infty$ with $\alpha = n/d = \Theta(1)$.

Armed with Theorem 4.2 and 4.5, we can now establish when it is better to initialize with the pre-trained $\mathbf{W}^*$, as opposed to the zero (null) initialization, as the size $k$ of the test-time training set varies.

**Corollary 4.6.** *Recall the definitions of $\alpha = n/d$ and $\gamma = k/d$. Consider the setting in Theorem 4.2 and test-time training described in Proposition 3.1 with both pre-trained and zero initializations. Then, under the isotropic covariance and noiseless setting, there exists a threshold $\gamma^\star$ given by $\gamma^\star \approx \dfrac{(\alpha+1)^2}{(\alpha+2)}$ such that $\gamma < \gamma^\star$ if and only if it is better to utilize the pre-trained initialization over the zero initialization $\mathbf{0}_{d \times d}$.*

Corollary 4.6 identifies a phase transition point as a function of $\gamma$ and $\alpha$ that distinguishes the region where test-time training outperforms training from scratch. In Figure 1b, we illustrate the loss $\mathcal{L}_{TT}(\mathbf{W}_{TT})$ as a function of $\gamma = k/d$ for pretrained and zero initializations, with a fixed $\alpha = n/d = 1/2$. Based on Corollary 4.6, the phase transition point for $\gamma$ is $\frac{(3/2)^2}{5/2} = 9/10$, which aligns with the empirical results.

The pre-trained initialization provides a significant advantage when the test-time training set is small, as it introduces a strong prior that facilitates better performance. However, as the test-time training set grows, this initialization becomes a limitation because the rank-one update to the pre-trained matrix is insufficient to achieve a near-zero loss. In contrast, with zero initialization, the rank-one update becomes highly effective when the test-time training set is sufficiently large, enabling it to achieve near-zero error. This demonstrates that while pre-trained initialization is beneficial in data-scarce scenarios, zero initialization is better suited for scenarios with sufficient test-time training data.

## 5. Analysis for General Covariance

In this section, we extend the previous analysis to the general case where $\Sigma_x$ and $\Sigma_\beta$ are any jointly diagonalizable covariances. Specifically, we assume the task covariance $\Sigma_\beta$ to be a non-isotropic diagonal matrix while keeping the feature covariance matrix $\Sigma_x$ isotropic.[1] This assumption enables us to characterize the loss function $\mathcal{L}_{TT}(\mathbf{W}_{TT})$ with respect to the alignment between the pre-trained initialization matrix $\mathbf{W}^*$ and the new task parameter $\boldsymbol{\beta}_{TT}$.

We start with the definition of two quantities, which will be used to express the optimal step size and the loss improvement via the test-time training.

**Definition 5.1.** For an arbitrary $\mathbf{W} \in \mathbb{R}^{d \times d}$, define $A = \boldsymbol{\beta}_{TT}^\top (\mathbf{I} - n\mathbf{W})^2 \boldsymbol{\beta}_{TT}$ and $B = n \|\boldsymbol{\beta}_{TT}\|_2^2 \|\mathbf{W}\|_F^2$.

Here, the term $A$ represents the *misalignment* between the test-time task parameter $\boldsymbol{\beta}_{TT}$ and the initial parameter $\mathbf{W}$, whereas the term $B$ represents the total signal power of the one-layer linear attention system with parameter $\mathbf{W}$. Note that the terms $A$ and $B$ are a function of the new task parameter $\boldsymbol{\beta}_{TT}$ and the initial weight parameter $\mathbf{W}$.

Using the definitions of $A$ and $B$, we are ready to generalize Proposition 4.1 to the non-isotropic and rank-deficient covariance matrices. Note that for the rank-deficient covariance matrix, we take the optimal $\mathbf{W}^*$ as the minimum Frobenius-norm matrix that minimizes the loss in (1).

**Proposition 5.2.** *Let $\Sigma_x = \mathbf{I}, \sigma^2 = 0$ and suppose $\Sigma_\beta$ is an arbitrary diagonal covariance matrix with the first $r$ diagonal entries being non-zero. Then, the minimizer $\mathbf{W}^*$ of the pre-training loss (1) that has minimal Frobenius norm and its population loss are given by*

$$\mathbf{W}^* = \left((n+1)\mathbf{I}' + tr(\Sigma_\beta)\Sigma_\beta^\dagger\right)^\dagger, \quad \mathcal{L}(\mathbf{W}^*) \approx A + B,$$

*where $\mathbf{I}' = diag(\mathbf{1}_r, \mathbf{0}_{d-r})$.*

We generalize the characterization of $\mathbf{W}^*$ obtained from Li et al. (2024, Corollary 1) to rank-deficient covariance matrices, and its loss with respect to the new task parameter $\boldsymbol{\beta}_{TT}$ is provided in Appendix C.

Note that the eigenvalues of $\mathbf{W}^*$ are smaller than $1/(n+1)$ as $\Sigma_\beta$ is a positive semi-definite matrix. This condition ensures the stability of the linear attention model described in (5) as the magnitude of the SM output scales with the context length $n$ when $\mathbf{W}$ is fixed. Assuming a similar criterion for the set of $\mathbf{W}$, we provide the optimal step size parameter and the corresponding test-time training loss.

**Theorem 5.3.** *Let $n/d = \Theta(1)$, $\Sigma_x = \mathbf{I}$, and $\sigma^2 = 0$. Suppose that the initial weight matrix $\mathbf{W}$ is a diagonal matrix whose eigenvalues are in the interval $[0, \frac{1}{n+1}]$.[2] Then, the*

---

[1]This assumption is equivalent to a more general case where $\Sigma_x$ and $\Sigma_\beta$ are jointly diagonalizable, as proved in Appendix C. Joint diagonalizability refers to the case where there exists a unitary matrix $\mathbf{Q} \in \mathbb{R}^{d \times d}$ such that the covariance matrices $\Sigma_\beta = \mathbf{Q}\Lambda_\beta \mathbf{Q}^\top$ and $\Sigma_x = \mathbf{Q}\Lambda_x \mathbf{Q}^\top$ can be expressed with diagonal $\Lambda_\beta$ and $\Lambda_x$ matrices.

[2]In the joint diagonalizability case, the restriction on $\mathbf{W}$ covers the space of $\mathbf{W}$ that is jointly diagonal with $\Sigma_\beta$ and $\Sigma_x$.

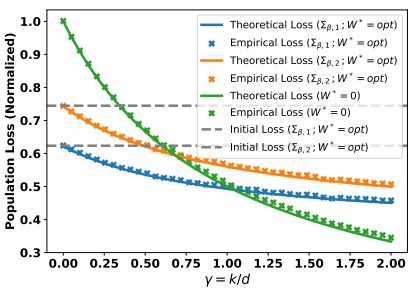
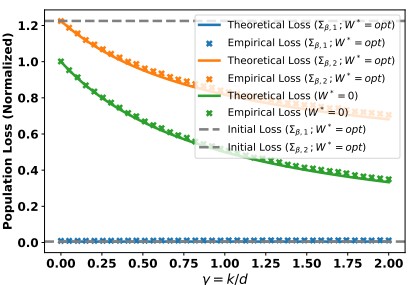
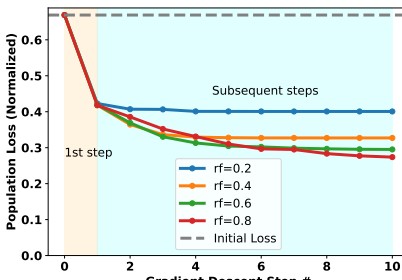

(a) Losses after test-time-training as a function of $\gamma = k/d$ for optimal weights $W^*$ based on two covariances and also $\mathbf{0}_{d \times d}$.

(b) Losses after test-time-training vs. $\gamma$ for optimal weights $W^*$ with worst/best aligned covariances and $\mathbf{0}_{d \times d}$.

(c) Comparison of losses after single and multiple steps of gradient descent at test time for different step size decays.

*Figure 2.* Population losses in single and multi-step update scenarios. $\Sigma_x = I$, $\sigma^2 = 0$ and step sizes are optimal based on the formula in Theorem 5.3. $W^* = \text{opt}$ and $W^* = 0$ represent the pre-trained matrix in (1) and zero (null) initialization, respectively. **(a): Setting:** $n = 300$; $d = 600$; $k$ changing between 0 and $4n$ with equal increments; $\Sigma_{\beta,1} = \text{diag}(I_{250}, 0.5 \cdot I_{250})$; $\Sigma_{\beta,2} = \text{diag}(0.5 \cdot I_{250}, I_{250})$; $\beta_{\text{TT}} = \left[\mathbf{1}_{250} \; ; 0.5 \cdot \mathbf{1}_{250}\right]$. Solid lines denote theoretical predictions which match the empirical (markers) results. **(b): Setting:** $n = 250$; $d = 500$; $k$ changing between 0 and $4n$ with equal increments; $\Sigma_{\beta,1} = \text{diag}(1, \mathbf{0}_{499})$; $\Sigma_{\beta,2} = \text{diag}(0, I_{499})$; $\beta_{\text{TT}} = \left[1 \; ; \mathbf{0}_{499}\right]$. **(c):** The figure illustrates the empirical results, which imply that with the initially optimal step size, a significant improvement can be gained by a single gradient step. Each line depicts a different reduce factor on the step size $\eta$ after each gradient step. **Setting:** $n = 50$; $d = 100$; $k = 50 \times d$; $\Sigma_\beta = I$; $\beta_{\text{TT}} = \mathbf{1}_{100}$; $W^*$ is optimal.

*optimal step size that minimizes the population loss given in* (3) *after the test-time training update is*

$$\eta^* \approx \frac{A}{2(k+d)\,n^2\,\|\beta_{TT}\|_2^2\,(A+B)}.$$

*With this optimal step-size $\eta^*$, the improvement in the loss due to test-time training and the initial loss are approximately*

$$\mathcal{L}(W) - \mathcal{L}_{TT}(W_{TT}) \approx \frac{k}{k+d}\frac{A^2}{A+B}\,, \quad \mathcal{L}(W) \approx A + B.$$

A more general and equivalent version of this result, which applies to any pair of jointly diagonalizable (not necessarily diagonal) covariance matrices $\Sigma_x$ and $\Sigma_\beta$ and to any $W$ that is also jointly diagonalizable with them, can be found in Appendix C. The above approximations are robust and become exact as $n, d \to \infty$ while having a fixed ratio $n/d$; see Remark 4.3. Theorem 5.3 characterizes the improvement due to test-time training for a rather general class of $W$. It shows that, for fixed $\|W\|_F^2$, the loss improvement increases monotonically with $A$. In other words, when the misalignment measure $A$ is larger, test-time training achieves greater performance gains. Consequently, considering the optimal pre-trained $W^*$, if the eigenvalue spectrum of $\Sigma_\beta$ aligns with the entries of the task $\beta_{\text{TT}}$ (i.e., larger eigenvalues match larger entries), then $A$ is larger and thus yields a bigger improvement.

Furthermore, we note that the optimal $\mathcal{L}(W)$ over all $W \in \mathbb{R}^{d \times d}$ is $O(n^{-1})$ by Lemma C.1. This optimal loss behavior can be achieved through test-time training with the initial

parameter satisfying $n\|W\|_2 < 1 - \delta(n,d)$ for some fixed $\delta(n,d) > 0$ as $k/d$ approaches infinity by Theorem 5.3. This means that test-time training achieves the optimal solution when we have zero or small initialization of $W$ as $k/d \to \infty$.

For the remainder of this section, we focus on the optimal pre-trained $W^*$ constructed in Proposition 5.2, i.e. the minimum-Frobenius-norm solution of the pre-training loss (1). When we initialize the pre-trained model weight as $W^*$, the loss $\mathcal{L}_{TT}(W_{\text{TT}})$ can be computed by combining Proposition 5.2 and Theorem 5.3. When we initialize the weights as $W = \mathbf{0}_{d \times d}$, then the loss $\mathcal{L}_{TT}(W_{\text{TT}})$ is obtained by using the fact that $\mathcal{L}(\mathbf{0}_{d \times d}) = \|\beta_{\text{TT}}\|_2^2$. By combining these results, we identify the phase transition point in the non-isotropic task covariance setting, which determines when test-time training outperforms training from scratch.

**Corollary 5.4.** *Consider the setting in Theorem 5.3. Recall the definitions of $\alpha = n/d$ and $\gamma = k/d$ and define $c_1 = \frac{A}{\|\beta_{TT}\|_2^2}$, $c_2 = \frac{B}{\|\beta_{TT}\|_2^2}$. Then, there exists a phase transition point $\gamma^\star \approx \frac{(c_1 + c_2) - (c_1 + c_2)^2}{c_2\,(2c_1 + c_2)}$ such that $\gamma < \gamma^\star$ if and only if it is better to utilize the pre-trained $W^*$ over the null initialization $\mathbf{0}_{d \times d}$.*

In a similar way as Corollary 4.4, the phase transition point mentioned above is approximate for finite $k, n,$ and $d$, and it becomes exact when $k, n,$ and $d$ approach infinity while keeping the ratios $\alpha$ and $\gamma$ fixed.

We illustrate the findings of Corollary 5.4 in Figures 2a and 2b and make the following three observations: *(i)* The

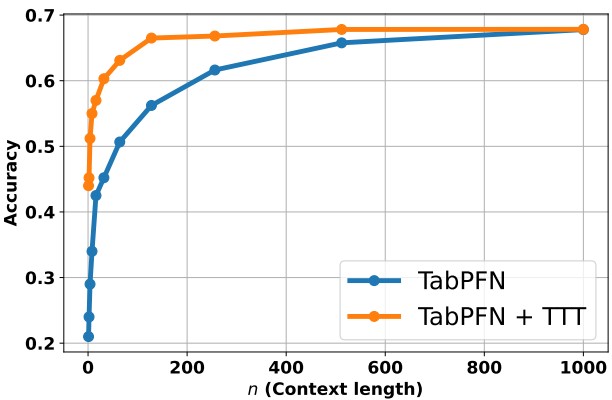

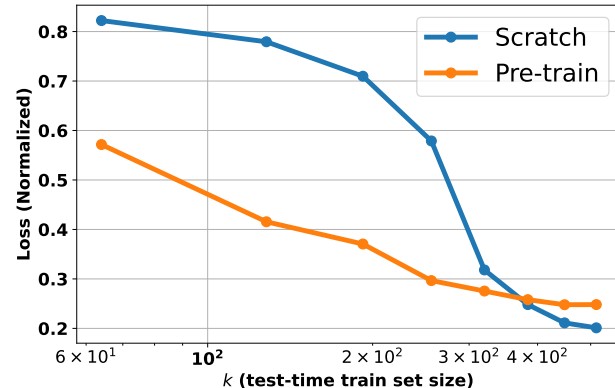

(a) Accuracy of TabPFN v2 model with and without test-time-training as a function of number of in-context samples $n$.

(b) Losses after test-time-training for GPT2 comparing pre-trained and scratch models as a function of test-time training set size $k$.

*Figure 3.* **(a):** Accuracy of TabPFN v2 with and without test-time training. **Setting:** $k = 1000$; $n$ changing between 1 and 1000. **(b):** Normalized loss after test-time-training with pre-training and zero initialization. **Setting:** $d = 60$; $n = 40$; $k$ changing between 64 and 512 in increments of 64; $\sigma^2 = 0.01$; $\Sigma_\beta = \text{diag}(0.1 \cdot I_{25}, 0.5 \cdot I_{10}, I_{25})$; $\Sigma_x = I$. We sample $\beta_{\text{TT}}$ from the distribution $\mathcal{N}(\mathbf{0}, \Sigma_{\beta_{\text{TT}}})$ where $\Sigma_{\beta_{\text{TT}}} = \text{diag}(I_{25}, 0.5 \cdot I_{10}, 0.1 \cdot I_{25})$.

phase transition point is monotonic as a function of $c_1$, while keeping $c_2$ fixed, which implies that it is monotonic as a function of the misalignment term $A$. This means that, as we increase the alignment between the task covariance $\Sigma_\beta$ and the new task parameter $\beta_{\text{TT}}$, the crossing point of the green and blue curves occurs later than that of the green and orange curves, as observed in Figure 2a. *(ii)* The worst-aligned case over the set of all possible $(\Sigma_\beta, \beta_{\text{TT}})$ pairs is the one that satisfies $\beta_{\text{TT}}^\top \Sigma_\beta \beta_{\text{TT}} = 0$. In this case, the corresponding $c_1$ is equal to 1, which implies that the phase transition point in Corollary 5.4 is less than 0 as $c_2 > 0$. This means that training from scratch is always better than utilizing the test-time training for the worst-aligned case, which is depicted in Figure 2b as the orange curve. *(iii)* The best-aligned case over the set of all possible $(\Sigma_\beta, \beta_{\text{TT}})$ pairs is the one that satisfies $\beta_{\text{TT}}^\top \Sigma_\beta \beta_{\text{TT}} / \text{tr}(\Sigma_\beta) = \|\beta_{\text{TT}}\|_2^2$. In this case, the corresponding $c_1$ scales on the order of $n^{-2}$ and $c_2$ on the order of $n^{-1}$, which in turn makes the phase transition point from Corollary 5.4 approximately $n$. When $k, n,$ and $d$ all grow while maintaining the ratios $\alpha = n/d$ and $\gamma = k/d$ fixed, this phase transition point approaches infinity. As a result, test-time training is always better than training from scratch in the best-aligned case, which is depicted in Figure 2b as the blue curve.

## 6. Experiments

In this section, we provide empirical results in light of our theoretical insights from previous sections. The first discussion of this section focuses on whether further computation via multi-step gradient descent at test time yields significant improvements beyond a single-step update. Then, we focus

on tabular learning by demonstrating how test-time training reduces context length for in-context learning on TabPFN. Finally, we demonstrate how the pre-trained model behaves against the scratch model under the test-time-training update for the GPT-2 model in the presence of a distribution shift.

### 6.1. Multi-Step Gradient Updates

In Figure 2c, we investigate the benefits of multiple gradient steps by choosing the optimal single-step size $\eta$ based on Theorem 4.2, and then applying different decay factors on step size after each subsequent step. We note that reducing the step size each time is necessary to ensure continued improvement of the loss. The results in Figure 2c confirm that a single-step test-time-training update can capture significant improvement, whereas additional steps yield diminishing returns compared to the initial step. Consequently, the single-step gradient update offers a favorable trade-off with less computation yet with the performance near multi-step finetuning.

### 6.2. Experiments on TabPFN

We demonstrate that test-time training (TTT) can significantly reduce the context length required for a given performance on TabPFN v2 (Hollmann et al., 2025). Specifically, we evaluate the TabPFN v2 model on the The Tremendous TabLib Trawl (T4) dataset (Gardner et al., 2024) for a more comprehensive evaluation, which is a large-scale high-quality collection of tabular benchmarks. Following the official TabPFN v2 implementation and our theoretical setup, we select the datasets containing at least 1,250 samples (with 1,000 for training, using an 80–20 split), limit

the number of classes to 10 by choosing the most frequent ones, and restrict datasets to 100 randomly selected features. We also convert regression tasks into 10-class classification tasks based on quantiles to maintain consistency across training and evaluation. For each selected dataset from T4, we evaluate TabPFN v2 with context length $n$ varying from 1 to 1000. For the TabPFN (blue curve in Figure 3a), we directly load the pre-trained model and vary the context window length during evaluation. In contrast, for TabPFN+TTT (orange curve in Figure 3a), we finetune the model using different context lengths with $k = 1000$ samples. As the context length $n$ decreases, the samples are divided into $1000/n$ groups, where each group undergoes 50 training iterations. We then report the average performance across all datasets with enough training samples.

In the previous sections, we have theoretically shown that TTT reduces the required in-context sample size for queries from a new task by performing only a single adaptation step, thus enhancing the efficiency of inference. As empirical corroboration of this, Figure 3a illustrates that TabPFN with test-time-training allows the model with 200 in-context samples to almost match the performance of the TabPFN without test-time-training with 1000 in-context samples. This means that test-time training reduces the number of required samples for tabular classification by about 5 times. It is worth emphasizing that the sample complexity benefit is more evident near the zero-shot regime as TTT helps the model memorize new task dynamics by improving the accuracy from 0.2 to 0.45. A log-scaled version of Figure 3a is provided in Appendix D (Figure 4) for a clearer view of performance gains across different scales of $n$.

### 6.3. Experiments on GPT-2

We demonstrate how the pre-trained model compares against the scratch model under test-time training with distribution shift using GPT-2 architecture (Radford et al., 2019), as shown in Figure 3b. Our results are in agreement with Figure 1b.

Throughout the experiments, we consider the linear model with Gaussian data following Section 3. In order to obtain a pre-trained model, we sample multiple tasks from the distribution with task covariance $\Sigma_\beta$ and train the model from scratch on these tasks until convergence. In contrast, in the scratch setting, the model is initialized randomly using the GPT-2 architecture without any pre-training. During test-time training, we evaluate the model on new task parameters sampled from the distribution $\Sigma_{\beta_{TT}}$, where the covariance structure is provided in the caption of Figure 3b. We then apply test-time training to both models by varying the test-time training set size and updating all parameters of the GPT-2 model. The results are averaged over all the new tasks sampled from $\Sigma_{\beta_{TT}}$. Finally, we repeat the entire procedure

three times using different random seeds and report the averaged results.

The results in Figure 3b show that using a pre-trained model with TTT is more advantageous when the test-time training set size is small. However, as the training set size increases, training from scratch outperforms the pre-trained model, aligning with Figure 1b.

## 7. Discussion

We have developed a theoretical framework to characterize how a single-step gradient update at test time enhances in-context learning. In the case of an isotropic covariance matrix, we analyzed the improvement in loss under test-time training as a function of the number of in-context samples $(n)$, embedding dimension $(d)$, and test-time training set size $(k)$. We then extended our analysis to the non-isotropic covariance case, examining how the alignment between the new task parameter $(\beta_{TT})$ and the task covariance matrix $(\Sigma_\beta)$ affects performance. Our results reveal a phase-transition threshold on $k$ in both isotropic and non-isotropic settings, beyond which zero initialization (training from scratch) can outperform pre-trained initialization. Empirical results align well with our theoretical findings, and together they underscore that single-step test-time training offers significant performance improvements at low computational costs. Overall, our findings highlight test-time training as a lightweight, supervised enhancement to standard in-context learning.

As a future direction, one can investigate the analysis of TTT under a more general architecture and data model. Possible extensions to the architecture model might be to analyze the TTT with softmax attention or multilayer attention, whereas possible extensions to the data model might be to analyze K-Fold cross-validation instead of the current 1-Fold validation approach. Additionally, investigating TTT in unsupervised or semi-supervised in-context learning (ICL) settings presents another promising direction for extending this work. We hope that our results will serve as a foundation for future research on the interaction of TTT and ICL methods.

## Impact Statement

This paper presents work on test-time training to provide a stronger understanding of the principles of modern AI models. There are many potential societal consequences of our work, none which we feel must be specifically highlighted here.

# Acknowledgements

H.A.G., M.E.I., X.Z., and S.O. were supported in part by the NSF grants CCF2046816, CCF-2403075, CCF-2008020, and the Office of Naval Research grant N000142412289. M. M. is funded by the European Union (ERC, INF$^2$, project number 101161364). Views and opinions expressed are, however, those of the author(s) only and do not necessarily reflect those of the European Union or the European Research Council Executive Agency. Neither the European Union nor the granting authority can be held responsible for them. M.S. is supported by the Packard Fellowship in Science and Engineering, a Sloan Research Fellowship in Mathematics, an NSF-CAREER under award #1846369, DARPA FastNICS program, and NSF-CIF awards #1813877 and #2008443, and NIH DP2LM014564-01. The authors also acknowledge further support from Open Philanthropy, OpenAI, Amazon Research, Google Research, and Microsoft Research.

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

## A. Proofs for Section 3

**Proposition 3.1.** *Consider the linear attention model with parameters $W \in \mathbb{R}^{d \times d}$. Suppose the test-time training loss function is defined as in (2) and define $u_{context} := X_{context}^\top y_{context} \in \mathbb{R}^d$. Then, for any step size $\eta > 0$, the new parameter $W_{TT}$ after one gradient-descent step from $W$ is given by the rank-1 update*

$$W_{TT} = W + 2\eta\, X_{train}^\top (y_{train} - X_{train} W u_{context})\, u_{context}^\top.$$

*Proof.* We derive the update on the weight matrix $W$ via a single gradient step using the loss function defined over the set $Z_{\text{train}} = [X_{\text{train}}\ y_{\text{train}}] \in \mathbb{R}^{k \times (d+1)}$. Note that $\text{SM}(Z_i, W) = \bar{x}_{n+i}^\top W X_{\text{context}}^\top y_{\text{context}} = \bar{x}_{n+i}^\top W u_{\text{context}}$. Then, for the training set $Z_{\text{train}}$, the vector of predicted values under $W$ is:

$$\hat{y}_{\text{train}}(W) = X_{\text{train}} W u_{\text{context}} \in \mathbb{R}^k.$$

Recall the objective given by (2):

$$\hat{\mathcal{L}}(W) \;=\; \sum_{i=n+1}^{n+k} (\bar{y}_i - \text{SM}(Z_i, W))^2 \;=\; \sum_{i=n+1}^{n+k} (\bar{y}_i - \bar{x}_i^\top W u_{\text{context}})^2 \;=\; \|y_{\text{train}} - X_{\text{train}} W u_{\text{context}}\|_2^2.$$

The gradient of $\hat{\mathcal{L}}$ w.r.t. $W$ is then obtained by differentiating the summation:

$$\nabla_W \hat{\mathcal{L}} \;=\; -2 \sum_{i=n+1}^{n+k} \left( \bar{y}_i - \bar{x}_i^\top W u_{\text{context}} \right) \bar{x}_i u_{\text{context}}^\top \;=\; -2 X_{\text{train}}^\top (y_{\text{train}} - X_{\text{train}} W u_{\text{context}}) u_{\text{context}}^\top.$$

This is a rank-1 update as $(X_{\text{train}}^\top (y_{\text{train}} - X_{\text{train}} W u_{\text{context}})) u_{\text{context}}^\top = pq^\top$ is a rank-1 matrix. Evaluated at $W$, the update after one gradient step with step size $\eta > 0$ becomes:

$$\begin{aligned} W_{\text{TT}} &= W - \eta \nabla_W \hat{\mathcal{L}}(W) \\ &= W + 2\eta\, X_{\text{train}}^\top (y_{\text{train}} - X_{\text{train}} W u_{\text{context}}) u_{\text{context}}^\top. \end{aligned}$$

This completes the proof. $\qquad\square$

## B. Proofs for Section 4

**Lemma B.1** (Validity of Gaussian Approximation – Isotropic Case). *Let $X \in \mathbb{R}^{n \times d}$ have i.i.d $\mathcal{N}(0,1)$ entries and let $\|w\| = 1$. Define $q = \frac{1}{n} X^\top X w$. $q$ can be written as $q = w + g + e$ such that $g \sim \mathcal{N}(0, I_d/n)$ and $e$ is a residual random variable that obeys*

$$\mathbb{E}[\|e\|^2] \le 9 \cdot \frac{n+d}{n^2}.$$

*Note that $e$ represents a lower order term as we have $\|w\| = 1$ and $\mathbb{E}[\|g\|^2] = d/n$.*

*Proof.* Set $h = Xw \sim \mathcal{N}(0, I_n)$. Let $P = I - ww^\top$. Decompose $X^\top = ww^\top X^\top + PX^\top$ and observe that $PX^\top$ is independent of $ww^\top X^\top = wh^\top$. Additionally, introduce independent $v \sim \mathcal{N}(0, I_n)$ and set $X'^\top = PX^\top + wv^\top$. Finally, set $g = \sqrt{n} X'^\top h/\|h\|$ and

$$r = PX^\top h - g = \underbrace{PX^\top(h - \sqrt{n}h/\|h\|)}_{r_1} - \underbrace{\sqrt{n}wv^\top h/\|h\|}_{r_2}.$$

To proceed, observe that $X'$ has i.i.d $\mathcal{N}(0,1)$ entries and hence $g \sim \mathcal{N}(0, nI_d)$. We can write

$$X^\top X w = X^\top h = w\|h\|^2 + PX^\top h = w\|h\|^2 + g + r \tag{7}$$

$$= nw + g + r + (\|h\|^2 - n)w. \tag{8}$$

Let us now set $e' = r + (\|h\|^2 - n)w$ and investigate $\mathbb{E}[\|e'\|^2]$. Recalling $r = r_1 - r_2$, we can apply the standard upper bounds

$$\mathbb{E}[\|e'\|^2] \le 3\,\mathbb{E}[(\|h\|^2 - n)^2] + 3\,\mathbb{E}[\|r_1\|^2] + 3\,\mathbb{E}[\|r_2\|^2]. \tag{9}$$

Next, from standard facts about chi-square random variable with $n$ degrees of freedom, we have $\mathbb{E}[(\|\boldsymbol{h}\|^2 - n)^2] = 2n$. Similarly, set $\tilde{\boldsymbol{h}} = \boldsymbol{h} - \sqrt{n}\boldsymbol{h}/\|\boldsymbol{h}\|$. Note that $\mathbb{E}[\|\tilde{\boldsymbol{h}}\|^2] = \mathbb{E}[(\|\boldsymbol{h}\| - \sqrt{n})^2] = 2n - 2\sqrt{n}\,\mathbb{E}[\|\boldsymbol{h}\|] \leq 2$. This step uses the fact that $\mathbb{E}[\|\boldsymbol{h}\|] = \frac{\sqrt{2}\,\Gamma(\frac{n+1}{2})}{\Gamma(\frac{n}{2})} \geq \frac{n}{\sqrt{n+1}}$, which can be obtained by induction as argued in Section 3.1 of (Chandrasekaran et al., 2012). Furthermore, considering that $X'$ is the sum of two orthogonal vectors, we obtain

$$\mathbb{E}[\|\boldsymbol{r}_1\|^2] = \mathbb{E}[\|\boldsymbol{P}\boldsymbol{X}^\top(\boldsymbol{h} - \sqrt{n}\boldsymbol{h}/\|\boldsymbol{h}\|)\|^2] \leq \mathbb{E}[\|\boldsymbol{X}'^\top(\boldsymbol{h} - \sqrt{n}\boldsymbol{h}/\|\boldsymbol{h}\|)\|^2] = d\,\mathbb{E}[\|\tilde{\boldsymbol{h}}\|^2] \leq 2d.$$

Finally, set $z = \boldsymbol{v}^\top\boldsymbol{h}/\|\boldsymbol{h}\| \sim \mathcal{N}(0, 1)$. We bound

$$\mathbb{E}[\|\boldsymbol{r}_2\|^2] \leq \mathbb{E}[\|\sqrt{n}wz\|^2] = n.$$

Aggregating these, we obtain

$$\mathbb{E}[\|\boldsymbol{e}'\|^2] \leq 3(2n + 2d + n) = 9n + 6d.$$

To conclude, set $\boldsymbol{e} = \boldsymbol{e}'/n$. After normalizing $\boldsymbol{X}^\top\boldsymbol{X}\boldsymbol{w}$ by $n$ and the change of variable $\boldsymbol{g} \leftarrow \boldsymbol{g}/n$ with $\boldsymbol{g} \sim \mathcal{N}(0, \boldsymbol{I}_d/n)$, we obtain

$$n^{-1}\boldsymbol{X}^\top\boldsymbol{X}\boldsymbol{w} = \boldsymbol{w} + \boldsymbol{g} + \boldsymbol{e},$$

such that $\mathbb{E}[\|\boldsymbol{e}\|^2] = \mathbb{E}[\|\boldsymbol{e}'\|^2]/n^2 \leq 9/n + 6d/n^2$. $\qquad\square$

**Lemma B.2** (Population Loss Expression). *Consider the linear attention model as the sequence model described in Section 3 with weight matrix* $\boldsymbol{W}$*. Suppose that the new task during the test-time training is* $\boldsymbol{\beta}_{TT}$*. In that case, the population loss of the model in* (3) *with respect to this task is given by:*

$$\mathcal{L}(\boldsymbol{W}) = \mathbb{E}_{(\boldsymbol{x},y),(\boldsymbol{x}_i,y_i)\sim\mathcal{D}_z^{TT}(\boldsymbol{\beta}_{TT})}\left[\left(y - \boldsymbol{x}^\top\boldsymbol{W}\boldsymbol{X}^\top\boldsymbol{Y}\right)^2\right]$$
$$= \boldsymbol{\beta}_{TT}^\top\left[\boldsymbol{\Sigma}_{\boldsymbol{x}} - n\boldsymbol{\Sigma}_{\boldsymbol{x}}\boldsymbol{W}\boldsymbol{\Sigma}_{\boldsymbol{x}} - n\boldsymbol{\Sigma}_{\boldsymbol{x}}\boldsymbol{W}^\top\boldsymbol{\Sigma}_{\boldsymbol{x}} + n(n+1)\boldsymbol{\Sigma}_{\boldsymbol{x}}\boldsymbol{W}^\top\boldsymbol{\Sigma}_{\boldsymbol{x}}\boldsymbol{W}\boldsymbol{\Sigma}_{\boldsymbol{x}} + n\,\mathrm{tr}(\boldsymbol{W}^\top\boldsymbol{\Sigma}_{\boldsymbol{x}}\boldsymbol{W}\boldsymbol{\Sigma}_{\boldsymbol{x}})\boldsymbol{\Sigma}_{\boldsymbol{x}}\right]\boldsymbol{\beta}_{TT} + \sigma^2 n\,\mathrm{tr}(\boldsymbol{W}^\top\boldsymbol{\Sigma}_{\boldsymbol{x}}\boldsymbol{W}\boldsymbol{\Sigma}_{\boldsymbol{x}}) + \sigma^2.$$

*Proof.* Since the target function $f$ is linear, the labels are given by:

$$y_i = f(\boldsymbol{x}_i) + \xi_i = \boldsymbol{\beta}_{TT}^\top\boldsymbol{x}_i + \xi_i.$$

We collect the samples into $\boldsymbol{X} \in \mathbb{R}^{n\times d}$ and $\boldsymbol{Y} \in \mathbb{R}^n$ as before, so that:

$$\boldsymbol{Y} = \boldsymbol{X}\boldsymbol{\beta}_{TT} + \boldsymbol{\xi},$$

where $\boldsymbol{\xi} = (\xi_1, \ldots, \xi_n)^\top$ represents noise on the context labels. Recall the prediction by the model:

$$\mathrm{SM}(\boldsymbol{Z}, \boldsymbol{W}) = \boldsymbol{x}^\top\boldsymbol{W}\boldsymbol{X}^\top\boldsymbol{Y}.$$

Then, we have the following population loss for the new task $\boldsymbol{\beta}_{TT}$:

$$\mathcal{L}(\boldsymbol{W}) = \mathbb{E}_{(\boldsymbol{x},y),(\boldsymbol{x}_i,y_i)\sim\mathcal{D}_z^{TT}(\boldsymbol{\beta}_{TT})}\left[(\boldsymbol{x}^\top\boldsymbol{\beta}_{TT} - \boldsymbol{x}^\top\boldsymbol{W}\boldsymbol{X}^\top\boldsymbol{Y})^2\right] + \sigma^2.$$

Since $\boldsymbol{Y} = \boldsymbol{X}\boldsymbol{\beta}_{TT} + \boldsymbol{\xi}$, we have:

$$\mathrm{SM}(\boldsymbol{Z}, \boldsymbol{W}) = \boldsymbol{x}^\top\boldsymbol{W}\boldsymbol{X}^\top\boldsymbol{X}\boldsymbol{\beta}_{TT} + \boldsymbol{x}^\top\boldsymbol{W}\boldsymbol{X}^\top\boldsymbol{\xi}.$$

The error is:

$$\boldsymbol{x}^\top\boldsymbol{\beta}_{TT} - \boldsymbol{x}^\top\boldsymbol{W}\boldsymbol{X}^\top\boldsymbol{X}\boldsymbol{\beta}_{TT} - \boldsymbol{x}^\top\boldsymbol{W}\boldsymbol{X}^\top\boldsymbol{\xi} = \boldsymbol{x}^\top(\boldsymbol{\beta}_{TT} - \boldsymbol{W}\boldsymbol{X}^\top\boldsymbol{X}\boldsymbol{\beta}_{TT}) - \boldsymbol{x}^\top\boldsymbol{W}\boldsymbol{X}^\top\boldsymbol{\xi}$$
$$= \boldsymbol{x}^\top(\boldsymbol{I} - \boldsymbol{W}\boldsymbol{X}^\top\boldsymbol{X})\boldsymbol{\beta}_{TT} - \boldsymbol{x}^\top\boldsymbol{W}\boldsymbol{X}^\top\boldsymbol{\xi}.$$

Therefore, the population loss is equal to:

$$\mathcal{L}(\boldsymbol{W}) = \mathbb{E}_{(\boldsymbol{x},y),(\boldsymbol{x}_i,y_i)\sim\mathcal{D}_z^{TT}(\boldsymbol{\beta}_{TT})}\left[\left(\boldsymbol{x}^\top(\boldsymbol{I} - \boldsymbol{W}\boldsymbol{X}^\top\boldsymbol{X})\boldsymbol{\beta}_{TT} - \boldsymbol{x}^\top\boldsymbol{W}\boldsymbol{X}^\top\boldsymbol{\xi}\right)^2\right] + \sigma^2$$
$$= \mathbb{E}_{\boldsymbol{X}}\left[\mathbb{E}_{\boldsymbol{x},\boldsymbol{\xi}}\left[\left(\boldsymbol{x}^\top(\boldsymbol{I} - \boldsymbol{W}\boldsymbol{X}^\top\boldsymbol{X})\boldsymbol{\beta}_{TT} - \boldsymbol{x}^\top\boldsymbol{W}\boldsymbol{X}^\top\boldsymbol{\xi}\right)^2\right]\right] + \sigma^2$$

We will first take the expectation with respect to query $x$ and noise vector $\xi$. Then, using independence, we will take the expectation of the resulting expression with respect to the context matrix $X$. Expanding the square gives

$$\left(x^\top(I - WX^\top X)\beta_{\text{TT}} - x^\top WX^\top \xi\right)^2 = (x^\top(I - WX^\top X)\beta_{\text{TT}})^2 - 2(x^\top(I - WX^\top X)\beta_{\text{TT}})(x^\top WX^\top \xi) + (x^\top WX^\top \xi)^2.$$

Now, let's consider each of these three terms individually. First of all, the expectation of cross-term is:

$$\mathbb{E}_{x,\xi}[-2(x^\top(I - WX^\top X)\beta_{\text{TT}})(x^\top WX^\top \xi)] = 0.$$

Besides, since the first term does not depend on $\xi$, we have:

$$\mathbb{E}_{x,\xi}[(x^\top(I - WX^\top X)\beta_{\text{TT}})^2] = \beta_{\text{TT}}^\top(I - WX^\top X)^\top \Sigma_x(I - WX^\top X)\beta_{\text{TT}}.$$

Consider now the last term $(x^\top WX^\top \xi)^2$. Recall that $\xi$ is zero-mean noise with $\mathbb{E}[\xi\xi^\top] = \sigma^2 I_n$, and $\mathbb{E}[xx^\top] = \Sigma_x$. First, condition on $x$:

$$(x^\top WX^\top \xi)^2 = \xi^\top(XW^\top xx^\top WX^\top)\xi.$$

Taking expectation over $\xi$:

$$\mathbb{E}_\xi[(x^\top WX^\top \xi)^2 \mid x] = \sigma^2\text{tr}(XW^\top xx^\top WX^\top).$$

Use the cyclic property of trace and the fact that $xx^\top$ is rank one:

$$\text{tr}(XW^\top xx^\top WX^\top) = \text{tr}(xx^\top WX^\top XW^\top) = x^\top(WX^\top XW^\top)x.$$

Thus, we obtain:

$$\mathbb{E}_\xi[(x^\top WX^\top \xi)^2 \mid x] = \sigma^2 x^\top(WX^\top XW^\top)x.$$

Taking expectation over $x$ yields:

$$\mathbb{E}_x[x^\top(WX^\top XW^\top)x] = \text{tr}(WX^\top XW^\top \Sigma_x),$$

since $\mathbb{E}[xx^\top] = \Sigma_x$. Combining both expectations:

$$\mathbb{E}_{x,\xi}[(x^\top WX^\top \xi)^2] = \sigma^2\text{tr}(WX^\top XW^\top \Sigma_x).$$

Thus, combining the three terms yields the following expression:

$$\beta_{\text{TT}}^\top(I - WX^\top X)^\top \Sigma_x(I - WX^\top X)\beta_{\text{TT}} + \sigma^2\text{tr}(WX^\top XW^\top \Sigma_x)$$

Now, we aim to compute the loss function averaged over all possible in-context example sets $X$:

$$\mathcal{L}(W) = \mathbb{E}_X\left[\beta_{\text{TT}}^\top\left(I - WX^\top X\right)^\top \Sigma_x\left(I - WX^\top X\right)\beta_{\text{TT}} + \sigma^2\text{tr}(WX^\top XW^\top \Sigma_x)\right] + \sigma^2.$$

Rewrite the first expression inside the expectation as

$$\beta_{\text{TT}}^\top A^\top \Sigma_x A \beta_{\text{TT}} \quad \text{where} \quad A = I - WX^\top X.$$

Then

$$\mathbb{E}_X\left[\beta_{\text{TT}}^\top A^\top \Sigma_x A \beta_{\text{TT}}\right] = \beta_{\text{TT}}^\top\left(\mathbb{E}_X[A^\top \Sigma_x A]\right)\beta_{\text{TT}}.$$

We can write

$$A^\top \Sigma_x A = \left(I - X^\top XW^\top\right)\Sigma_x\left(I - WX^\top X\right).$$

Hence

$$\mathbb{E}_X[A \Sigma_x A] = \Sigma_x - \Sigma_x W \mathbb{E}[X^\top X] - \mathbb{E}[X^\top X]W^\top \Sigma_x + \mathbb{E}[(X^\top X)W^\top \Sigma_x W(X^\top X)].$$

We know that $\mathbb{E}[X^\top X] = n\Sigma_x$, thus, our expression becomes:

$$\mathbb{E}_X[A \Sigma_x A] = \Sigma_x - n\Sigma_x W \Sigma_x - n\Sigma_x W^\top \Sigma_x + \mathbb{E}[(X^\top X)W^\top \Sigma_x W(X^\top X)].$$

Using Lemma B.3, we know the following fact:

$$\mathbb{E}_X\left[(X^\top X)M(X^\top X)\right] = n(n+1)\Sigma_x M\Sigma_x + n\text{tr}(M\Sigma_x)\Sigma_x.$$

Finally, the overall expression becomes:

$$\mathbb{E}_X[A\,\Sigma_x\,A] \;=\; \Sigma_x \;-\; n\Sigma_x\,W\,\Sigma_x \;-\; n\Sigma_x W^\top\,\Sigma_x \;+\; n(n+1)\Sigma_x W^\top\Sigma_x W\Sigma_x + n\mathrm{tr}(W^\top\Sigma_x W\Sigma_x)\Sigma_x.$$

Adding the constant terms to the expectation gives:

$$\beta_{\mathrm{TT}}^\top\left[\Sigma_x \;-\; n\Sigma_x\,W\,\Sigma_x \;-\; n\Sigma_x W^\top\,\Sigma_x \;+\; n(n+1)\Sigma_x W^\top\Sigma_x W\Sigma_x + n\mathrm{tr}(W^\top\Sigma_x W\Sigma_x)\Sigma_x\right]\beta_{\mathrm{TT}}.$$

Now that we have calculated the expectation of the first term, let's focus on the noise term involving trace inside the expectation. We have:

$$\mathbb{E}_X\left[\mathrm{tr}(WX^\top XW^\top\Sigma_x)\right] = n\mathrm{tr}(W^\top\Sigma_x W\Sigma_x).$$

As a result, we arrive at the final expression:

$$\begin{aligned}
\mathcal{L}(W) &= \mathbb{E}_{X,(x,y)\sim\mathcal{D}}\left[\left((x^\top\beta_{\mathrm{TT}} + \xi - x^\top WX^\top Y)^2\right)\right]\\
&= \beta_{\mathrm{TT}}^\top\left[\Sigma_x \;-\; n\Sigma_x\,W\,\Sigma_x \;-\; n\Sigma_x W^\top\,\Sigma_x \;+\; n(n+1)\Sigma_x W^\top\Sigma_x W\Sigma_x + n\mathrm{tr}(W^\top\Sigma_x W\Sigma_x)\Sigma_x\right]\beta_{\mathrm{TT}}\\
&\quad + \sigma^2 n\mathrm{tr}(W^\top\Sigma_x W\Sigma_x) + \sigma^2.
\end{aligned}$$

$\square$

**Lemma B.3.** *Let $X \in \mathbb{R}^{n\times d}$ be a random matrix whose rows $x_i \in \mathbb{R}^d$, for $i = 1,\ldots,n$, are i.i.d. drawn from $\mathcal{N}(0,\Sigma)$. Let $M \in \mathbb{R}^{d\times d}$ be any fixed symmetric matrix. Then, under these assumptions,*

$$\mathbb{E}_X\left[(X^\top X)\,M\,(X^\top X)\right] \;=\; n(n+1)\,\Sigma\,M\,\Sigma \;+\; n\,\mathrm{tr}(M\Sigma)\,\Sigma.$$

*Proof.* We can write the matrix $X^\top X$ as:

$$X^\top X \;=\; \sum_{i=1}^{n} x_i\,x_i^\top.$$

Therefore,

$$(X^\top X)\,M\,(X^\top X) \;=\; \left(\sum_{i=1}^{n} x_i x_i^\top\right) M\left(\sum_{j=1}^{n} x_j x_j^\top\right) \;=\; \sum_{i=1}^{n}\sum_{j=1}^{n} x_i\left(x_i^\top M\,x_j\right)x_j^\top.$$

Taking expectation over $X$, we split the sum into the diagonal part ($i = j$) and the off-diagonal part ($i \neq j$):

$$\mathbb{E}_X\left[(X^\top X)\,M\,(X^\top X)\right] = \underbrace{\sum_{i=1}^{n}\mathbb{E}\left[x_i\,(x_i^\top M\,x_i)\,x_i^\top\right]}_{\text{(i) diagonal terms}} \;+\; \underbrace{\sum_{i\neq j}\mathbb{E}\left[x_i\,(x_i^\top M\,x_j)\,x_j^\top\right]}_{\text{(ii) off-diagonal terms}}.$$

**(i) Diagonal Terms ($i = j$).** Each $x_i \sim \mathcal{N}(0,\Sigma)$, so we set

$$\mathbb{E}\left[x_i(x_i^\top M\,x_i)\,x_i^\top\right] \;=\; \mathbb{E}\left[\left(x_i^\top M\,x_i\right)x_i\,x_i^\top\right].$$

Recall the standard result, which follows from Wick's Theorem (Petersen & Pedersen, 2012)[Section 8.2.4] for a zero-mean Gaussian $x \sim \mathcal{N}(0,\Sigma)$ and any symmetric matrix $A$:

$$\mathbb{E}\left[x(x^\top A\,x)\,x^\top\right] \;=\; 2\Sigma A\Sigma + \mathrm{tr}(A\Sigma)\Sigma.$$

Substituting $A = M$, it follows that

$$\mathbb{E}\left[x_i(x_i^\top M\,x_i)\,x_i^\top\right] = 2\Sigma M\Sigma + \mathrm{tr}(M\Sigma)\Sigma.$$

Since we have $n$ diagonal terms, summing over $i = 1,\ldots,n$ yields

$$\sum_{i=1}^{n}\mathbb{E}\left[x_i\,(x_i^\top M\,x_i)\,x_i^\top\right] = n\left[2\Sigma M\Sigma + \mathrm{tr}(M\Sigma)\Sigma\right].$$

**(ii) Off-Diagonal Terms** ($i \neq j$). For $i \neq j$, the vectors $x_i$ and $x_j$ are independent. Then,

$$\mathbb{E}\left[x_i \left(x_i^\top M x_j\right) x_j^\top\right] = \mathbb{E}[x_i x_i^\top] M \mathbb{E}[x_j x_j^\top] = \Sigma M \Sigma.$$

Hence, considering that we have $n$ diagonal terms and $n(n-1)$ off-diagonal terms, combining (i) + (ii) gives:

$$\mathbb{E}_X\left[(X^\top X) M (X^\top X)\right] = n\left[2\Sigma M \Sigma + \text{tr}(M\Sigma)\Sigma\right] + n(n-1)\Sigma M \Sigma = n(n+1)\Sigma M \Sigma + n\text{tr}(M\Sigma)\Sigma.$$

This completes the proof. $\qquad \square$

**Proposition 4.1.** *Let $\Sigma_\beta, \Sigma_x = I, \sigma^2 = 0$ and let $\beta_{TT}$ be an arbitrary new task parameter. Then, the optimal pre-trained model's parameter $W^*$ and its population loss are given by*

$$W^* = \frac{1}{n+d+1}I, \qquad \mathcal{L}(W^*) = \|\beta_{TT}\|^2 \frac{d+1}{n+d+1}.$$

*Proof.* Recall the loss expression from Lemma B.2:

$$\mathcal{L}(W) = \beta_{TT}^\top\left[\Sigma_x - n\Sigma_x W\Sigma_x - n\Sigma_x W^\top \Sigma_x + n(n+1)\Sigma_x W^\top \Sigma_x W\Sigma_x + n\text{tr}(W^\top\Sigma_x W\Sigma_x)\Sigma_x\right]\beta_{TT} + \sigma^2 n\text{tr}(W^\top\Sigma_x W\Sigma_x) + \sigma^2.$$

We will simplify the expression under the assumptions $\Sigma_x, \Sigma_\beta = I$ and $W^* = \frac{1}{n+d+1+\sigma^2}I$, where the optimal pre-trained $W^*$ follows from Theorem 1 of Li et al. (2024). The first expression of interest is:

$$\Sigma_x - n\Sigma_x W\Sigma_x - n\Sigma_x W^\top \Sigma_x + n(n+1)\Sigma_x W^\top \Sigma_x W\Sigma_x + n\text{tr}(W^\top\Sigma_x W\Sigma_x)\Sigma_x.$$

Substituting $\Sigma_x$ and $W^*$ yields:

$$I - \frac{n}{n+d+1+\sigma^2}I - \frac{n}{n+d+1+\sigma^2}I + \frac{n(n+1)}{(n+d+1+\sigma^2)^2}I + \frac{nd}{(n+d+1+\sigma^2)^2}I$$

$$= \frac{(d+1+\sigma^2-n)(n+d+1+\sigma^2) + n(n+d+1)}{(n+d+1+\sigma^2)^2}I$$

$$= \frac{(d+1+\sigma^2)(n+d+1+\sigma^2) - \sigma^2 n}{(n+d+1+\sigma^2)^2}I.$$

In addition, the noise term is:

$$\sigma^2 n\text{tr}(W^{*\top}\Sigma_x W^*\Sigma_x) = \sigma^2 \frac{nd}{(n+d+1+\sigma^2)^2}.$$

Combining these, the final closed-form loss is

$$\mathcal{L}(W^*) = \|\beta_{TT}\|^2 \frac{(d+1+\sigma^2)(n+d+1+\sigma^2) - \sigma^2 n}{(n+d+1+\sigma^2)^2} + \sigma^2 \frac{nd}{(n+d+1+\sigma^2)^2} + \sigma^2.$$

Finally, setting $\sigma^2 = 0$ in this expression yields the desired result. $\qquad \square$

**Theorem 4.2.** *Let $n/d = \Theta(1)$. Recall the definition of the expected loss $\mathcal{L}_{TT}(W_{TT})$ with respect to $\mathcal{S}_{TT}$ in (4). In the isotropic covariance and noiseless setting ($\sigma^2 = 0$), the optimal step-size that minimizes $\mathcal{L}_{TT}(W_{TT})$ is*

$$\eta^* \approx \frac{d}{2(k+d)\,n^2\,(n+d)\,\|\beta_{TT}\|_2^2}.$$

*With this optimal step-size $\eta^*$, the improvement in the loss due to the test-time training is*

$$\mathcal{L}(W^*) - \mathcal{L}_{TT}(W_{TT}) \approx \frac{k}{k+d}\frac{d^3}{(n+d)^3}\,\|\beta_{TT}\|_2^2.$$

*Proof.* We consider the noiseless ($\sigma^2 = 0$) and *isotropic* case where $\Sigma_x = \Sigma_\beta = I$. Throughout the proof, we omit lower order terms like $+1$ in $(d + 1)$ or $2n$ in $(n^2 + 2n)$ as they're negligible compared to the higher order of the same variable (see Remark 4.3 for error discussion). This convention allows us to simplify the subsequent expressions.

Now, we aim to calculate the expected gain in the loss with respect to $(n + k)$ samples used during the test-time training process. That is, we will take the expectation of the loss function given by Lemma B.2 and compute $\mathcal{L}(W^*) - \mathcal{L}_{TT}(W_{TT}) = \mathbb{E}_{X_{\text{train}}, X_{\text{context}}} [\Delta\mathcal{L}]$ where $\Delta\mathcal{L}$ is given by:

$$
\begin{aligned}
\Delta\mathcal{L} &= \mathcal{L}(W^*) - \mathcal{L}(W_{TT}) \\
&= \beta_{TT}^\top \Big[ \Sigma_x - n\Sigma_x W^* \Sigma_x - n\Sigma_x W^{*\top} \Sigma_x + n(n+1)\Sigma_x W^{*\top} \Sigma_x W^* \Sigma_x + n\operatorname{tr}(W^{*\top}\Sigma_x W^* \Sigma_x)\Sigma_x \Big] \beta_{TT} \\
&\quad - \beta_{TT}^\top \Big[ \Sigma_x - n\Sigma_x W_{TT} \Sigma_x - n\Sigma_x W_{TT}^\top \Sigma_x + n(n+1)\Sigma_x W_{TT}^\top \Sigma_x W_{TT}\Sigma_x + n\operatorname{tr}(W_{TT}^\top \Sigma_x W_{TT}\Sigma_x)\Sigma_x \Big] \beta_{TT} \\
&= \beta_{TT}^\top \Big[ n\Sigma_x (W_{TT} - W^*) \Sigma_x + n\Sigma_x \left(W_{TT}^\top - W^{*\top}\right) \Sigma_x + n(n+1)\Sigma_x \left(W^{*\top}\Sigma_x W^* - W_{TT}^\top \Sigma_x W_{TT}\right)\Sigma_x \\
&\quad + n\left(\operatorname{tr}(W^{*\top}\Sigma_x W^* \Sigma_x) - \operatorname{tr}(W_{TT}^\top \Sigma_x W_{TT}\Sigma_x)\right)\Sigma_x \Big] \beta_{TT}. \quad (10)
\end{aligned}
$$

In the noiseless and $\Sigma_x = \Sigma_\beta = I$ case, the optimal pre-trained $W^*$ becomes $W^* = \frac{1}{n+d+1}I$ as shown in Theorem 1 of Li et al. (2024). Recall that when there's no noise and the linear task is $\beta_{TT}$, the gradient update given by Proposition 3.1 becomes:

$$
W_{TT} = W^* + 2\eta X_{\text{train}}^\top X_{\text{train}}(\beta_{TT} - W^* X_{\text{context}}^\top X_{\text{context}} \beta_{TT})\beta_{TT}^\top X_{\text{context}}^\top X_{\text{context}}.
$$

A key technical challenge while using the above update is that the expectation of $\Delta\mathcal{L}$ in (10) involves $8th$ moments of $X_{\text{context}}$. To overcome this problem, we will use the Gaussian approximation following Lemma B.1 by approximating $X_{\text{context}}^\top X_{\text{context}} \beta_{TT} \approx n\Sigma_x \beta_{TT} + \sqrt{n}\Sigma_x^{1/2}g$ where $g \sim \mathcal{N}(0, \|\beta_{TT}\|^2 I)$. This non-isotropic form can be obtained by factoring out $\Sigma_x^{1/2}$ and reducing the problem to the isotropic case from Lemma B.1. Also define $\Delta W := W_{TT} - W^*$. Then, using the approximation and the fact that $\Sigma_x = I$, we can write:

$$
\begin{aligned}
\Delta W &= 2\eta X_{\text{train}}^\top X_{\text{train}} \left(\beta_{TT} - \frac{n}{n+d+1}\Sigma_x \beta_{TT} - \frac{\sqrt{n}}{n+d+1}\Sigma_x^{1/2}g\right)\left(n\beta_{TT}^\top \Sigma_x + \sqrt{n}g^\top \Sigma_x^{1/2}\right) \\
&= 2\eta X_{\text{train}}^\top X_{\text{train}} \Big(n\beta_{TT}\beta_{TT}^\top \Sigma_x + \sqrt{n}\beta_{TT}g^\top \Sigma_x^{1/2} - \frac{n^2}{n+d+1}\Sigma_x \beta_{TT}\beta_{TT}^\top \Sigma_x - \frac{n\sqrt{n}}{n+d+1}\Sigma_x \beta_{TT}g^\top \Sigma_x^{1/2} \\
&\quad - \frac{n\sqrt{n}}{n+d+1}\Sigma_x^{1/2}g\beta_{TT}^\top \Sigma_x - \frac{n}{n+d+1}\Sigma_x^{1/2}gg^\top \Sigma_x^{1/2}\Big) \\
&= 2\eta X_{\text{train}}^\top X_{\text{train}} \left(\frac{n(d+1)}{n+d+1}\beta_{TT}\beta_{TT}^\top + \frac{\sqrt{n}(d+1)}{n+d+1}\beta_{TT}g^\top - \frac{n\sqrt{n}}{n+d+1}g\beta_{TT}^\top - \frac{n}{n+d+1}gg^\top\right) \\
\Delta W^\top &= 2\eta \left(\frac{n(d+1)}{n+d+1}\beta_{TT}\beta_{TT}^\top + \frac{\sqrt{n}(d+1)}{n+d+1}g\beta_{TT}^\top - \frac{n\sqrt{n}}{n+d+1}\beta_{TT}g^\top - \frac{n}{n+d+1}gg^\top\right)X_{\text{train}}^\top X_{\text{train}}.
\end{aligned}
$$

Notice that plugging $\Sigma_x = I$ in Equation (10), we observe the following difference term in two of the expressions:

$$
\begin{aligned}
W^{*\top}W^* - W_{TT}^\top W_{TT} &= W^{*\top}\Sigma_x W^* - (W^{*\top} + \Delta W^\top)\Sigma_x (W^* + \Delta W) \\
&= -W^{*\top}\Delta W - \Delta W^\top W^* - \Delta W^\top \Delta W.
\end{aligned}
$$

Hence, we need to calculate both first-order expectations $\mathbb{E}_{X_{\text{train}}, g}[\Delta W], \mathbb{E}_{X_{\text{train}}, g}[\Delta W^\top]$ and second-order expectation $\mathbb{E}_{X_{\text{train}}, g}[\Delta W^\top \Delta W]$. By direct calculation, we obtain:

$$
\mathbb{E}_{X_{\text{train}}, g}[\Delta W] = 2\eta k \left(\frac{n(d+1)}{n+d+1}\beta_{TT}\beta_{TT}^\top - \frac{n}{n+d+1}\|\beta_{TT}\|_2^2 I\right) = \mathbb{E}_{X_{\text{train}}, g}\left[\Delta W^\top\right]. \quad (11)
$$

For the second-order expectation $\mathbb{E}_{X_{\text{train}}}[X_{\text{train}}^\top X_{\text{train}} X_{\text{train}}^\top X_{\text{train}}]$, we will benefit from the fact that $\mathbb{E}_X[(X^\top X)(X^\top X)] = k(k+1)\Sigma^2 + k\operatorname{tr}(\Sigma)\Sigma$ where $X \in \mathbb{R}^{k \times d}$ and its rows are drawn i.i.d from $\mathcal{N}(0, \Sigma)$. This result follows by plugging $M = I$ into Lemma B.3. In our setting, $\Sigma_x = I$, which gives

$$
\mathbb{E}_{X_{\text{train}}}\left[X_{\text{train}}^\top X_{\text{train}} X_{\text{train}}^\top X_{\text{train}}\right] = k(k+d+1)I.
$$

Using the above result, we obtain:

$$\mathbb{E}_{X_{\text{train}}, g}\left[\Delta W^\top \Delta W\right]$$

$$= 4\eta^2 \mathbb{E}_g\left[\left(\frac{n(d+1)}{n+d+1}\beta_{\text{TT}}\beta_{\text{TT}}^\top + \frac{\sqrt{n}(d+1)}{n+d+1}g\beta_{\text{TT}}^\top - \frac{n\sqrt{n}}{n+d+1}\beta_{\text{TT}}g^\top - \frac{n}{n+d+1}gg^\top\right)k(k+d+1)I\right.$$

$$\left.\left(\frac{n(d+1)}{n+d+1}\beta_{\text{TT}}\beta_{\text{TT}}^\top + \frac{\sqrt{n}(d+1)}{n+d+1}\beta_{\text{TT}}g^\top - \frac{n\sqrt{n}}{n+d+1}g\beta_{\text{TT}}^\top - \frac{n}{n+d+1}gg^\top\right)\right]$$

$$= 4\eta^2 k(k+d+1)\mathbb{E}_g\left[\left(\frac{n(d+1)}{n+d+1}\beta_{\text{TT}}\beta_{\text{TT}}^\top + \frac{\sqrt{n}(d+1)}{n+d+1}g\beta_{\text{TT}}^\top - \frac{n\sqrt{n}}{n+d+1}\beta_{\text{TT}}g^\top - \frac{n}{n+d+1}gg^\top\right)\right.$$

$$\left.\left(\frac{n(d+1)}{n+d+1}\beta_{\text{TT}}\beta_{\text{TT}}^\top + \frac{\sqrt{n}(d+1)}{n+d+1}\beta_{\text{TT}}g^\top - \frac{n\sqrt{n}}{n+d+1}g\beta_{\text{TT}}^\top - \frac{n}{n+d+1}gg^\top\right)\right]$$

$$= 4\eta^2 \frac{k(k+d+1)n}{(n+d+1)^2}\mathbb{E}_g\left[\left(\sqrt{n}(d+1)\beta_{\text{TT}}\beta_{\text{TT}}^\top + (d+1)g\beta_{\text{TT}}^\top - n\beta_{\text{TT}}g^\top - \sqrt{n}gg^\top\right)\left(\sqrt{n}(d+1)\beta_{\text{TT}}\beta_{\text{TT}}^\top\right.\right.$$

$$\left.\left. + (d+1)\beta_{\text{TT}}g^\top - ng\beta_{\text{TT}}^\top - \sqrt{n}gg^\top\right)\right]$$

$$= 4\eta^2 \frac{k(k+d+1)n}{(n+d+1)^2}\mathbb{E}_g\left[n(d+1)^2\|\beta_{\text{TT}}\|_2^2\beta_{\text{TT}}\beta_{\text{TT}}^\top - n(d+1)\beta_{\text{TT}}\beta_{\text{TT}}^\top gg^\top + (d+1)^2\|\beta_{\text{TT}}\|_2^2 gg^\top\right.$$

$$\left. + n^2\|g\|_2^2\beta_{\text{TT}}\beta_{\text{TT}}^\top - n(d+1)gg^\top\beta_{\text{TT}}\beta_{\text{TT}}^\top - n(d+1)\beta_{\text{TT}}\beta_{\text{TT}}^\top gg^\top - n(d+1)gg^\top\beta_{\text{TT}}\beta_{\text{TT}}^\top + ngg^\top gg^\top\right]$$

$$= 4\eta^2 \frac{k(k+d+1)n}{(n+d+1)^2}\left[n(d+1)^2\|\beta_{\text{TT}}\|_2^2\beta_{\text{TT}}\beta_{\text{TT}}^\top - n(d+1)\|\beta_{\text{TT}}\|_2^2\beta_{\text{TT}}\beta_{\text{TT}}^\top + (d+1)^2\|\beta_{\text{TT}}\|_2^4 I\right.$$

$$\left. + n^2 d\|\beta_{\text{TT}}\|_2^2\beta_{\text{TT}}\beta_{\text{TT}}^\top - 3n(d+1)\|\beta_{\text{TT}}\|_2^2\beta_{\text{TT}}\beta_{\text{TT}}^\top + n(d+2)\|\beta_{\text{TT}}\|_2^4 I\right]$$

$$= 4\eta^2 \frac{k(k+d+1)n}{(n+d+1)^2}\left[n\left((d+1)(d-3) + nd\right)\|\beta_{\text{TT}}\|_2^2\beta_{\text{TT}}\beta_{\text{TT}}^\top + \left((d+1)^2 + n(d+2)\right)\|\beta_{\text{TT}}\|_2^4 I\right].$$

Applying the first-order expectation results from (11), we obtain that the first two terms of the loss difference are:

$$\mathbb{E}_{X_{\text{train}}, g}\left[\beta_{\text{TT}}^\top n\Sigma_x\left(W_{\text{TT}} - W^*\right)\Sigma_x\beta_{\text{TT}}\right] = 2\eta kn\left(\frac{n(d+1)}{n+d+1}\|\beta_{\text{TT}}\|_2^4 - \frac{n}{n+d+1}\|\beta_{\text{TT}}\|_2^4\right)$$

$$= 2\eta kn\frac{nd}{n+d+1}\|\beta_{\text{TT}}\|_2^4, \tag{12}$$

$$\mathbb{E}_{X_{\text{train}}, g}\left[\beta_{\text{TT}}^\top n\Sigma_x\left(W_{\text{TT}}^\top - W^{*\top}\right)\Sigma_x\beta_{\text{TT}}\right] = 2\eta kn\frac{nd}{n+d+1}\|\beta_{\text{TT}}\|_2^4. \tag{13}$$

The other two terms in Equation (10) are:

$$\mathbb{E}_{X_{\text{train}}, g}\left[\beta_{\text{TT}}^\top n(n+1)\Sigma_x\left(W^{*\top}\Sigma_x W^* - W_{\text{TT}}^\top \Sigma_x W_{\text{TT}}\right)\Sigma_x\beta_{\text{TT}}\right]$$

$$= n(n+1)\beta_{\text{TT}}^\top \mathbb{E}_{X_{\text{train}}, g}\left[-W^{*\top}\Delta W - \Delta W^\top W^* - \Delta W^\top \Delta W\right]\beta_{\text{TT}}$$

$$= -4\eta kn(n+1)\frac{nd}{(n+d+1)^2}\|\beta_{\text{TT}}\|_2^4 - 4\eta^2\frac{k(k+d+1)n^2(n+1)}{(n+d+1)^2}\left[n\left(d^2 - 2d - 3 + nd\right)\|\beta_{\text{TT}}\|_2^6 + \left((d+1)^2 + n(d+2)\right)\|\beta_{\text{TT}}\|_2^6\right], \tag{14}$$

and

$$\mathbb{E}_{X_{\text{train}}, g}\left[\beta_{\text{TT}}^\top n\left(\text{tr}(W^{*\top}\Sigma_x W^*\Sigma_x) - \text{tr}(W_{\text{TT}}^\top \Sigma_x W_{\text{TT}}\Sigma_x)\right)\Sigma_x\beta_{\text{TT}}\right]$$

$$= n\|\beta_{\text{TT}}\|_2^2 \mathbb{E}_{X_{\text{train}}, g}\left[\text{tr}(-W^{*\top}\Delta W - \Delta W^\top W^* - \Delta W^\top \Delta W)\right]$$

$$= -4\eta^2\frac{k(k+d+1)n^2}{(n+d+1)^2}\left[n\left(d^2 - 2d - 3 + nd\right)\|\beta_{\text{TT}}\|_2^6 + d\left((d+1)^2 + n(d+2)\right)\|\beta_{\text{TT}}\|_2^6\right] - 4\eta k\frac{n^2}{(n+d+1)^2}\|\beta_{\text{TT}}\|_2^4. \tag{15}$$

Combining (12), (13), (14), (15) in Equation (10), the expected improvement in the loss is the following:

$$\mathcal{L}(W^*) - \mathcal{L}_{TT}(W_{\text{TT}}) = 4\eta k\frac{n^2 d}{n+d+1}\|\beta_{\text{TT}}\|_2^4 - 4\eta k\frac{n^2}{(n+d+1)^2}\|\beta_{\text{TT}}\|_2^4 - 4\eta k\frac{n^2(n+1)d}{(n+d+1)^2}\|\beta_{\text{TT}}\|_2^4$$

$$- 4\eta^2\frac{k(k+d+1)n^2}{(n+d+1)^2}\left[n(n+2)(d^2 - 2d - 3 + nd)\|\beta_{\text{TT}}\|_2^6 + (n+d+1)\left((d+1)^2 + n(d+2)\right)\|\beta_{\text{TT}}\|_2^6\right].$$

Rearranging and approximating by omitting lower-order terms, we obtain:

$$4\eta k \frac{n^2 d^2}{(n+d+1)^2} \|\boldsymbol{\beta}_{\text{TT}}\|_2^4 - 4\eta^2 \frac{k(k+d+1)n^2 d(n+d)(n^2+d)}{(n+d+1)^2} \|\boldsymbol{\beta}_{\text{TT}}\|_2^6. \tag{16}$$

Taking the derivative of this expression and setting to 0 yields that the optimal $\eta^*$ solution is approximately:

$$\eta^* \approx \frac{d}{2(k+d+1)(n^2+d)(n+d)\|\boldsymbol{\beta}_{\text{TT}}\|_2^2}. \tag{17}$$

Plugging the optimal $\eta^*$ value in (17) into Equation (16), we obtain that the loss improvement is approximately:

$$\Delta\mathcal{L} \approx \frac{2kd^3}{(n+d+1)^2(k+d+1)(n+d)(1+\frac{d}{n^2})}\|\boldsymbol{\beta}_{\text{TT}}\|_2^2 - \frac{kd^3}{(n+d+1)^2(k+d+1)(n+d)(1+\frac{d}{n^2})}\|\boldsymbol{\beta}_{\text{TT}}\|_2^2$$

$$= \frac{k}{k+d+1} \frac{d^3}{(n+d+1)^2(n+d)\left(1+\frac{d}{n^2}\right)}\|\boldsymbol{\beta}_{\text{TT}}\|_2^2.$$

Considering that $n/d = \Theta(1)$, we conclude that

$$\Delta\mathcal{L} \approx \frac{k}{k+d} \frac{d^3}{(n+d)^3} \|\boldsymbol{\beta}_{\text{TT}}\|_2^2.$$

$\square$

**Corollary 4.4.** *Recall the definitions of $\gamma = k/d$, $\alpha = n/d$, and consider the loss $\mathcal{L}_{TT}(\boldsymbol{W}_{TT})$ as a function of $\alpha$ in the isotropic covariance and noiseless setting. If $\gamma > \frac{1}{2}$, then the loss $\mathcal{L}_{TT}(\boldsymbol{W}_{TT})$ is non-monotonic in $\alpha$. Specifically, the loss $\mathcal{L}_{TT}(\boldsymbol{W}_{TT})$ is increasing for $\alpha < \sqrt{\frac{3\gamma}{\gamma+1}} - 1$ and decreasing for $\alpha > \sqrt{\frac{3\gamma}{\gamma+1}} - 1$. Conversely, if $\gamma < \frac{1}{2}$, then $\mathcal{L}_{TT}(\boldsymbol{W}_{TT})$ is monotonic in $\alpha$.*

*Proof.* Using Theorem 4.2, the new loss $\mathcal{L}_{TT}(\boldsymbol{W}_{\text{TT}})$ is approximately:

$$\mathcal{L}_{TT}(\boldsymbol{W}_{\text{TT}}) \approx \|\boldsymbol{\beta}_{\text{TT}}\|_2^2 \left(\frac{d}{n+d} - \frac{k}{k+d}\left(\frac{d}{n+d}\right)^3\right) = \|\boldsymbol{\beta}_{\text{TT}}\|_2^2 \left(a - \frac{k}{k+d}a^3\right) \text{ where } a := \frac{d}{n+d} \in (0,1).$$

Since $\|\boldsymbol{\beta}_{\text{TT}}\|_2$ is fixed, we focus on the second multiplier and define $f(a) := a - \frac{k}{k+d}a^3$. Computing the derivative of $f(a)$ gives:

$$f'(a) = 1 - 3\frac{k}{k+d}a^2.$$

Solving $f'(a) = 0$ yields $a_{\text{crit}} = \sqrt{\frac{k+d}{3k}}$. This lies in $(0,1)$ when $\sqrt{\frac{k+d}{3k}} < 1$, i.e. $k > \frac{d}{2}$. In that case, $f(a)$ has exactly one turning point in $(0,1)$, so $f$ increases on $(0, a_{\text{crit}})$ and decreases on $(a_{\text{crit}}, 1)$, making it non-monotonic. On the other hand, in the $k < \frac{d}{2}$ regime, one finds

$$\sqrt{\frac{k+d}{3k}} > 1 \implies a_{\text{crit}} \notin (0,1),$$

hence $f'(a)$ cannot change sign over $(0,1)$. Therefore, $f(a)$ is monotonic in $a = \frac{d}{n+d}$, which implies that $\mathcal{L}_{TT}(\boldsymbol{W}_{\text{TT}})$ is also monotonic in the ratio $n/d$. Finally, since $a = \frac{1}{\alpha+1}$, the threshold until which $\mathcal{L}_{TT}(\boldsymbol{W}_{\text{TT}})$ increases translates to $\sqrt{\frac{3\gamma}{\gamma+1}} - 1$ in terms of $\alpha$. Consequently, for $\gamma > 1/2$, the new loss is non-monotonic with respect to $\alpha = n/d$, whereas for $\gamma < 1/2$, it remains monotonic. This completes the argument. $\square$

**Theorem 4.5.** *Consider the isotropic covariance and noiseless setting ($\sigma^2 = 0$). Suppose the initial weight matrix is $\boldsymbol{W}^* = \boldsymbol{0}_{d\times d}$. Then, the optimal step-size that minimizes $\mathcal{L}_{TT}(\boldsymbol{W}_{TT})$ is*

$$\eta^* = \frac{1}{2(k+d+1)(n^2+4n+3+d)\|\boldsymbol{\beta}_{TT}\|_2^2}.$$

*With this optimal step-size $\eta^*$, the improvement in the loss due to test-time training is*

$$\mathcal{L}(W^*) - \mathcal{L}_{TT}(W_{TT}) = \frac{k}{k+d+1} \frac{n^2}{n^2 + 4n + 3 + d} \|\boldsymbol{\beta}_{TT}\|_2^2,$$

*where $\mathcal{L}(W^*) = \|\boldsymbol{\beta}_{TT}\|_2^2$.*

*Proof.* In the proof, we will handle the general noisy scenario, and setting $\sigma^2 = 0$ in the final expression will recover the result for the noiseless setting. Similar to the proof of Theorem 4.2, we aim to calculate the expected gain in the loss with respect to $(n + k)$ samples used during the test-time training process. During test time, we draw $(n + k)$ total samples $\{(\bar{\boldsymbol{x}}_i, \bar{y}_i)\}_{i=1}^{n+k}$ drawn at test time, each with an associated noise variable $\{\xi_i\}_{i=1}^{n+k}$. Let's denote $\boldsymbol{\xi}_{\text{train}} = \begin{bmatrix} \xi_1 & \dots & \xi_n \end{bmatrix}^\top$ and $\boldsymbol{\xi}_{\text{context}} = \begin{bmatrix} \xi_{n+1} & \dots & \xi_{n+k} \end{bmatrix}^\top$. We will then evaluate the expectation of the loss function given by Lemma B.2 and compute $\mathcal{L}(W^*) - \mathcal{L}_{TT}(W_{TT}) = \mathbb{E}_{X_{\text{train}}, \boldsymbol{\xi}_{\text{train}}, X_{\text{context}}, \boldsymbol{\xi}_{\text{context}}} [\Delta \mathcal{L}]$ where $\Delta \mathcal{L}$ is given by:

$$
\begin{aligned}
\Delta \mathcal{L} &= \mathcal{L}(W^*) - \mathcal{L}(W_{TT}) \\
&= \boldsymbol{\beta}_{TT}^\top \Big[ \boldsymbol{\Sigma}_x - n\boldsymbol{\Sigma}_x W^* \boldsymbol{\Sigma}_x - n\boldsymbol{\Sigma}_x W^{*\top} \boldsymbol{\Sigma}_x + n(n+1)\boldsymbol{\Sigma}_x W^{*\top} \boldsymbol{\Sigma}_x W^* \boldsymbol{\Sigma}_x + n\text{tr}(W^{*\top} \boldsymbol{\Sigma}_x W^* \boldsymbol{\Sigma}_x) \boldsymbol{\Sigma}_x \Big] \boldsymbol{\beta}_{TT} \\
&\quad + \sigma^2 n\text{tr}(W^{*\top} \boldsymbol{\Sigma}_x W^* \boldsymbol{\Sigma}_x) \\
&\quad - \boldsymbol{\beta}_{TT}^\top \Big[ \boldsymbol{\Sigma}_x - n\boldsymbol{\Sigma}_x W_{TT} \boldsymbol{\Sigma}_x - n\boldsymbol{\Sigma}_x W_{TT}^\top \boldsymbol{\Sigma}_x + n(n+1)\boldsymbol{\Sigma}_x W_{TT}^\top \boldsymbol{\Sigma}_x W_{TT} \boldsymbol{\Sigma}_x + n\text{tr}(W_{TT}^\top \boldsymbol{\Sigma}_x W_{TT} \boldsymbol{\Sigma}_x) \boldsymbol{\Sigma}_x \Big] \boldsymbol{\beta}_{TT} \\
&\quad - \sigma^2 n\text{tr}(W_{TT}^\top \boldsymbol{\Sigma}_x W_{TT} \boldsymbol{\Sigma}_x) \\
&= \boldsymbol{\beta}_{TT}^\top \Big[ n\boldsymbol{\Sigma}_x (W_{TT} - W^*) \boldsymbol{\Sigma}_x + n\boldsymbol{\Sigma}_x (W_{TT}^\top - W^{*\top}) \boldsymbol{\Sigma}_x + n(n+1)\boldsymbol{\Sigma}_x (W^{*\top} \boldsymbol{\Sigma}_x W^* - W_{TT}^\top \boldsymbol{\Sigma}_x W_{TT}) \boldsymbol{\Sigma}_x \\
&\quad + n \big( \text{tr}(W^{*\top} \boldsymbol{\Sigma}_x W^* \boldsymbol{\Sigma}_x) - \text{tr}(W_{TT}^\top \boldsymbol{\Sigma}_x W_{TT} \boldsymbol{\Sigma}_x) \big) \boldsymbol{\Sigma}_x \Big] \boldsymbol{\beta}_{TT} + \sigma^2 n \big( \text{tr}(W^{*\top} \boldsymbol{\Sigma}_x W^* \boldsymbol{\Sigma}_x) - \text{tr}(W_{TT} \boldsymbol{\Sigma}_x W_{TT} \boldsymbol{\Sigma}_x) \big).
\end{aligned}
\tag{18}
$$

Recall that when the linear task is $\boldsymbol{\beta}_{TT}$ and $W^* = \boldsymbol{0}_{d \times d}$, the gradient update given by Proposition 3.1 becomes:

$$
\begin{aligned}
W_{TT} &= W^* + 2\eta \Big[ X_{\text{train}}^\top X_{\text{train}} \big( \boldsymbol{\beta}_{TT} - W^* X_{\text{context}}^\top \boldsymbol{y}_{\text{context}} \big) + X_{\text{train}}^\top \boldsymbol{\xi}_{\text{train}} \Big] (X_{\text{context}}^\top \boldsymbol{y}_{\text{context}})^\top \\
&= W^* + 2\eta \Big[ X_{\text{train}}^\top X_{\text{train}} \big( \boldsymbol{\beta}_{TT} - W^* X_{\text{context}}^\top (X_{\text{context}} \boldsymbol{\beta}_{TT} + \boldsymbol{\xi}_{\text{context}}) \big) + X_{\text{train}}^\top \boldsymbol{\xi}_{\text{train}} \Big] (X_{\text{context}}^\top (X_{\text{context}} \boldsymbol{\beta}_{TT} + \boldsymbol{\xi}_{\text{context}}))^\top \\
&= W^* + 2\eta \Big[ X_{\text{train}}^\top X_{\text{train}} \boldsymbol{\beta}_{TT} + X_{\text{train}}^\top \boldsymbol{\xi}_{\text{train}} \Big] (X_{\text{context}}^\top (X_{\text{context}} \boldsymbol{\beta}_{TT} + \boldsymbol{\xi}_{\text{context}}))^\top \\
&= 2\eta \Big[ X_{\text{train}}^\top X_{\text{train}} \boldsymbol{\beta}_{TT} + X_{\text{train}}^\top \boldsymbol{\xi}_{\text{train}} \Big] \Big[ \boldsymbol{\beta}_{TT}^\top X_{\text{context}}^\top X_{\text{context}} + \boldsymbol{\xi}_{\text{context}}^\top X_{\text{context}} \Big].
\end{aligned}
$$

Then, we obtain its transpose as

$$W_{TT}^\top = 2\eta \Big[ X_{\text{context}}^\top X_{\text{context}} \boldsymbol{\beta}_{TT} + X_{\text{context}}^\top \boldsymbol{\xi}_{\text{context}} \Big] \Big[ \boldsymbol{\beta}_{TT}^\top X_{\text{train}}^\top X_{\text{train}} + \boldsymbol{\xi}_{\text{train}}^\top X_{\text{train}} \Big].$$

Thus, the expression $W_{TT}^\top W_{TT}$ becomes:

$$
\begin{aligned}
W_{TT}^\top W_{TT} = 4\eta^2 &\Big[ X_{\text{context}}^\top X_{\text{context}} \boldsymbol{\beta}_{TT} + X_{\text{context}}^\top \boldsymbol{\xi}_{\text{context}} \Big] \Big[ \boldsymbol{\beta}_{TT}^\top X_{\text{train}}^\top X_{\text{train}} + \boldsymbol{\xi}_{\text{train}}^\top X_{\text{train}} \Big] \\
&\Big[ X_{\text{train}}^\top X_{\text{train}} \boldsymbol{\beta}_{TT} + X_{\text{train}}^\top \boldsymbol{\xi}_{\text{train}} \Big] \Big[ \boldsymbol{\beta}_{TT}^\top X_{\text{context}}^\top X_{\text{context}} + \boldsymbol{\xi}_{\text{context}}^\top X_{\text{context}} \Big].
\end{aligned}
$$

Therefore, we can write:

$$
\begin{aligned}
\mathbb{E}_{X_{\text{train}}, \boldsymbol{\xi}_{\text{train}}, X_{\text{context}}, \boldsymbol{\xi}_{\text{context}}} \Big[ W_{TT}^\top W_{TT} \Big] = 4\eta^2 \, \mathbb{E}_{X_{\text{train}}, \boldsymbol{\xi}_{\text{train}}, X_{\text{context}}, \boldsymbol{\xi}_{\text{context}}} \Big[ &X_{\text{context}}^\top \boldsymbol{\xi}_{\text{context}} \boldsymbol{\xi}_{\text{train}}^\top X_{\text{train}} X_{\text{train}}^\top \boldsymbol{\xi}_{\text{train}} \boldsymbol{\xi}_{\text{context}}^\top X_{\text{context}} \\
&+ X_{\text{context}}^\top \boldsymbol{\xi}_{\text{context}} \boldsymbol{\beta}_{TT}^\top X_{\text{train}}^\top X_{\text{train}} X_{\text{train}}^\top X_{\text{train}} \boldsymbol{\beta}_{TT} \boldsymbol{\xi}_{\text{context}}^\top X_{\text{context}} \\
&+ X_{\text{context}}^\top X_{\text{context}} \boldsymbol{\beta}_{TT} \boldsymbol{\xi}_{\text{train}}^\top X_{\text{train}} X_{\text{train}}^\top \boldsymbol{\xi}_{\text{train}} \boldsymbol{\beta}_{TT}^\top X_{\text{context}}^\top X_{\text{context}} \\
&+ X_{\text{context}}^\top X_{\text{context}} \boldsymbol{\beta}_{TT} \boldsymbol{\beta}_{TT}^\top X_{\text{train}}^\top X_{\text{train}} X_{\text{train}}^\top X_{\text{train}} \boldsymbol{\beta}_{TT} \boldsymbol{\beta}_{TT}^\top X_{\text{context}}^\top X_{\text{context}} \Big].
\end{aligned}
$$

By plugging $M = I$ in Lemma B.3 and considering that $\boldsymbol{\Sigma}_x = I$, we have the following fact:

$$\mathbb{E}_{X_{\text{train}}} \Big[ X_{\text{train}}^\top X_{\text{train}} X_{\text{train}}^\top X_{\text{train}} \Big] = k(k + d + 1)I.$$

Applying this identity twice and using the fact that $\mathbb{E}_{X_{\text{train}}}\left[X_{\text{train}}X_{\text{train}}^\top\right] = \text{tr}(\Sigma_x)I = dI$, we obtain:

$$
\begin{aligned}
&\mathbb{E}_{X_{\text{train}},\xi_{\text{train}},X_{\text{context}},\xi_{\text{context}}}\left[W_{\text{TT}}^\top W_{\text{TT}}\right]\\
&= 4\eta^2\left(\sigma^4 knd I + \sigma^2 k(k+d+1)n\|\beta_{\text{TT}}\|_2^2 I + \sigma^2 kd\left(n(n+1)\beta_{\text{TT}}\beta_{\text{TT}}^\top + n\|\beta_{\text{TT}}\|_2^2 I\right)\right.\\
&\quad\left. + k(k+d+1)\|\beta_{\text{TT}}\|_2^2\left(n(n+1)\beta_{\text{TT}}\beta_{\text{TT}}^\top + n\|\beta_{\text{TT}}\|_2^2 I\right)\right)\\
&= 4\eta^2\left(\sigma^2 kn\left(\sigma^2 d + (k+d+1)\|\beta_{\text{TT}}\|_2^2\right)I + k\left(n(n+1)\beta_{\text{TT}}\beta_{\text{TT}}^\top + n\|\beta_{\text{TT}}\|_2^2 I\right)\left(\sigma^2 d + (k+d+1)\|\beta_{\text{TT}}\|_2^2\right)\right)\\
&= 4\eta^2 kn\left(\sigma^2 d + (k+d+1)\|\beta_{\text{TT}}\|_2^2\right)\left(\sigma^2 I + (n+1)\beta_{\text{TT}}\beta_{\text{TT}}^\top + \|\beta_{\text{TT}}\|_2^2 I\right).
\end{aligned}
\tag{19}
$$

Besides that, we calculate the first order expectations of $W_{\text{TT}}$ as:

$$
\begin{aligned}
\mathbb{E}_{X_{\text{train}},\xi_{\text{train}},X_{\text{context}},\xi_{\text{context}}}\left[\beta_{\text{TT}}^\top n\Sigma_x\left(W_{\text{TT}} - W^*\right)\Sigma_x\beta_{\text{TT}}\right] &= 2\eta n\beta_{\text{TT}}^\top \mathbb{E}_{X_{\text{train}},X_{\text{context}}}\left[X_{\text{train}}^\top X_{\text{train}}\beta_{\text{TT}}\beta_{\text{TT}}^\top X_{\text{context}}^\top X_{\text{context}}\right]\beta_{\text{TT}}\\
&= 2\eta kn^2\|\beta_{\text{TT}}\|_2^4,
\end{aligned}
\tag{20}
$$

and similarly,

$$
\mathbb{E}_{X_{\text{train}},\xi_{\text{train}},X_{\text{context}},\xi_{\text{context}}}\left[\beta_{\text{TT}}^\top n\Sigma_x\left(W_{\text{TT}}^\top - W^{*\top}\right)\Sigma_x\beta_{\text{TT}}\right] = 2\eta kn^2\|\beta_{\text{TT}}\|_2^4.
\tag{21}
$$

The next term in the loss difference is:

$$
\begin{aligned}
&\mathbb{E}_{X_{\text{train}},\xi_{\text{train}},X_{\text{context}},\xi_{\text{context}}}\left[\beta_{\text{TT}}^\top n(n+1)\Sigma_x\left(W^{*\top}\Sigma_x W^* - W_{\text{TT}}^\top\Sigma_x W_{\text{TT}}\right)\Sigma_x\beta_{\text{TT}}\right]\\
&= -4\eta^2 kn^2(n+1)\left((n+2)\|\beta_{\text{TT}}\|_2^2 + \sigma^2\right)\|\beta_{\text{TT}}\|_2^2\left(\sigma^2 d + (k+d+1)\|\beta_{\text{TT}}\|_2^2\right).
\end{aligned}
\tag{22}
$$

Using (19), the last term is:

$$
\begin{aligned}
&\mathbb{E}_{X_{\text{train}},\xi_{\text{train}},X_{\text{context}},\xi_{\text{context}}}\left[\beta_{\text{TT}}^\top n\left(\text{tr}(W^{*\top}\Sigma_x W^*\Sigma_x) - \text{tr}(W_{\text{TT}}^\top\Sigma_x W_{\text{TT}}\Sigma_x)\right)\Sigma_x\beta_{\text{TT}}\right]\\
&= -4\eta^2 n\beta_{\text{TT}}^\top\text{tr}(\mathbb{E}_{X_{\text{train}},\xi_{\text{train}},X_{\text{context}},\xi_{\text{context}}}\left[W_{\text{TT}}^\top W_{\text{TT}}\right])\beta_{\text{TT}}.\\
&= -4\eta^2 kn^2\left(\sigma^2 d + (k+d+1)\|\beta_{\text{TT}}\|_2^2\right)\left(\sigma^2 d + (n+d+1)\|\beta_{\text{TT}}\|_2^2\right)\|\beta_{\text{TT}}\|_2^2.
\end{aligned}
\tag{23}
$$

Finally, the noise term is equal to:

$$
\begin{aligned}
&\mathbb{E}_{X_{\text{train}},\xi_{\text{train}},X_{\text{context}},\xi_{\text{context}}}\left[\sigma^2 n\left(\text{tr}(W^{*\top}\Sigma_x W^*\Sigma_x) - \text{tr}(W_{\text{TT}}\Sigma_x W_{\text{TT}}\Sigma_x)\right)\right]\\
&= -4\eta^2 kn^2\sigma^2\left(\sigma^2 d + (k+d+1)\|\beta_{\text{TT}}\|_2^2\right)\left(\sigma^2 d + (n+d+1)\|\beta_{\text{TT}}\|_2^2\right).
\end{aligned}
\tag{24}
$$

Combining the Equations (20), (21), (22), (23), (24) in the expectation of the loss difference (18), we obtain the following quadratic expression of $\eta$:

$$
\begin{aligned}
&\mathcal{L}(W^*) - \mathcal{L}_{TT}(W_{\text{TT}})\\
&= 4\eta kn^2\|\beta_{\text{TT}}\|_2^4 - 4\eta^2 kn^2(n+1)\left((n+2)\|\beta_{\text{TT}}\|_2^2 + \sigma^2\right)\|\beta_{\text{TT}}\|_2^2\left(\sigma^2 d + (k+d+1)\|\beta_{\text{TT}}\|_2^2\right)\\
&\quad - 4\eta^2 kn^2\left(\sigma^2 d + (k+d+1)\|\beta_{\text{TT}}\|_2^2\right)\left(\sigma^2 d + (n+d+1)\|\beta_{\text{TT}}\|_2^2\right)\|\beta_{\text{TT}}\|_2^2\\
&\quad - 4\eta^2 kn^2\sigma^2\left(\sigma^2 d + (k+d+1)\|\beta_{\text{TT}}\|_2^2\right)\left(\sigma^2 d + (n+d+1)\|\beta_{\text{TT}}\|_2^2\right)\\
&= 4kn^2\left[\eta\|\beta_{\text{TT}}\|_2^4 - \eta^2\left(\sigma^2 d + (k+d+1)\|\beta_{\text{TT}}\|_2^2\right)\left[\left(\sigma^2 + \|\beta_{\text{TT}}\|_2^2\right)\left(\sigma^2 d + (n+d+1)\|\beta_{\text{TT}}\|_2^2\right)\right.\right.\\
&\quad\left.\left. + (n+1)\left((n+2)\|\beta_{\text{TT}}\|_2^2 + \sigma^2\right)\|\beta_{\text{TT}}\|_2^2\right]\right]\\
&= 4kn^2\left[\eta\|\beta_{\text{TT}}\|_2^4 - \eta^2\left(\sigma^2 d + (k+d+1)\|\beta_{\text{TT}}\|_2^2\right)\left(\sigma^4 d + \|\beta_{\text{TT}}\|_2^2\left((n^2+4n+3+d)\|\beta_{\text{TT}}\|_2^2 + 2\sigma^2(n+d+1)\right)\right)\right].
\end{aligned}
$$

Setting the derivative to 0 and solving for $\eta$ gives:

$$
\eta^\star = \frac{\|\beta_{\text{TT}}\|_2^4}{2\left(\sigma^2 d + (k+d+1)\|\beta_{\text{TT}}\|_2^2\right)\left(\sigma^4 d + \|\beta_{\text{TT}}\|_2^4(n^2+4n+3+d) + 2\sigma^2(n+d+1)\|\beta_{\text{TT}}\|_2^2\right)}.
$$

At this optimal value $\eta^\star$, the improvement on the loss is:

$$\frac{k\,\|\boldsymbol{\beta}_{\text{TT}}\|_2^2}{\sigma^2 d + (k + d + 1)\,\|\boldsymbol{\beta}_{\text{TT}}\|_2^2}\frac{n^2\,\|\boldsymbol{\beta}_{\text{TT}}\|_2^4}{\left(\sigma^4 d + \|\boldsymbol{\beta}_{\text{TT}}\|_2^4\,(n^2 + 4n + 3 + d) + 2\sigma^2(n + d + 1)\,\|\boldsymbol{\beta}_{\text{TT}}\|_2^2\right)}\|\boldsymbol{\beta}_{\text{TT}}\|_2^2.$$

In the noiseless setting, the optimal $\eta^\star$ corresponds to:

$$\eta^* = \frac{1}{2(k + d + 1)(n^2 + 4n + 3 + d)\|\boldsymbol{\beta}_{\text{TT}}\|_2^2}.$$

At that optimal value $\eta^\star$, the improvement is:

$$\frac{k}{k + d + 1}\frac{n^2}{n^2 + 4n + 3 + d}\|\boldsymbol{\beta}_{\text{TT}}\|_2^2.$$

This completes the proof. As a final note, recall that the initial loss is $\|\boldsymbol{\beta}_{\text{TT}}\|_2^2$. Thus, the new loss of the system after the test-time-training update is:

$$\left(1 - \frac{k}{k + d + 1}\frac{n^2}{n^2 + 4n + 3 + d}\right)\|\boldsymbol{\beta}_{\text{TT}}\|_2^2.$$

In the proportional $n, d$ regime and when $k/d \to \infty$, this gives an error of order $O(n^{-1})$-matching that of the optimal $W^{\text{opt}}$ from Lemma C.1:

$$\frac{4n + d + 3}{n^2 + 4n + 3 + d}\,\|\boldsymbol{\beta}_{\text{TT}}\|_2^2.$$

$\square$

**Corollary 4.6.** *Recall the definitions of $\alpha = n/d$ and $\gamma = k/d$. Consider the setting in Theorem 4.2 and test-time training described in Proposition 3.1 with both pre-trained and zero initializations. Then, under the isotropic covariance and noiseless setting, there exists a threshold $\gamma^\star$ given by $\gamma^\star \approx \dfrac{(\alpha + 1)^2}{(\alpha + 2)}$ such that $\gamma < \gamma^\star$ if and only if it is better to utilize the pre-trained initialization over the zero initialization $\mathbf{0}_{d\times d}$.*

*Proof.* Considering that the initial loss of the weight matrix $W = \mathbf{0}_{d\times d}$ is $\|\boldsymbol{\beta}_{\text{TT}}\|_2^2$ and the improvement given by Theorem 4.5, the new loss after the test-time-training update is approximately:

$$\left(1 - \frac{k}{k + d}\right)\|\boldsymbol{\beta}_{\text{TT}}\|_2^2.$$

On the other hand, recall that by Proposition 4.1, the loss of $W^*$ before the update is $\frac{d+1}{n+d+1}\|\boldsymbol{\beta}_{\text{TT}}\|_2^2$. Combining this with the improvement given by Theorem 4.2, the new loss after the test-time-training update is approximately:

$$\left(\frac{d}{n + d} - \frac{k}{k + d}\frac{d^3}{(n + d)^3}\right)\|\boldsymbol{\beta}_{\text{TT}}\|_2^2.$$

Let us define $\beta := \frac{k}{k+d}$ and $\theta := \frac{d}{n+d}$. Then, we check when it's better (yields smaller loss) to use the pre-trained matrix $W^*$ over zero initialization with the below inequality:

$$1 - \beta > \theta - \beta\theta^3 \iff 1 - \theta > \beta\left(1 - \theta^3\right)$$

$$\iff \frac{1}{\beta} > \theta^2 + \theta + 1$$

$$\iff \frac{d}{k} > \theta^2 + \theta$$

$$\iff \frac{d}{\frac{d}{n+d}\left(1 + \frac{d}{n+d}\right)} > k$$

$$\iff \frac{(n + d)^2}{n + 2d} > k \iff \frac{(\alpha + 1)^2}{\alpha + 2} > \gamma = k/d \text{ where } \alpha = \frac{n}{d}.$$

Thus, we conclude our argument. $\square$

## C. Proofs for Section 5

**Lemma C.1** (Optimal $W$)**.** *When the system is noiseless ($\sigma^2 = 0$), the optimal $W$ which minimizes the population risk with respect to the new task $\boldsymbol{\beta}_{TT}$ given by Lemma B.2 and its corresponding loss are:*

$$W^{opt} = \frac{\boldsymbol{\beta}_{TT}\boldsymbol{\beta}_{TT}^\top}{(n+2)\,\|\boldsymbol{\beta}_{TT}\|_2^2} \qquad \mathcal{L}(W^{opt}) = \frac{2}{n+2}\,\|\boldsymbol{\beta}_{TT}\|_2^2.$$

*Proof.* Recall that when $\boldsymbol{\Sigma}_x = I$ and $\sigma^2 = 0$, the loss formula given by Lemma B.2 becomes:

$$\mathcal{L}(W) = \boldsymbol{\beta}_{\mathrm{TT}}^\top \Big[ I \; - \; nW \; - \; nW^\top \; + \; n(n+1)W^\top W + n\mathrm{tr}(W^\top W)\Big] \boldsymbol{\beta}_{\mathrm{TT}}. \tag{25}$$

The gradient of this expression with respect to $W$ is:

$$\nabla_W \mathcal{L}(W) = -\,2n\,\boldsymbol{\beta}_{\mathrm{TT}}\boldsymbol{\beta}_{\mathrm{TT}}^\top + 2n(n+1)W\boldsymbol{\beta}_{\mathrm{TT}}\boldsymbol{\beta}_{\mathrm{TT}}^\top \; + \; 2n\,\boldsymbol{\beta}_{\mathrm{TT}}^\top\boldsymbol{\beta}_{\mathrm{TT}}W \; = \; 0.$$

Simplifying, this gives:

$$(n+1)W\boldsymbol{\beta}_{\mathrm{TT}}\boldsymbol{\beta}_{\mathrm{TT}}^\top + \boldsymbol{\beta}_{\mathrm{TT}}^\top\boldsymbol{\beta}_{\mathrm{TT}}W = \boldsymbol{\beta}_{\mathrm{TT}}\boldsymbol{\beta}_{\mathrm{TT}}^\top. \tag{26}$$

Multiply (26) on the right by $\boldsymbol{\beta}_{\mathrm{TT}}$. This yields:

$$W\boldsymbol{\beta}_{\mathrm{TT}} = \frac{1}{n+2}\,\boldsymbol{\beta}_{\mathrm{TT}}.$$

So $\boldsymbol{\beta}_{\mathrm{TT}}$ is an eigenvector of $W$ corresponding to the eigenvalue $1/(n+2)$. Next, consider any vector $v$ that is orthonormal (or simply orthogonal) to $v$, that is $v^\top \boldsymbol{\beta}_{\mathrm{TT}} = 0$. Multiplying (26) on the right by $v$:

$$\|\boldsymbol{\beta}_{\mathrm{TT}}\|^2\, W v = 0 \quad \implies \quad W v = 0 \quad \text{whenever} \quad v^\top \boldsymbol{\beta}_{\mathrm{TT}} = 0.$$

Hence, all other eigenvalues of $W$ are 0. Using the eigendecomposition of $W$, this uniquely specifies $W$ as $c \cdot \boldsymbol{\beta}_{\mathrm{TT}}\boldsymbol{\beta}_{\mathrm{TT}}^\top$. Solving for $c$ yields the unique solution to $\nabla_W \mathcal{L}(W) = 0$:

$$W^{\mathrm{opt}} = \frac{\boldsymbol{\beta}_{\mathrm{TT}}\boldsymbol{\beta}_{\mathrm{TT}}^\top}{(n+2)\,\|\boldsymbol{\beta}_{\mathrm{TT}}\|_2^2}$$

Finally, plugging this $W^{\mathrm{opt}}$ to (25) yields the following error:

$$\mathcal{L}(W^{\mathrm{opt}}) = \frac{2}{n+2}\,\|\boldsymbol{\beta}_{\mathrm{TT}}\|_2^2.$$

$\square$

**Lemma C.2** (Eigenvalues)**.** *Consider the optimal pre-trained $W^*$ matrix, which minimizes the population loss over all possible tasks sampled from $\Sigma_\beta$ in (1). Then, all eigenvalues of $\bar{W}^* = \boldsymbol{\Sigma}_x^{1/2}W^*\boldsymbol{\Sigma}_x^{1/2}$ lie in the interval $[0, \frac{1}{n+1}]$.*

*Proof.* By Theorem 1 of Li et al. (2024), we write the following:

$$W^* = \boldsymbol{\Sigma}_x^{-1/2}\bar{W}^*\boldsymbol{\Sigma}_x^{-1/2} \text{ where } \bar{W}^* = \Big((n+1)I_d + MA^{-1}\Big)^{-1} \text{ where } M = \mathrm{tr}(A) + \sigma^2 \text{ and } A = \boldsymbol{\Sigma}_x^{1/2}\boldsymbol{\Sigma}_\beta\boldsymbol{\Sigma}_x^{1/2}$$

We know that the matrices $\boldsymbol{\Sigma}_x$ and $\boldsymbol{\Sigma}_\beta$ are jointly diagonalizable with $\boldsymbol{\Sigma}_x = Q\boldsymbol{\Lambda}_x Q^\top$ and $\boldsymbol{\Sigma}_\beta = Q\boldsymbol{\Lambda}_\beta Q^\top$ where $Q$ is an orthonormal matrix $Q \in \mathbb{R}^{d \times d}$ and $\boldsymbol{\Lambda}_x, \boldsymbol{\Lambda}_\beta \in \mathbb{R}^{d \times d}$ are diagonal matrices with entries $\{\lambda_i\}$ and $\{\beta_i\}$, respectively. Then,

$$A = Q\left(\boldsymbol{\Lambda}_x^{1/2}\,\boldsymbol{\Lambda}_\beta\,\boldsymbol{\Lambda}_x^{1/2}\right)Q^\top.$$

Define $\boldsymbol{\Lambda}_A := \boldsymbol{\Lambda}_x^{1/2}\,\boldsymbol{\Lambda}_\beta\,\boldsymbol{\Lambda}_x^{1/2}$, which is also diagonal. Hence

$$A^{-1} = Q\,\boldsymbol{\Lambda}_A^{-1}\,Q^\top,$$

where $\mathbf{\Lambda}_A^{-1} = \text{diag}\left(1/(\lambda_i\beta_i)\right)$. Inside $\bar{W}^*$, we have

$$(n+1)\,I + M\,A^{-1} = Q\,(n+1)\,I\,Q^\top + M\left(Q\,\mathbf{\Lambda}_A^{-1}\,Q^\top\right) = Q\left[(n+1)\,I + M\,\mathbf{\Lambda}_A^{-1}\right]Q^\top.$$

We know that $\mathbf{\Lambda}_{\text{diag}} := (n+1)\,I + M\,\mathbf{\Lambda}_A^{-1}$ is also diagonal with diagonal entries $(n+1) + \frac{M}{\lambda_i\beta_i}$. It follows that

$$\bar{W}^* = \left((n+1)\,I + M\,A^{-1}\right)^{-1} = Q\,\text{diag}\!\left(\frac{\lambda_i\beta_i}{(n+1)\lambda_i\beta_i + M}\right)Q^\top,$$

where $M = \text{tr}(A) + \sigma^2 = \text{tr}(\mathbf{\Lambda}_\beta\mathbf{\Lambda}_x) = \sigma^2 + \sum_{i=1}^d \lambda_i\beta_i$. This concludes the proof. $\qquad\square$

**Lemma C.3** (Covariance Shift). *Consider the setting in Section 3 where*

- *The true labels are generated by $y = x^\top\beta + \xi$ with $x \sim \mathcal{N}(0, \Sigma_x)$, $\beta \sim \mathcal{N}(0, \Sigma_\beta)$, and noise $\xi \sim \mathcal{N}(0, \sigma^2)$.*

- *The prediction of a linear attention model is of the form*

$$\hat{y} = x^\top W X^\top y,$$

*where the context $X$ in $\mathbb{R}^{n\times d}$ collects previous samples $x_i^\top$ as rows, $y$ is the corresponding label vector, and $W \in \mathbb{R}^{d\times d}$ is the model parameter matrix.*

*Then, the invertible transformation*

$$(x, X, \beta, W) \mapsto (\bar{x}, \bar{X}, \bar{\beta}, \bar{W}) \text{ where } \bar{x} = \Sigma_x^{-1/2}x, \quad \bar{X} = X\Sigma_x^{-1/2}, \quad \bar{\beta} = \Sigma_x^{1/2}\beta, \quad \bar{W} = \Sigma_x^{1/2}W\Sigma_x^{1/2},$$

*preserves the population risks in (1) and (3). More precisely, defining $\bar{\mathcal{L}}(\bar{W})$ to be the population loss (3) evaluated under the transformed setting $(\bar{x}, \bar{X}, \bar{\beta}, \bar{W})$, we have*

$$\mathcal{L}(W) = \bar{\mathcal{L}}(\bar{W}).$$

*Proof.* Since $x \sim \mathcal{N}(0, \Sigma_x)$, multiplying by $\Sigma_x^{-1/2}$ yields $\bar{x} = \Sigma_x^{-1/2}x \sim \mathcal{N}(0, I_d)$. Likewise, because $\beta \sim \mathcal{N}(0, \Sigma_\beta)$, we have $\bar{\beta} = \Sigma_x^{1/2}\beta \sim \mathcal{N}\left(0, \Sigma_x^{1/2}\Sigma_\beta\Sigma_x^{1/2}\right)$. Next, we observe

$$\bar{X} := X\Sigma_x^{-1/2} \implies \bar{X}\bar{\beta} = \left(X\Sigma_x^{-1/2}\right)\left(\Sigma_x^{1/2}\beta\right) = X\beta,$$

ensuring that label vector corresponding to context data $y = X\beta + \xi = \bar{X}\bar{\beta} + \xi$ stays the same. Also, note that the label is preserved for the query token $y = x^\top\beta + \xi = \bar{x}^\top\bar{\beta} + \xi$, as well. Under the map $x \mapsto \bar{x} = \Sigma_x^{-1/2}x$, $X \mapsto \bar{X} = X\Sigma_x^{-1/2}$, $\beta \mapsto \bar{\beta} = \Sigma_x^{1/2}\beta$, $W \mapsto \bar{W} = \Sigma_x^{1/2}W\Sigma_x^{1/2}$, the prediction of the linear attention model is also preserved:

$$x^\top W X^\top y = (\Sigma_x^{-1/2}x)^\top\Sigma_x^{1/2}W\Sigma_x^{1/2}\left(\Sigma_x^{-1/2}X^\top y\right) = \bar{x}^\top\left(\Sigma_x^{1/2}W\Sigma_x^{1/2}\right)\left(\bar{X}^\top y\right) = \bar{x}^\top\bar{W}\bar{X}^\top y.$$

As a result, the errors in the labels remain numerically unchanged as

$$(y - x^\top W X^\top y)^2 = (y - \bar{x}^\top\bar{W}\bar{X}^\top y)^2.$$

Hence, this implies that the population losses described in (1), (3) are preserved.

$$\mathcal{L}(W) = \bar{\mathcal{L}}(\bar{W}).$$

In particular, if $W^*$ is the unique minimizer of the pre-training loss in (1) in the original coordinates, its counterpart in the transformed system is precisely

$$\bar{W}^* = \Sigma_x^{1/2}W^*\Sigma_x^{1/2}.$$

This completes the proof. $\qquad\square$

**Theorem C.4.** *Let $n/d = \Theta(1)$ and $\sigma^2 = 0$. Suppose the covariance matrices are jointly diagonalizable by an orthogonal matrix $Q \in \mathbb{R}^{d \times d}$, so that $\Sigma_x = Q \Lambda_x Q^\top$ and $\Sigma_\beta = Q \Lambda_\beta Q^\top$. Let $W$ be any matrix that's also jointly diagonalizable by $Q$ such that if $\Sigma_x^{1/2} W \Sigma_x^{1/2} = Q \Lambda Q^\top$, then all eigenvalues of $\Lambda$ lie in the interval $[0, \frac{1}{n+1}]$. Define $\tilde{\beta}_{TT} := Q^\top \Sigma_x^{1/2} \beta_{TT}$, $A := \tilde{\beta}_{TT}^\top (I - n\Lambda)^2 \tilde{\beta}_{TT}$, and $B := n \|\tilde{\beta}_{TT}\|_2^2 \operatorname{tr}(\Lambda^2)$. Then, the optimal step size that minimizes the population loss given in* (3) *after the test-time training update is*

$$\eta^* \approx \frac{A}{2(k+d)\, n^2 \,\|\tilde{\beta}_{TT}\|_2^2 \,(A+B)}.$$

*With this optimal step-size $\eta^*$, the improvement in the loss due to test-time training and the initial loss are approximately*

$$\mathcal{L}(W) - \mathcal{L}_{TT}(W_{TT}) \approx \frac{k}{k+d}\frac{A^2}{A+B}, \quad \mathcal{L}(W) \approx A + B.$$

*Proof.* We consider the general non-isotropic covariance scenario where $\Sigma_x$ and $\Sigma_\beta$ may be arbitrary covariance matrices. Our analysis will hold for any initial weight matrix $W$ in $\mathbb{R}^{d \times d}$ such that if $\Sigma_x^{1/2} W \Sigma_x^{1/2} = Q \Lambda Q^\top$, all eigenvalues of $\Lambda$ are in $\left[0, \frac{1}{n+1}\right]$. Now, we aim to calculate the expected gain in the loss with respect to $(n+k)$ samples used during the test-time training process. That is, we will take the expectation of the loss function given by Lemma B.2 and compute $\mathcal{L}(W) - \mathcal{L}_{TT}(W_{TT}) = \mathbb{E}_{X_{\text{train}}, X_{\text{context}}}[\Delta \mathcal{L}]$ where $\Delta \mathcal{L}$ is given by:

$$
\begin{aligned}
\Delta \mathcal{L} &= \mathcal{L}(W) - \mathcal{L}(W_{TT}) \\
&= \beta_{TT}^\top \Big[ n \Sigma_x (W_{TT} - W) \Sigma_x + n \Sigma_x \left(W_{TT}^\top - W^\top\right) \Sigma_x + n(n+1) \Sigma_x \left(W^\top \Sigma_x W - W_{TT}^\top \Sigma_x W_{TT}\right) \Sigma_x \\
&\quad + n \left(\operatorname{tr}(W^\top \Sigma_x W \Sigma_x) - \operatorname{tr}(W_{TT}^\top \Sigma_x W_{TT} \Sigma_x)\right) \Sigma_x \Big] \beta_{TT}.
\end{aligned}
\tag{27}
$$

Now, recall the test-time training update from Proposition 3.1 when the new task is $\beta_{TT}$:

$$W_{TT} - W = 2\eta\, X_{\text{train}}^\top X_{\text{train}} \left(I - W X_{\text{context}}^\top X_{\text{context}}\right) \beta_{TT} \beta_{TT}^\top X_{\text{context}}^\top X_{\text{context}},$$

$$W_{TT}^\top - W^\top = 2\eta X_{\text{context}}^\top X_{\text{context}} \beta_{TT} \beta_{TT}^\top \left(I - X_{\text{context}}^\top X_{\text{context}} W^\top\right) X_{\text{train}}^\top X_{\text{train}}.$$

As in previous theorems, we will use the Gaussian approximation following Lemma B.1 by approximating $X_{\text{context}}^\top X_{\text{context}} \beta_{TT} \approx n \Sigma_x \beta_{TT} + \sqrt{n} \Sigma_x^{1/2} g$ where $g \sim \mathcal{N}(0, \|\beta_{TT}^2\| I)$. This non-isotropic form can be obtained by factoring out $\Sigma_x^{1/2}$ and reducing the problem to the isotropic case from Lemma B.1. Before going forward, let's apply the covariance shift discussed in Lemma C.3, which transforms:

$$\bar{x} = \Sigma_x^{-1/2} x \sim \mathcal{N}(0, I_d), \quad \bar{\beta}_{TT} = \Sigma_x^{1/2} \beta_{TT} \sim \mathcal{N}(0, \Sigma_x^{1/2} \Sigma_\beta \Sigma_x^{1/2}), \text{ and } \bar{W} = \Sigma_x^{1/2} W \Sigma_x^{1/2}.$$

Likewise, we define the transformed versions of training and context data as $\bar{X}_{\text{train}} := X_{\text{train}} \Sigma_x^{-1/2}$ and $\bar{X}_{\text{context}} := X_{\text{context}} \Sigma_x^{-1/2}$. Now, suppose that $\bar{W} = Q \Lambda Q^\top$ and let $\Delta W := \bar{W}_{TT} - \bar{W} = \Sigma_x^{1/2} W_{TT} \Sigma_x^{1/2} - \Sigma_x^{1/2} W \Sigma_x^{1/2}$. Then,

$$
\begin{aligned}
\Delta W &= 2\eta \bar{X}_{\text{train}}^\top \bar{X}_{\text{train}} \left(\bar{\beta}_{TT} - Q \Lambda Q^\top \bar{X}_{\text{context}}^\top \bar{X}_{\text{context}} \bar{\beta}_{TT}\right) \bar{\beta}_{TT}^\top \bar{X}_{\text{context}}^\top \bar{X}_{\text{context}} \\
&\approx 2\eta \bar{X}_{\text{train}}^\top \bar{X}_{\text{train}} \left(\bar{\beta}_{TT} - n Q \Lambda Q^\top \bar{\beta}_{TT} - \sqrt{n} Q \Lambda Q^\top g\right)\left(n \bar{\beta}_{TT}^\top + \sqrt{n} g^\top\right) \\
&= 2\eta \bar{X}_{\text{train}}^\top \bar{X}_{\text{train}} \left(n \bar{\beta}_{TT} \bar{\beta}_{TT}^\top + \sqrt{n} \bar{\beta}_{TT} g^\top - n^2 Q \Lambda Q^\top \bar{\beta}_{TT} \bar{\beta}_{TT}^\top - n\sqrt{n} Q \Lambda Q^\top \bar{\beta}_{TT} g^\top - n\sqrt{n} Q \Lambda Q^\top g \bar{\beta}_{TT}^\top - n Q \Lambda Q^\top g g^\top\right) \\
&= 2\eta \bar{X}_{\text{train}}^\top \bar{X}_{\text{train}} \left(n\left(I - n Q \Lambda Q^\top\right)\bar{\beta}_{TT} \bar{\beta}_{TT}^\top + \sqrt{n}\left(I - n Q \Lambda Q^\top\right)\bar{\beta}_{TT} g^\top - n\sqrt{n} Q \Lambda Q^\top g \bar{\beta}_{TT}^\top - n Q \Lambda Q^\top g g^\top\right) \\
\Delta W^\top &= 2\eta \left(n \bar{\beta}_{TT} \bar{\beta}_{TT}^\top \left(I - n Q \Lambda Q^\top\right) + \sqrt{n} g \bar{\beta}_{TT}^\top \left(I - n Q \Lambda Q^\top\right) - n\sqrt{n} \bar{\beta}_{TT} g^\top Q \Lambda Q^\top - n g g^\top Q \Lambda Q^\top\right) \bar{X}_{\text{train}}^\top \bar{X}_{\text{train}}
\end{aligned}
$$

By plugging the above definitions of $\bar{W}_{TT}, \bar{W}$ into Equation (27), we encounter the following difference term in two of the expressions:

$$
\begin{aligned}
\bar{W}^\top \bar{W} - \bar{W}_{TT}^\top \bar{W}_{TT} &= \bar{W}^\top \bar{W} - (\bar{W}^\top + \Delta W^\top)(\bar{W} + \Delta W) \\
&= -\bar{W}^\top \Delta W - \Delta W^\top \bar{W} - \Delta W^\top \Delta W.
\end{aligned}
\tag{28}
$$

Hence, we need to calculate both the first-order expectations $\mathbb{E}_{\bar{X}_{\text{train}},g}[\Delta W], \mathbb{E}_{\bar{X}_{\text{train}},g}[\Delta W^\top]$ and the second-order expectation $\mathbb{E}_{\bar{X}_{\text{train}},g}[\Delta W^\top \Delta W]$. Calculating the first-order expectations gives:

$$\mathbb{E}_{\bar{X}_{\text{train}},g}[\Delta W] = 2\eta kn\left(\left(I - nQ\Lambda Q^\top\right)\bar{\beta}_{\text{TT}}\bar{\beta}_{\text{TT}}^\top - \|\bar{\beta}_{\text{TT}}\|_2^2 \, Q\Lambda Q^\top\right), \tag{29}$$

$$\mathbb{E}_{\bar{X}_{\text{train}},g}\left[\Delta W^\top\right] = 2\eta kn\left(\bar{\beta}_{\text{TT}}\bar{\beta}_{\text{TT}}^\top\left(I - nQ\Lambda Q^\top\right) - \|\bar{\beta}_{\text{TT}}\|_2^2 \, Q\Lambda Q^\top\right). \tag{30}$$

Using (29) and (30), the first two terms of the loss difference are:

$$\mathbb{E}_{\bar{X}_{\text{train}},g}\left[\bar{\beta}_{\text{TT}}^\top n\left(\bar{W}_{\text{TT}} - \bar{W}\right)\bar{\beta}_{\text{TT}}\right] = 2\eta kn\left(n\|\bar{\beta}_{\text{TT}}\|_2^2 \bar{\beta}_{\text{TT}}^\top\left(I - nQ\Lambda Q^\top\right)\bar{\beta}_{\text{TT}} - n\|\bar{\beta}_{\text{TT}}\|_2^2 \bar{\beta}_{\text{TT}}^\top Q\Lambda Q^\top\bar{\beta}_{\text{TT}}\right), \tag{31}$$

$$\mathbb{E}_{\bar{X}_{\text{train}},g}\left[\bar{\beta}_{\text{TT}}^\top n\left(\bar{W}_{\text{TT}}^\top - \bar{W}^\top\right)\bar{\beta}_{\text{TT}}\right] = 2\eta kn\left(n\|\bar{\beta}_{\text{TT}}\|_2^2 \bar{\beta}_{\text{TT}}^\top\left(I - nQ\Lambda Q^\top\right)\bar{\beta}_{\text{TT}} - n\|\bar{\beta}_{\text{TT}}\|_2^2 \bar{\beta}_{\text{TT}}^\top Q\Lambda Q^\top\bar{\beta}_{\text{TT}}\right). \tag{32}$$

Similar to proofs of previous theorems, by plugging $M = I$ in Lemma B.3, we know that $\mathbb{E}_X[(X^\top X)(X^\top X)] = k(k+1)\Sigma^2 + k\text{tr}(\Sigma)\Sigma$ where $X \in \mathbb{R}^{k \times d}$ has its rows drawn i.i.d from $\mathcal{N}(0,\Sigma)$. Therefore,

$$\mathbb{E}_{\bar{X}_{\text{train}}}\left[\bar{X}_{\text{train}}^\top \bar{X}_{\text{train}}\bar{X}_{\text{train}}^\top \bar{X}_{\text{train}}\right] = k(k+d+1)I.$$

Utilizing the above fact, we compute:

$$\mathbb{E}_{\bar{X}_{\text{train}},g}\left[\Delta W^\top \Delta W\right]$$
$$= 4\eta^2 \mathbb{E}_g\Big[\big(n\bar{\beta}_{\text{TT}}\bar{\beta}_{\text{TT}}^\top\left(I - nQ\Lambda Q^\top\right) + \sqrt{n}g\bar{\beta}_{\text{TT}}^\top\left(I - nQ\Lambda Q^\top\right) - n\sqrt{n}\bar{\beta}_{\text{TT}}g^\top Q\Lambda Q^\top - ngg^\top Q\Lambda Q^\top\big)$$
$$\qquad k(k+d+1)I\big(n\left(I - nQ\Lambda Q^\top\right)\bar{\beta}_{\text{TT}}\bar{\beta}_{\text{TT}}^\top + \sqrt{n}\left(I - nQ\Lambda Q^\top\right)\bar{\beta}_{\text{TT}}g^\top - n\sqrt{n}Q\Lambda Q^\top g\bar{\beta}_{\text{TT}}^\top - nQ\Lambda Q^\top gg^\top\big)\Big]$$
$$= 4\eta^2 k(k+d+1)\mathbb{E}_g\Big[n^2\bar{\beta}_{\text{TT}}\bar{\beta}_{\text{TT}}^\top\left(I - nQ\Lambda Q^\top\right)\left(I - nQ\Lambda Q^\top\right)\bar{\beta}_{\text{TT}}\bar{\beta}_{\text{TT}}^\top - n^2\bar{\beta}_{\text{TT}}\bar{\beta}_{\text{TT}}^\top\left(I - nQ\Lambda Q^\top\right)Q\Lambda Q^\top gg^\top$$
$$\quad - n^2 gg^\top Q\Lambda Q^\top\left(I - nQ\Lambda Q^\top\right)\bar{\beta}_{\text{TT}}\bar{\beta}_{\text{TT}}^\top + n^2 gg^\top Q\Lambda Q^\top Q\Lambda Q^\top gg^\top + ng\bar{\beta}_{\text{TT}}^\top\left(I - nQ\Lambda Q^\top\right)\left(I - nQ\Lambda Q^\top\right)\bar{\beta}_{\text{TT}}g^\top$$
$$\quad + n^3\bar{\beta}_{\text{TT}}g^\top Q\Lambda Q^\top Q\Lambda Q^\top g\bar{\beta}_{\text{TT}}^\top - n^2 g\bar{\beta}_{\text{TT}}^\top\left(I - nQ\Lambda Q^\top\right)Q\Lambda Q^\top g\bar{\beta}_{\text{TT}}^\top - n^2\bar{\beta}_{\text{TT}}g^\top Q\Lambda Q^\top\left(I - nQ\Lambda Q^\top\right)\bar{\beta}_{\text{TT}}g^\top\Big].$$

Let's inspect each term inside the expectation in the above sum:

$$\mathbb{E}_g\left[n^3\bar{\beta}_{\text{TT}}g^\top Q\Lambda Q^\top Q\Lambda Q^\top g\bar{\beta}_{\text{TT}}^\top\right] = n^3\bar{\beta}_{\text{TT}}\mathbb{E}_g\left[g^\top Q\Lambda^2 Q^\top g\right]\bar{\beta}_{\text{TT}}^\top$$
$$= n^3\bar{\beta}_{\text{TT}}\mathbb{E}_z\left[z^\top \Lambda^2 z\right]\bar{\beta}_{\text{TT}}^\top \text{ where } z \sim \mathcal{N}\left(0, \|\bar{\beta}_{\text{TT}}\|_2^2 I\right)$$
$$= n^3\|\bar{\beta}_{\text{TT}}\|_2^2 \, \text{tr}(\Lambda^2)\bar{\beta}_{\text{TT}}\bar{\beta}_{\text{TT}}^\top$$

$$\mathbb{E}_g\left[n^2 g\bar{\beta}_{\text{TT}}^\top\left(I - nQ\Lambda Q^\top\right)Q\Lambda Q^\top g\bar{\beta}_{\text{TT}}^\top\right] = n^2\mathbb{E}_g\left[g\left(\bar{\beta}_{\text{TT}}^\top\left(I - nQ\Lambda Q^\top\right)Q\Lambda Q^\top g\right)\right]\bar{\beta}_{\text{TT}}^\top$$
$$= n^2\mathbb{E}_g\left[g\left(g^\top Q\Lambda Q^\top\left(I - nQ\Lambda Q^\top\right)\bar{\beta}_{\text{TT}}\right)\right]\bar{\beta}_{\text{TT}}^\top$$
$$= n^2\|\bar{\beta}_{\text{TT}}\|_2^2 Q\Lambda Q^\top\left(I - nQ\Lambda Q^\top\right)\bar{\beta}_{\text{TT}}\bar{\beta}_{\text{TT}}^\top$$

$$\mathbb{E}_g\left[n^2\bar{\beta}_{\text{TT}}g^\top Q\Lambda Q^\top\left(I - nQ\Lambda Q^\top\right)\bar{\beta}_{\text{TT}}g^\top\right] = n^2\bar{\beta}_{\text{TT}}\mathbb{E}_g\left[\left(g^\top Q\Lambda Q^\top\left(I - nQ\Lambda Q^\top\right)\bar{\beta}_{\text{TT}}\right)g^\top\right]$$
$$= n^2\bar{\beta}_{\text{TT}}\mathbb{E}_g\left[\left(\bar{\beta}_{\text{TT}}^\top\left(I - nQ\Lambda Q^\top\right)Q\Lambda Q^\top g\right)g^\top\right]$$
$$= n^2\|\bar{\beta}_{\text{TT}}\|_2^2 \bar{\beta}_{\text{TT}}\bar{\beta}_{\text{TT}}^\top\left(I - nQ\Lambda Q^\top\right)Q\Lambda Q^\top$$

$$\mathbb{E}_g\left[ng\bar{\beta}_{\text{TT}}^\top\left(I - nQ\Lambda Q^\top\right)\left(I - nQ\Lambda Q^\top\right)\bar{\beta}_{\text{TT}}g^\top\right] = n\mathbb{E}_g\left[g\left(\bar{\beta}_{\text{TT}}^\top\left(I - nQ\Lambda Q^\top\right)\left(I - nQ\Lambda Q^\top\right)\bar{\beta}_{\text{TT}}\right)g^\top\right]$$
$$= n\bar{\beta}_{\text{TT}}^\top\left(I - nQ\Lambda Q^\top\right)\left(I - nQ\Lambda Q^\top\right)\bar{\beta}_{\text{TT}}\mathbb{E}_g\left[gg^\top\right]$$
$$= n\|\bar{\beta}_{\text{TT}}\|_2^2\left(\bar{\beta}_{\text{TT}}^\top\left(I - nQ\Lambda Q^\top\right)\left(I - nQ\Lambda Q^\top\right)\bar{\beta}_{\text{TT}}\right)I$$

$$\mathbb{E}_g\left[n^2 gg^\top Q\Lambda Q^\top Q\Lambda Q^\top gg^\top\right] = n^2\mathbb{E}_g\left[gg^\top Q\Lambda^2 Q^\top gg^\top\right]$$
$$= n^2\|\bar{\beta}_{\text{TT}}\|_2^4\left(\text{tr}(\Lambda^2)I + 2Q\Lambda^2 Q^\top\right)$$

$$\mathbb{E}_g\left[n^2 gg^\top Q\Lambda Q^\top\left(I - nQ\Lambda Q^\top\right)\bar{\beta}_{\text{TT}}\bar{\beta}_{\text{TT}}^\top\right] = n^2\|\bar{\beta}_{\text{TT}}\|_2^2 Q\Lambda Q^\top\left(I - nQ\Lambda Q^\top\right)\bar{\beta}_{\text{TT}}\bar{\beta}_{\text{TT}}^\top$$

$$\mathbb{E}_g\left[n^2\bar{\beta}_{\text{TT}}\bar{\beta}_{\text{TT}}^\top\left(I - nQ\Lambda Q^\top\right)Q\Lambda Q^\top gg^\top\right] = n^2\|\bar{\beta}_{\text{TT}}\|_2^2 \bar{\beta}_{\text{TT}}\bar{\beta}_{\text{TT}}^\top\left(I - nQ\Lambda Q^\top\right)Q\Lambda Q^\top.$$

Hence, considering that the eigenvalues of $\mathbf{\Lambda}$ are smaller than $\frac{1}{n+1}$, we drop lower order terms after combining the results above:

$$
\begin{aligned}
\mathbb{E}_{\bar{X}_{\text{train}},g}\left[\Delta \boldsymbol{W}^\top \Delta \boldsymbol{W}\right] &= 4\eta^2 k(k+d+1)\Big[n^3\|\bar{\boldsymbol{\beta}}_{\text{TT}}\|_2^2\,\text{tr}(\boldsymbol{\Lambda}^2)\bar{\boldsymbol{\beta}}_{\text{TT}}\bar{\boldsymbol{\beta}}_{\text{TT}}^\top + n^2\bar{\boldsymbol{\beta}}_{\text{TT}}\bar{\boldsymbol{\beta}}_{\text{TT}}^\top \boldsymbol{Q}\,(\boldsymbol{I}-n\boldsymbol{\Lambda})^2\,\boldsymbol{Q}^\top\bar{\boldsymbol{\beta}}_{\text{TT}}\bar{\boldsymbol{\beta}}_{\text{TT}}^\top \\
&\quad - 2n^2\|\bar{\boldsymbol{\beta}}_{\text{TT}}\|_2^2\,\boldsymbol{Q}\left(\boldsymbol{\Lambda}-n\boldsymbol{\Lambda}^2\right)\boldsymbol{Q}^\top\bar{\boldsymbol{\beta}}_{\text{TT}}\bar{\boldsymbol{\beta}}_{\text{TT}}^\top - 2n^2\|\bar{\boldsymbol{\beta}}_{\text{TT}}\|_2^2\,\bar{\boldsymbol{\beta}}_{\text{TT}}\bar{\boldsymbol{\beta}}_{\text{TT}}^\top \boldsymbol{Q}\left(\boldsymbol{\Lambda}-n\boldsymbol{\Lambda}^2\right)\boldsymbol{Q}^\top \\
&\quad + n\,\|\bar{\boldsymbol{\beta}}_{\text{TT}}\|_2^2\left(\bar{\boldsymbol{\beta}}_{\text{TT}}^\top \boldsymbol{Q}\,(\boldsymbol{I}-n\boldsymbol{\Lambda})^2\,\boldsymbol{Q}^\top\bar{\boldsymbol{\beta}}_{\text{TT}}\right)\boldsymbol{I} + n^2\|\bar{\boldsymbol{\beta}}_{\text{TT}}\|_2^4\left(\text{tr}(\boldsymbol{\Lambda}^2)\boldsymbol{I}+2\boldsymbol{Q}\boldsymbol{\Lambda}^2\boldsymbol{Q}^\top\right)\Big] \\
&\approx 4\eta^2 k(k+d+1)\Big[n^3\|\bar{\boldsymbol{\beta}}_{\text{TT}}\|_2^2\,\text{tr}(\boldsymbol{\Lambda}^2)\bar{\boldsymbol{\beta}}_{\text{TT}}\bar{\boldsymbol{\beta}}_{\text{TT}}^\top + n^2\bar{\boldsymbol{\beta}}_{\text{TT}}\bar{\boldsymbol{\beta}}_{\text{TT}}^\top \boldsymbol{Q}\,(\boldsymbol{I}-n\boldsymbol{\Lambda})^2\,\boldsymbol{Q}^\top\bar{\boldsymbol{\beta}}_{\text{TT}}\bar{\boldsymbol{\beta}}_{\text{TT}}^\top \\
&\quad + n\,\|\bar{\boldsymbol{\beta}}_{\text{TT}}\|_2^2\left(\bar{\boldsymbol{\beta}}_{\text{TT}}^\top \boldsymbol{Q}\,(\boldsymbol{I}-n\boldsymbol{\Lambda})^2\,\boldsymbol{Q}^\top\bar{\boldsymbol{\beta}}_{\text{TT}}\right)\boldsymbol{I}\Big].
\end{aligned}
$$

Thus, we reach the following result:

$$
\mathbb{E}_{\bar{X}_{\text{train}},g}\left[\bar{\boldsymbol{\beta}}_{\text{TT}}^\top \Delta \boldsymbol{W}^\top \Delta \boldsymbol{W}\bar{\boldsymbol{\beta}}_{\text{TT}}\right] \approx 4\eta^2 k(k+d+1)\Big[n^3\|\bar{\boldsymbol{\beta}}_{\text{TT}}\|_2^6\,\text{tr}(\boldsymbol{\Lambda}^2) + (n^2+n)\|\bar{\boldsymbol{\beta}}_{\text{TT}}\|_2^4\,\bar{\boldsymbol{\beta}}_{\text{TT}}^\top \boldsymbol{Q}\,(\boldsymbol{I}-n\boldsymbol{\Lambda})^2\,\boldsymbol{Q}^\top\bar{\boldsymbol{\beta}}_{\text{TT}}\Big]. \tag{33}
$$

Now, recall from Equation (28) that the second-order difference is in the form:

$$
\bar{\boldsymbol{W}}^\top \bar{\boldsymbol{W}} \; - \; \bar{\boldsymbol{W}}_{\text{TT}}^\top \bar{\boldsymbol{W}}_{\text{TT}} = -\bar{\boldsymbol{W}}^\top \Delta \boldsymbol{W} - \Delta \boldsymbol{W}^\top \bar{\boldsymbol{W}} - \Delta \boldsymbol{W}^\top \Delta \boldsymbol{W}.
$$

Therefore, we also need to calculate the expectations of the terms $\bar{\boldsymbol{W}}^\top \Delta \boldsymbol{W}, \Delta \boldsymbol{W}^\top \bar{\boldsymbol{W}}$. Using previous results (29) and (30), we get:

$$
\begin{aligned}
\mathbb{E}_{\bar{X}_{\text{train}},g}\left[-\bar{\boldsymbol{W}}^\top \Delta \boldsymbol{W}\right] &= -2\eta k\,\boldsymbol{Q}\boldsymbol{\Lambda}\boldsymbol{Q}^\top\left(n\left(\boldsymbol{I}-n\boldsymbol{Q}\boldsymbol{\Lambda}\boldsymbol{Q}^\top\right)\bar{\boldsymbol{\beta}}_{\text{TT}}\bar{\boldsymbol{\beta}}_{\text{TT}}^\top - n\|\bar{\boldsymbol{\beta}}_{\text{TT}}\|_2^2\,\boldsymbol{Q}\boldsymbol{\Lambda}\boldsymbol{Q}^\top\right) \\
&= -2\eta kn\left(\boldsymbol{Q}(\boldsymbol{\Lambda}-n\boldsymbol{\Lambda}^2)\boldsymbol{Q}^\top\bar{\boldsymbol{\beta}}_{\text{TT}}\bar{\boldsymbol{\beta}}_{\text{TT}}^\top - \|\bar{\boldsymbol{\beta}}_{\text{TT}}\|_2^2\,\boldsymbol{Q}\boldsymbol{\Lambda}^2\boldsymbol{Q}^\top\right) \\
\mathbb{E}_{\bar{X}_{\text{train}},g}\left[-\Delta \boldsymbol{W}^\top \bar{\boldsymbol{W}}\right] &= -2\eta kn\left(\bar{\boldsymbol{\beta}}_{\text{TT}}\bar{\boldsymbol{\beta}}_{\text{TT}}^\top \boldsymbol{Q}(\boldsymbol{\Lambda}-n\boldsymbol{\Lambda}^2)\boldsymbol{Q}^\top - \|\bar{\boldsymbol{\beta}}_{\text{TT}}\|_2^2\,\boldsymbol{Q}\boldsymbol{\Lambda}^2\boldsymbol{Q}^\top\right).
\end{aligned}
$$

This gives us:

$$
\mathbb{E}_{\bar{X}_{\text{train}},g}\left[-\bar{\boldsymbol{\beta}}_{\text{TT}}^\top \bar{\boldsymbol{W}}^\top \Delta \boldsymbol{W}\bar{\boldsymbol{\beta}}_{\text{TT}}\right] = -2\eta kn\left(\|\bar{\boldsymbol{\beta}}_{\text{TT}}\|_2^2\,\bar{\boldsymbol{\beta}}_{\text{TT}}^\top \boldsymbol{Q}(\boldsymbol{\Lambda}-n\boldsymbol{\Lambda}^2)\boldsymbol{Q}^\top\bar{\boldsymbol{\beta}}_{\text{TT}} - \|\bar{\boldsymbol{\beta}}_{\text{TT}}\|_2^2\,\bar{\boldsymbol{\beta}}_{\text{TT}}^\top \boldsymbol{Q}\boldsymbol{\Lambda}^2\boldsymbol{Q}^\top\bar{\boldsymbol{\beta}}_{\text{TT}}\right) \tag{34}
$$

$$
\mathbb{E}_{\bar{X}_{\text{train}},g}\left[-\bar{\boldsymbol{\beta}}_{\text{TT}}^\top \Delta \boldsymbol{W}^\top \bar{\boldsymbol{W}}\bar{\boldsymbol{\beta}}_{\text{TT}}\right] = -2\eta kn\left(\|\bar{\boldsymbol{\beta}}_{\text{TT}}\|_2^2\,\bar{\boldsymbol{\beta}}_{\text{TT}}^\top \boldsymbol{Q}(\boldsymbol{\Lambda}-n\boldsymbol{\Lambda}^2)\boldsymbol{Q}^\top\bar{\boldsymbol{\beta}}_{\text{TT}} - \|\bar{\boldsymbol{\beta}}_{\text{TT}}\|_2^2\,\bar{\boldsymbol{\beta}}_{\text{TT}}^\top \boldsymbol{Q}\boldsymbol{\Lambda}^2\boldsymbol{Q}^\top\bar{\boldsymbol{\beta}}_{\text{TT}}\right). \tag{35}
$$

Exploiting (33), (34) and (35), we get

$$
\begin{aligned}
&\mathbb{E}_{\bar{X}_{\text{train}},g}\left[\bar{\boldsymbol{\beta}}_{\text{TT}}^\top\, n(n+1)\left(\bar{\boldsymbol{W}}^\top \bar{\boldsymbol{W}} \; - \; \bar{\boldsymbol{W}}_{\text{TT}}^\top \bar{\boldsymbol{W}}_{\text{TT}}\right)\bar{\boldsymbol{\beta}}_{\text{TT}}\right] \\
&\approx -4\eta^2 k(k+d+1)n(n+1)\Big[n^3\|\bar{\boldsymbol{\beta}}_{\text{TT}}\|_2^6\,\text{tr}(\boldsymbol{\Lambda}^2) + (n^2+n)\|\bar{\boldsymbol{\beta}}_{\text{TT}}\|_2^4\,\bar{\boldsymbol{\beta}}_{\text{TT}}^\top \boldsymbol{Q}\,(\boldsymbol{I}-n\boldsymbol{\Lambda})^2\,\boldsymbol{Q}^\top\bar{\boldsymbol{\beta}}_{\text{TT}}\Big] \\
&\quad - 4\eta kn^2(n+1)\left(\|\bar{\boldsymbol{\beta}}_{\text{TT}}\|_2^2\,\bar{\boldsymbol{\beta}}_{\text{TT}}^\top \boldsymbol{Q}(\boldsymbol{\Lambda}-n\boldsymbol{\Lambda}^2)\boldsymbol{Q}^\top\bar{\boldsymbol{\beta}}_{\text{TT}} - \|\bar{\boldsymbol{\beta}}_{\text{TT}}\|_2^2\,\bar{\boldsymbol{\beta}}_{\text{TT}}^\top \boldsymbol{Q}\boldsymbol{\Lambda}^2\boldsymbol{Q}^\top\bar{\boldsymbol{\beta}}_{\text{TT}}\right)
\end{aligned} \tag{36}
$$

$$
\begin{aligned}
&\mathbb{E}_{\bar{X}_{\text{train}},g}\left[\bar{\boldsymbol{\beta}}_{\text{TT}}^\top\, n\left(\text{tr}(\bar{\boldsymbol{W}}^\top \bar{\boldsymbol{W}}) - \text{tr}(\bar{\boldsymbol{W}}_{\text{TT}}^\top \bar{\boldsymbol{W}}_{\text{TT}})\right)\bar{\boldsymbol{\beta}}_{\text{TT}}\right] \\
&\approx -4\eta^2 k(k+d+1)\Big[n^4\|\bar{\boldsymbol{\beta}}_{\text{TT}}\|_2^6\,\text{tr}(\boldsymbol{\Lambda}^2) + n^2(n+d)\|\bar{\boldsymbol{\beta}}_{\text{TT}}\|_2^4\,\bar{\boldsymbol{\beta}}_{\text{TT}}^\top \boldsymbol{Q}\,(\boldsymbol{I}-n\boldsymbol{\Lambda})^2\,\boldsymbol{Q}^\top\bar{\boldsymbol{\beta}}_{\text{TT}}\Big] \\
&\quad - 4\eta kn^2\left[\|\bar{\boldsymbol{\beta}}_{\text{TT}}\|_2^2\,\bar{\boldsymbol{\beta}}_{\text{TT}}^\top \boldsymbol{Q}(\boldsymbol{\Lambda}-n\boldsymbol{\Lambda}^2)\boldsymbol{Q}^\top\bar{\boldsymbol{\beta}}_{\text{TT}} - \|\bar{\boldsymbol{\beta}}_{\text{TT}}\|_2^4\,\text{tr}(\boldsymbol{\Lambda}^2)\right].
\end{aligned} \tag{37}
$$

Combining (31), (32), (36) and (37), and plugging into the Equation (27) gives the expected loss improvement:

$$
\begin{aligned}
&\mathcal{L}(\boldsymbol{W}) - \mathcal{L}_{TT}(\boldsymbol{W}_{\text{TT}}) \\
&\approx 4\eta kn^2\left(\|\bar{\boldsymbol{\beta}}_{\text{TT}}\|_2^2\,\bar{\boldsymbol{\beta}}_{\text{TT}}^\top \boldsymbol{Q}\,(\boldsymbol{I}-n\boldsymbol{\Lambda})\,\boldsymbol{Q}^\top\,\bar{\boldsymbol{\beta}}_{\text{TT}} - \|\bar{\boldsymbol{\beta}}_{\text{TT}}\|_2^2\,\bar{\boldsymbol{\beta}}_{\text{TT}}^\top \boldsymbol{Q}\boldsymbol{\Lambda}\boldsymbol{Q}^\top\,\bar{\boldsymbol{\beta}}_{\text{TT}}\right) \\
&\quad - 4\eta kn^2(n+1)\left(\|\bar{\boldsymbol{\beta}}_{\text{TT}}\|_2^2\,\bar{\boldsymbol{\beta}}_{\text{TT}}^\top \boldsymbol{Q}(\boldsymbol{\Lambda}-n\boldsymbol{\Lambda}^2)\boldsymbol{Q}^\top\bar{\boldsymbol{\beta}}_{\text{TT}} - \|\bar{\boldsymbol{\beta}}_{\text{TT}}\|_2^2\,\bar{\boldsymbol{\beta}}_{\text{TT}}^\top \boldsymbol{Q}\boldsymbol{\Lambda}^2\boldsymbol{Q}^\top\,\bar{\boldsymbol{\beta}}_{\text{TT}}\right) \\
&\quad - 4\eta kn^2\left[\|\bar{\boldsymbol{\beta}}_{\text{TT}}\|_2^2\,\bar{\boldsymbol{\beta}}_{\text{TT}}^\top \boldsymbol{Q}(\boldsymbol{\Lambda}-n\boldsymbol{\Lambda}^2)\boldsymbol{Q}^\top\bar{\boldsymbol{\beta}}_{\text{TT}} - \|\bar{\boldsymbol{\beta}}_{\text{TT}}\|_2^4\,\text{tr}(\boldsymbol{\Lambda}^2)\right] \\
&\quad - 4\eta^2 k(k+d+1)n\left[n^3(n+2)\|\bar{\boldsymbol{\beta}}_{\text{TT}}\|_2^6\,\text{tr}(\boldsymbol{\Lambda}^2) + n\left((n+1)^2+n+d\right)\|\bar{\boldsymbol{\beta}}_{\text{TT}}\|_2^4\,\bar{\boldsymbol{\beta}}_{\text{TT}}^\top \boldsymbol{Q}\,(\boldsymbol{I}-n\boldsymbol{\Lambda})^2\,\boldsymbol{Q}^\top\bar{\boldsymbol{\beta}}_{\text{TT}}\right] \\
&\approx 4\eta kn^2\|\bar{\boldsymbol{\beta}}_{\text{TT}}\|_2^2\,\bar{\boldsymbol{\beta}}_{\text{TT}}^\top \boldsymbol{Q}\,(\boldsymbol{I}-n\boldsymbol{\Lambda})^2\,\boldsymbol{Q}^\top\bar{\boldsymbol{\beta}}_{\text{TT}} - 4\eta^2 k(k+d+1)n^2\Big[n^3\,\|\bar{\boldsymbol{\beta}}_{\text{TT}}\|_2^6\,\text{tr}(\boldsymbol{\Lambda}^2) + (n^2+d)\|\bar{\boldsymbol{\beta}}_{\text{TT}}\|_2^4\,\bar{\boldsymbol{\beta}}_{\text{TT}}^\top \boldsymbol{Q}\,(\boldsymbol{I}-n\boldsymbol{\Lambda})^2\,\boldsymbol{Q}^\top\bar{\boldsymbol{\beta}}_{\text{TT}}\Big].
\end{aligned} \tag{38, 39}
$$

This is a quadratic expression in $\eta$, and we can solve for the optimal $\eta^*$ by setting the derivative to 0. This way, the optimal step size is approximately:

$$\eta^* \approx \frac{\bar{\beta}_{\mathrm{TT}}^\top \boldsymbol{Q} \left(\boldsymbol{I} - n\boldsymbol{\Lambda}\right)^2 \boldsymbol{Q}^\top \bar{\beta}_{\mathrm{TT}}}{2(k+d+1) \|\bar{\beta}_{\mathrm{TT}}\|_2^2 \left[n^3 \|\bar{\beta}_{\mathrm{TT}}\|_2^2 \mathrm{tr}(\boldsymbol{\Lambda}^2) + (n^2+d)\bar{\beta}_{\mathrm{TT}}^\top \boldsymbol{Q} \left(\boldsymbol{I} - n\boldsymbol{\Lambda}\right)^2 \boldsymbol{Q}^\top \bar{\beta}_{\mathrm{TT}}\right]}.$$

Plugging this optimal $\eta^*$ value into Equation (39), the population loss gain becomes:

$$\frac{k}{k+d+1} \frac{\left(\bar{\beta}_{\mathrm{TT}}^\top \boldsymbol{Q} \left(\boldsymbol{I} - n\boldsymbol{\Lambda}\right)^2 \boldsymbol{Q}^\top \bar{\beta}_{\mathrm{TT}}\right)^2}{n \|\bar{\beta}_{\mathrm{TT}}\|_2^2 \mathrm{tr}(\boldsymbol{\Lambda}^2) + \left(1 + \frac{d}{n^2}\right)\bar{\beta}_{\mathrm{TT}}^\top \boldsymbol{Q} \left(\boldsymbol{I} - n\boldsymbol{\Lambda}\right)^2 \boldsymbol{Q}^\top \bar{\beta}_{\mathrm{TT}}}.$$

Thus, considering the definitions $\tilde{\beta}_{\mathrm{TT}} = \boldsymbol{Q}^\top \bar{\beta}_{\mathrm{TT}}$ and $A = \tilde{\beta}_{\mathrm{TT}}^\top (\boldsymbol{I} - n\boldsymbol{\Lambda})^2 \tilde{\beta}_{\mathrm{TT}}$, $B = n\mathrm{tr}(\boldsymbol{\Lambda}^2) \|\bar{\beta}_{\mathrm{TT}}\|_2^2$ (notice that $\ell_2$ norm is unitarily-invariant, so that $\|\tilde{\beta}_{\mathrm{TT}}\| = \|\bar{\beta}_{\mathrm{TT}}\|$ ) and recalling $n/d = \Theta(1)$ recovers the desired final expression. Also, by viewing the initial task as $\tilde{\beta}_{\mathrm{TT}}$, we match the diagonal covariance form described in Section 5. Therefore, we conclude our argument

$$\mathcal{L}(\boldsymbol{W}) - \mathcal{L}_{TT}(\boldsymbol{W}_{\mathrm{TT}}) \approx \frac{k}{k+d} \frac{A^2}{A+B}.$$

For the second part of the proposition, let's write the initial loss by applying the covariance shift from Lemma C.3 so that the transformed features $\bar{x} = \boldsymbol{\Sigma}_x^{-1/2} x$ have identity covariance and $\bar{\boldsymbol{W}} = \boldsymbol{\Sigma}_x^{1/2} \boldsymbol{W}^* \boldsymbol{\Sigma}_x^{1/2}$ is the corresponding minimizer of the transformed system. Again, we set $\tilde{\beta}_{\mathrm{TT}} = \boldsymbol{Q}^\top \bar{\beta}_{\mathrm{TT}}$ in order to have diagonal covariances. Assuming $\sigma = 0$ and plugging these in the population loss formula from Lemma B.2 with respect to the task $\bar{\beta}_{\mathrm{TT}}$ gives:

$$
\begin{aligned}
\mathcal{L}(\boldsymbol{W}) &= \bar{\beta}_{\mathrm{TT}}^\top \Big[\boldsymbol{I} - n\boldsymbol{I}\bar{\boldsymbol{W}}\boldsymbol{I} - n\boldsymbol{I}\bar{\boldsymbol{W}}^\top \boldsymbol{I} + n(n+1)\boldsymbol{I}\bar{\boldsymbol{W}}^\top \boldsymbol{I}\bar{\boldsymbol{W}}\boldsymbol{I} + n\mathrm{tr}(\bar{\boldsymbol{W}}^\top \boldsymbol{I}\bar{\boldsymbol{W}}\boldsymbol{I})\boldsymbol{I}\Big]\bar{\beta}_{\mathrm{TT}} + \sigma^2 n\mathrm{tr}(\bar{\boldsymbol{W}}^\top \boldsymbol{I}\bar{\boldsymbol{W}}\boldsymbol{I}) + \sigma^2 \\
&= \|\bar{\beta}_{\mathrm{TT}}\|_2^2 - 2n\bar{\beta}_{\mathrm{TT}}^\top \boldsymbol{Q}\boldsymbol{\Lambda}\boldsymbol{Q}^\top \bar{\beta}_{\mathrm{TT}} + n(n+1)\bar{\beta}_{\mathrm{TT}}^\top \boldsymbol{Q}\boldsymbol{\Lambda}^2 \boldsymbol{Q}^\top \bar{\beta}_{\mathrm{TT}} + n\mathrm{tr}(\boldsymbol{\Lambda}^2)\|\bar{\beta}_{\mathrm{TT}}\|_2^2 \\
&\approx \|\bar{\beta}_{\mathrm{TT}}\|_2^2 - 2n\bar{\beta}_{\mathrm{TT}}^\top \boldsymbol{Q}\boldsymbol{\Lambda}\boldsymbol{Q}^\top \bar{\beta}_{\mathrm{TT}} + n^2 \bar{\beta}_{\mathrm{TT}}^\top \boldsymbol{Q}\boldsymbol{\Lambda}^2 \boldsymbol{Q}^\top \bar{\beta}_{\mathrm{TT}} + n\mathrm{tr}(\boldsymbol{\Lambda}^2)\|\bar{\beta}_{\mathrm{TT}}\|_2^2 \\
&= \bar{\beta}_{\mathrm{TT}}^\top \boldsymbol{Q}(\boldsymbol{I} - n\boldsymbol{\Lambda})^2 \boldsymbol{Q}^\top \bar{\beta}_{\mathrm{TT}} + n\|\bar{\beta}_{\mathrm{TT}}\|_2^2 \mathrm{tr}(\boldsymbol{\Lambda}^2) \\
&= \tilde{\beta}_{\mathrm{TT}}^\top (\boldsymbol{I} - n\boldsymbol{\Lambda})^2 \tilde{\beta}_{\mathrm{TT}} + n\|\tilde{\beta}_{\mathrm{TT}}\|_2^2 \mathrm{tr}(\boldsymbol{\Lambda}^2) \\
&= A + B.
\end{aligned}
$$

Hence, $\mathcal{L}(\boldsymbol{W}) \approx A + B$. The proof is complete. $\square$

**Proposition 5.2.** *Let* $\boldsymbol{\Sigma}_x = \boldsymbol{I}, \sigma^2 = 0$ *and suppose* $\boldsymbol{\Sigma}_\beta$ *is an arbitrary diagonal covariance matrix with the first $r$ diagonal entries being non-zero. Then, the minimizer* $\boldsymbol{W}^*$ *of the pre-training loss* (1) *that has minimal Frobenius norm and its population loss are given by*

$$\boldsymbol{W}^* = \left((n+1)\boldsymbol{I}' + \mathrm{tr}(\boldsymbol{\Sigma}_\beta)\boldsymbol{\Sigma}_\beta^\dagger\right)^\dagger, \quad \mathcal{L}(\boldsymbol{W}^*) \approx A + B,$$

*where* $\boldsymbol{I}' = diag(\mathbf{1}_r, \mathbf{0}_{d-r})$.

*Proof.* While Li et al. (2024) derives the solution form in the full-rank case, the same closed-form expression can be adapted when $\boldsymbol{\Sigma}_\beta$ is low-rank. Indeed, if $\boldsymbol{\Sigma}_\beta$ has zeros in its last $d - r$ diagonal entries, those coordinates do not appear in the pre-training loss. Consequently, there is a set of minimizers $\boldsymbol{W}$ which differ only in those degenerate directions. Among these, the minimal-Frobenius-norm criterion forces all degenerate coordinates to be zero as any nonzero component in that subspace would increase $\|\boldsymbol{W}\|_F$. Consequently,

$$\boldsymbol{W}^* = \left((n+1)\boldsymbol{I}' + \mathrm{tr}(\boldsymbol{\Sigma}_\beta)\boldsymbol{\Sigma}_\beta^\dagger\right)^{-1}$$

zeroes out precisely the degenerate directions and is the unique Frobenius-minimal solution.

For the second part of the proposition, the argument directly follows from the proof of Theorem 5.3. $\square$

**Corollary 5.4.** *Consider the setting in Theorem 5.3. Recall the definitions of $\alpha = n/d$ and $\gamma = k/d$ and define $c_1 = \dfrac{A}{\|\beta_{TT}\|_2^2}$, $c_2 = \dfrac{B}{\|\beta_{TT}\|_2^2}$. Then, there exists a phase transition point $\gamma^\star \approx \dfrac{(c_1 + c_2) - (c_1 + c_2)^2}{c_2 (2c_1 + c_2)}$ such that $\gamma < \gamma^\star$ if and only if it is better to utilize the pre-trained $\boldsymbol{W}^*$ over the null initialization $\mathbf{0}_{d \times d}$.*

*Proof.* Recall the population loss for $W$ given by Lemma B.2:

$$\mathcal{L}(W) = \boldsymbol{\beta}_{\mathrm{TT}}^\top \left[ \Sigma_x - n\Sigma_x W \Sigma_x - n\Sigma_x W^\top \Sigma_x + n(n+1)\Sigma_x W^\top \Sigma_x W \Sigma_x + n\mathrm{tr}(W^\top \Sigma_x W \Sigma_x)\Sigma_x \right] \boldsymbol{\beta}_{\mathrm{TT}} + \sigma^2 n\mathrm{tr}(W^\top \Sigma_x W \Sigma_x) + \sigma^2.$$

First, plugging $W = \mathbf{0}_{d\times d}$ yields the initial loss of $\|\tilde{\boldsymbol{\beta}}_{\mathrm{TT}}\|_2^2$. Also, setting $\bar{W}^* = \mathbf{0}_{d\times d}$ in Theorem 5.3, we know that the improvement by test-time-training is approximately:

$$\frac{k}{k+d} \|\tilde{\boldsymbol{\beta}}_{\mathrm{TT}}\|_2^2.$$

At the same time, still by Theorem 5.3, we know that if our initial weight matrix is the pre-trained $W^*$ the improvement is approximately:

$$\frac{k}{k+d}\frac{A^2}{A+B}.$$

Whereas, the corresponding initial loss is approximately $\mathcal{L}(W^*) \approx A + B$. Therefore, we check when it's better to use pre-trained matrix $W^*$ over $\mathbf{0}_{d\times d}$ (null) matrix with the below inequality, which compares the losses after the test-time-training update:

$$A + B - \frac{k}{k+d}\frac{A^2}{A+B} < \|\tilde{\boldsymbol{\beta}}_{\mathrm{TT}}\|_2^2 - \frac{k}{k+d}\|\tilde{\boldsymbol{\beta}}_{\mathrm{TT}}\|_2^2.$$

In lines with the proof of Corollary 4.6, denote $\beta := \frac{k}{k+d}$. Then, a series of algebraic manipulations give:

$$
\begin{aligned}
A + B - \beta\frac{A^2}{A+B} < \|\tilde{\boldsymbol{\beta}}_{\mathrm{TT}}\|_2^2 - \beta\|\tilde{\boldsymbol{\beta}}_{\mathrm{TT}}\|_2^2 &\iff c_1 + c_2 - \beta\frac{c_1^2}{c_1+c_2} < 1 - \beta \\
&\iff \beta\left(1 - \frac{c_1^2}{c_1+c_2}\right) < 1 - (c_1+c_2) \\
&\iff \frac{1}{\beta} > \frac{c_1 + c_2 - c_1^2}{(c_1+c_2)(1-(c_1+c_2))} \\
&\iff 1 + \frac{d}{k} > 1 + \frac{c_2(2c_1+c_2)}{c_1+c_2-(c_1+c_2)^2} \\
&\iff \frac{k}{d} = \gamma < \frac{c_1+c_2-(c_1+c_2)^2}{c_2(2c_1+c_2)}.
\end{aligned}
$$

This completes our argument. □

## D. Further Experimental Results for Section 6

To illustrate the improvement more clearly, we also provide a version of Figure 3a with the x-axis on a log scale, shown in Figure 4.

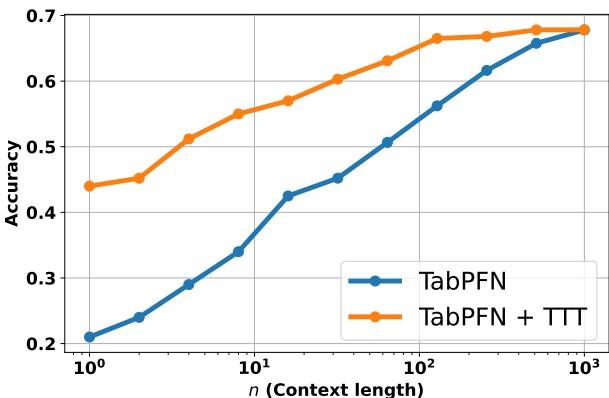

*Figure 4.* Accuracy of TabPFN model with and without test-time-training as a function of number of in-context samples *n* with the x-axis in log scale.

