# OpenReview forum: "Test-Time Training Provably Improves Transformers as In-context Learners"
_ICML.cc/2025/Conference — ICML 2025 poster_

### Official Review · Reviewer_XqSw · 2025-03-09

**Overall Recommendation:** 4

**Summary:**

The paper investigates the theoretical and empirical advantages of Test-Time Training (TTT) for improving transformers as in-context learners. Authors develop a theoretical framework to characterize how a single-step gradient update at test time enhances in-context learning. The authors provide a rigorous theoretical proof. The analysis covers the role of alignment between pretraining distribution and target tasks, the ability of TTT to mitigate distribution shift, and its sample complexity benefits. Empirical evaluations support the theoretical findings and demonstrate that TTT significantly reduces required sample sizes while maintaining performance.

**Claims And Evidence:**

The paper provides sound theoretical backing and empirical results to support the claims.
However, the reliance on linear transformers may limit generalizability to more complex architectures.

**Essential References Not Discussed:**

No

**Experimental Designs Or Analyses:**

Yes, the experimental design is valid and directly supports the theoretical claims, including the experiments on TabPFN (demonstrating efficiency improvements and sample size reductions) and the simulations with GPT-2 (showing TTT for handling distribution shift).

Potential limitations are the narrow focus on TabPFN and the need for additional benchmarks to validate broader applicability.

**Methods And Evaluation Criteria:**

Yes.

**Other Comments Or Suggestions:**

No.

**Other Strengths And Weaknesses:**

Strength:
Very novel and natural theoretical explanation for TTT and ICL.
Weaknesses:
Limited to simple or toy linear transformer architectures and basic TabPFN tasks, restricting broader applicability.

**Questions For Authors:**

1. Would it be necessary to analyze the effects of different optimizers separately?
2. Do you know about work that explains in-context learning using gradient updates? Since TTT also relies on gradient updates, what are the deeper differences and connections between the two when viewed from the perspective of gradient updates?

**Relation To Broader Scientific Literature:**

Yes, I think this paper can help the broad community to understand TTT better and provide new insights for utilizing TTT. Also, the experiments on TabPFN shows new opportunities to reduce test time scaling cost (when using ICL).

**Theoretical Claims:**

Yes, the paper presents sound and rigorous theoretical derivations.
I've read and checked the Analysis for Isotropic Covariances and  Analysis for General Covariance parts for the theoretical claims and there's no obvious issues.

---

> ### Author Rebuttal · Authors · 2025-04-01
>
> We thank the reviewer for their favorable evaluation of our work and for the helpful feedback. We now reply to each of the points raised below.
>
> > **Experimental Designs Or Analyses.** Yes, the experimental design is valid ... Potential limitations are the narrow focus on TabPFN and the need for additional benchmarks to validate broader applicability.
>
> **Response:** Thank you for your feedback. We chose to conduct our experiments with the TabPFN model because it's well-aligned with our theoretical setting with similar token encodings but different prior distributions. This favorable agreement made TabPFN a particularly good setting to conduct large-scale experiments on how TTT reduces the context length needed. For the benchmarks used while evaluating TabPFN, we realized that we did not provide enough contextual information in the paper for our choice of the T4 dataset (Gardner et al., 2024), which is an extensive collection of public benchmarks containing diverse real tabular tasks. In the revision, we will include how we filtered the datasets in T4 according to the requirements of TabPFN and our problem setup, as well as how we constructed the experimental pipeline. Still, we agree that additional domains beyond tabular data would help demonstrate the broader applicability of our theoretical claims.
>
>
>
> > **Weakness.** Limited to simple or toy linear transformer architectures and basic TabPFN tasks, restricting broader applicability.
>
> We thank the reviewer for the feedback. The TabPFN v2 model itself has been pre-trained on a wide range of tabular distributions, and we evaluate it on the T4 dataset (Gardner et al., 2024) through many diverse real-life tabular tasks. So, we actually demonstrate how TTT reduces the number of in-context examples across a broad range of realistic classification tasks by plotting the average performance across all tasks in T4. This extensive coverage enables us to. For the discussion on model architecture, we refer the reviewer to our related response for the weakness mentioned by Reviewer StUH.
>
> > **Q1.** Would it be necessary to analyze the effects of different optimizers separately?
>
> **Response:**
>
> We appreciate this question. In our TabPFN experiments, we initially used AdamW as the optimizer, as it's known to work well with transformer architectures. Based on your question, we tried the same experiment using the standard SGD with momentum and observed minimal differences in final performance. The results are available [here](https://anonymous.4open.science/r/TestTimeTrainingProvablyImprovesTransformersAsInContextLearners-F845/TabPFN_SGD.pdf).
>
>
>
> > **Q2.** Do you know about work that explains in-context learning using gradient updates? Since TTT also relies on gradient updates, what are the deeper differences and connections between the two when viewed from the perspective of gradient updates?
>
> **Response:**
>
> Thank you for highlighting this connection. The existing works (e.g., Ahn *et al.*, 2023; Mahankali *et al.*, 2023; Zhang *et al.*, 2023; Li *et al.*, 2024) show that in-context learning can be viewed as implicit gradient descent within a transformer's forward pass which effectively simulates a preconditioned GD governed by the task prior. In contrast, our TTT approach explicitly updates the model parameters at inference and provides task-specific adaptation rather than one governed only by the prior. In ICL, any gradient descent occurs internally without updating the weights; as a result, the model can only implement the preconditioned GD it meta‐learned from pretraining. Instead, TTT uses a real gradient step on test data and handles the distribution shift with a low computational cost. Finally, while most ICL analyses operate at the population level (i.e., with infinite samples), our setting provides finite‐sample guarantees on performance improvements on how a single‐step update can reduce the needed context length.
>
>
> Ahn, K., Cheng, X., Daneshmand, H., and Sra, S. Transformers learn to implement preconditioned gradient descent for in‐context learning, 2023. URL https://arxiv.org/abs/2306.00297
>
> Zhang, R., Frei, S., and Bartlett, P. L. Trained transformers learn linear models in-context, 2023. URL https://arxiv.org/abs/2306.09927.
>
> Li, Y., Rawat, and Oymak, S. Fine-grained analysis of in-context linear estimation: Data, architecture, and beyond, 2024. URL https://arxiv.org/abs/2407.10005.
>
> Mahankali, A. V., Hashimoto, T., and Ma, T. One step of gradient descent is provably the optimal in-context learner with one layer of linear self-attention, 2024. URL https://arxiv.org/abs/2307.03576

---

> > ### Comment · Reviewer_XqSw · 2025-04-04
> >
> > Thanks for answering my questions and addressing my concern, I'll keep my score.

---

### Official Review · Reviewer_StUH · 2025-03-13

**Overall Recommendation:** 3

**Summary:**

This paper examines the impact of test-time training (TTT) on the performance of a single-layer linear attention model (without an MLP) after a single gradient step during test-time fine-tuning. The authors focus on the problem of in-context linear regression and characterize the performance improvement achieved through one step of gradient descent for a linear attention model in terms of the context length (\(n\)) and the ambient dimension (\(d\)).

**Claims And Evidence:**

Yes, the claim well supported with empirical experiment on synthetic data.

**Essential References Not Discussed:**

NA

**Experimental Designs Or Analyses:**

The experiments are reasonable; however, to support the claim that this theory provides insight into broader in-context learning (ICL) problems, it would be beneficial to include additional ICL tasks, such as a two-layer ReLU neural network, or expand the evaluation with more datasets for the TabPFN experiments(currently it is only tested on 1 data set).

**Methods And Evaluation Criteria:**

Yes, they all make sense.

**Other Comments Or Suggestions:**

trying on more data set for TabPFN would be an interesting addition helps with completeness.

**Other Strengths And Weaknesses:**

I found the paper interesting, and the theoretical insights were valuable. The experiment with real TabPFN was also compelling. The main limitation of the paper is its focus on a single-step gradient descent and a simplified model. However, despite this constraint, the paper still provides meaningful insights.

**Questions For Authors:**

1. How would performing additional gradient descent steps at test time affect the results? Specifically, how would it impact the loss improvement described in the paper's theorem?

2. Can this approach be extended to a one-layer transformer with softmax attention and an MLP? How challenging would this extension be, and what factors contribute to the difficulty?

**Relation To Broader Scientific Literature:**

NA

**Theoretical Claims:**

The theoretical framework is sound; however, it is highly limited to a single attention layer (without an MLP) and applies only to the linear regression problem.

---

> ### Author Rebuttal · Authors · 2025-04-01
>
> We thank the reviewer for the detailed feedback and positive evaluation of our work. We address the weaknesses and questions brought up below.
>
> > **Experimental Designs Or Analyses.** The experiments are reasonable; however, to support the claim that this theory provides insight into broader in-context learning (ICL) problems, it would be beneficial to include additional ICL tasks, such as a two-layer ReLU neural network, or expand the evaluation with more datasets for the TabPFN experiments (currently it is only tested on 1 data set).
>
> **Response:** Thank you for highlighting this important detail. In lines 379–381, we mistakenly referred to a “single dataset,” but we intended to say “each dataset” in the The Tremendous TabLib Trawl (T4) benchmark (Gardner et al., 2024). T4 is a large-scale, high-quality collection of public tabular benchmarks containing approximately 4 million tables. Our choice to use T4 was indeed to make evaluations more comprehensive by covering a broad range of data. As stated in lines 412-413, we report the average performance across all datasets in T4 with sufficient samples in Figure 3-a.
>
> > **Weakness.** The main limitation of the paper is its focus on a single-step gradient descent and a simplified model.
>
> **Response:** Thank you for your feedback. We want to highlight that we focus on single‐step gradient descent because it is both practically motivated due to its efficiency and theoretically interesting. The TTT update can be implemented in a single forward-backward pass with appropriate masking, as described by the Remark in Section 3, which makes it a meaningful choice with minimal test-time compute overhead. Moreover, our theory and experiments demonstrate that single-step gradient descent can be sufficient, which aligns with the existing practical works (Akyürek et al., 2024) that just a few gradient steps offer significant test-time improvements. Also, even with the linear attention model, the analysis poses significant technical challenges: we must calculate the 8th moments of matrices, and since this is practically intractable, we rely on Gaussian approximation techniques presented in Lemma B.1 in Appendix B.
>
> > **Q1.** How would performing additional gradient descent steps at test time affect the results? Specifically, how would it impact the loss improvement described in the paper's theorem?
>
> **Response:**  Thank you for your thoughtful question. Theoretically, there are technical difficulties in analyzing multiple steps of gradient descent as each subsequent update introduces higher-order dependencies and makes it hard to compute the expectation $\mathbb{E}[\mathbf{W}^\top \mathbf{W}]$. However, as we discussed in our multi-gradient step experiment in Figure 2c, one would expect the improvement obtained from the next steps to diminish. This can also be intuitively seen in Section 5, as the improvement is given by $A^2/(A+B)$ where $A$ is the misalignment and $B$ is the magnitude term. Each gradient step reduces $A$ as we decrease the misalignment between $\mathbf{W}$ and $\boldsymbol{\beta}_{TT}$, and that $B$ remains relatively stable with the updates. Then, it can be argued that the loss improvement decreases in the next gradient steps.
>
> > **Q2.** Can this approach be extended to a one-layer transformer with softmax attention and an MLP? How challenging would this extension be, and what factors contribute to the difficulty?
>
> **Response:** We thank the reviewer for raising this point. For a single-layer transformer with softmax attention and possibly an MLP layer at the end, the gradient involves calculating terms like $\nabla_{\mathbf{W} } \mathrm{softmax}(\mathbf{X} \mathbf{W} \mathbf{X}^\top)$ where $\mathbf{W}_Q \mathbf{W}_K^\top = \mathbf{W}$. For example, considering one row of this resulting matrix $\mathbf{s} = \begin{bmatrix} s_1 ~ \cdots ~ s_n \end{bmatrix}^\top = \mathrm{softmax}(\mathbf{X}\mathbf{W}\mathbf{z}) \in \mathbb{R}^{n}$ and taking its derivative w.r.t $\mathbf{W}$ gives the following form:
>
> $$
> \qquad \frac{\partial s_i}{\partial \mathbf{W}} = s_i  (\mathbf{x_i} - \sum_{j} s_j \mathbf{x_j}) \mathbf{z}^{\top}.
> $$
>
> As we analyze the performance of the model after the gradient descent update, we encounter expressions involving high-order moments of data matrices while calculating $\mathbb{E}[\mathbf{W}^\top \mathbf{W}]$. With this much complicated gradient form, analyzing the update and especially calculating high-order moments of the data matrix $\mathbf{X}$ becomes far more challenging and intractable. Also, adding an MLP layer might further complicate the backward pass by adding another layer of non-linear transformation.

---

### Official Review · Reviewer_KMYg · 2025-03-13

**Overall Recommendation:** 3

**Summary:**

This paper investigates how Test-Time Training (TTT) affects transformer models' in-context learning capabilities. The authors formulate this as a two-stage problem: a model is first trained on a pre-training dataset, then further trained on test data using single-step gradient descent. They provide theoretical proofs for performance gains through TTT, analyze the relationship between context length and test data size, and compare TTT with direct training on test data from scratch. Experimental results validate their theoretical findings.

**Claims And Evidence:**

Yes. The claims are supported by theoretical proofs and experimental validation.

**Essential References Not Discussed:**

NA

**Experimental Designs Or Analyses:**

While the theoretical validation experiments are sound, the real-world experiments in Section 6.2 have methodological issues. The comparison between TabPFN and TabPFN+TTT should account for the additional 1000 samples used in TTT as part of the context length. The comparison in Figure 3a is therefore problematic. Additionally, for the T4 benchmark experiments, more detailed information about test dataset size and sampling methodology would be beneficial.

**Methods And Evaluation Criteria:**

The paper presents a simplified version of TTT compared to practical implementations. Their approach fixes the context and uses k different mini-batch data with MSE loss to train the model, testing it using the same context as training. This simplification differs significantly from TTT used in practice such as [1],[2], potentially limiting the practical insights this work can provide.

[1] Akyürek, E., et al. "The surprising effectiveness of test-time training for abstract reasoning."
 [2] Sun, Y., et al. "Learning to (learn at test time): RNNs with expressive hidden states."

**Other Comments Or Suggestions:**

If there are any misunderstandings on my part, please point them out, and I will reconsider my evaluation of this work.

**Other Strengths And Weaknesses:**

**Strengths**:
1. Comprehensive comparison of TTT performance with different initialization strategies (pre-trained vs. zero initialization)
2. Thorough analysis of performance gains relative to context and test example numbers, with experimental results supporting theoretical findings

**Weaknesses**:
1. The paper's oversimplified TTT procedure. The main comparison between randomly initialized models and pretrained models lacks practical insights due to this oversimplified framework.
2. The experiments in Section 6.2 need more detailed analysis and better clarification of the methodological choices.
3. The expressions and claims in propositions and theorems are unclear and could be better formulated.

**Questions For Authors:**

please refer to the weakness part in Other Strengths And Weaknesses, I don’t have more question, my main concern is the oversimplified problem setting. This simplification makes the approach significantly diverge from practical TTT applications.

**Relation To Broader Scientific Literature:**

The paper's main contribution lies in proving that fine-tuning on labeled test data can improve performance. However, this empirical result is somewhat expected, limiting the paper's potential impact.

**Theoretical Claims:**

The mathematical proofs contain imprecise statements, particularly in the use of approximation symbols ($\approx$) without specifying error bounds. For a paper focused on rigorous theoretical analysis, such imprecision is problematic, as seen in Theorem 4.2 and Corollary 4.5.

---

> ### Author Rebuttal · Authors · 2025-04-01
>
> We thank the reviewer for their detailed evaluation. We reply to each of the points below.
>
> > **Theoretical claims**. The mathematical proofs contain imprecise statements. ...
>
> **Response:** Thank you for pointing this out. The approximations in Theorems 4.2, 5.3, and Corollary 4.5 stem from ignoring lower order terms and using Gaussian Approximation in Lemma B.1, which rigorously bounds the error term. In the proportional regime ($n/d = \Theta(1)$), all approximations yield an error on the order of $n^{-1}$, while the main loss terms are in the constant order. Consequently, any approximation error is small, which is evident in the close match between theoretical and empirical curves in Figures 1-2. We also wanted to kindly note that **after Theorems 4.2 and 5.3 in lines 190-192 and 290-291, we refer readers to Remark B.5 in Appendix B for discussion on the validity of the approximations.** We appreciate your feedback, and to clarify these points, we plan to move Remark B.5 into the main body.
>
> > **Methods And Evaluation Criteria**. Their approach fixes the context and uses k different mini-batch data ..., testing it using the same context as training.
>
> > **Weakness 1.** The paper's oversimplified TTT procedure. ... comparison between randomly initialized models and pretrained models lacks practical insights due to this oversimplified framework.
>
> **Response:** Thank you for your question. While our single-step TTT setup may not capture every real-world scenario, it's both theoretically interesting and practically informative. The main reason behind using the same context is to make gradient update tractable. For testing, however, we do not reuse the same context as in training but instead evaluate the model over *any* new context–query pairs by taking expectations (lines 140–142). Even under the current setup, there are major technical challenges, such as computing $\mathbb{E}[\mathbf{W}^\top \mathbf{W}]$, which requires the Gaussian Approximation technique and a rigorous error analysis. Without fixing the context in the training, it's practically infeasible to obtain a closed-form update (Proposition 3.1) and compute this expectation. Still, we believe that our framework presents two clear, theoretically verified practical insights:
>
> 1) We characterize how alignment between the pre-trained model and target task affects TTT improvements and phase-transition points, which is illustrated in Figures 2a-2b and practically verified by GPT-2 experiments of Figure 2c.
>
> 2) Both our theory and experiments indicate that one-step TTT can be sufficient, which complements the existing practical findings (Akyürek et al., 2024) that a few gradient steps yield significant test-time improvements.
>
> Overall, we believe our framework offers useful practical insights and can serve as a baseline theoretical reference for future research on TTT-ICL interactions.
>
> > **Experimental Designs Or Analyses.** .... the real-world experiments in Section 6.2 have methodological issues. ...
> >
> > **Weakness 2.** The experiments in Section 6.2 need more detailed analysis and better clarification of the methodological choices.
>
> **Response:** Thank you for your feedback. While TabPFN+TTT indeed uses an extra $1000$ samples, our main goal is to demonstrate how TTT reduces the needed sample complexity for each query from the new task, not just one. The TTT update is performed only once per task to adapt the model, which helps improve the inference time efficiency as each subsequent evaluation needs much fewer in-context examples. We want to highlight that this is more evident near the zero-shot regime as TTT helps the model "memorize" new task dynamics and improves accuracy from 0.2 to 0.45. Akyürek et al. (2024) also argue that during TTT, for each task, the model is evaluated on a batch of test points after a small number of gradient steps on a test-time training set, specifically in Section 5 for BBH tasks.
>
> Regarding the dataset, we use the T4 benchmark from Gardner et al. (2024), which is a large set of real-world tabular tasks and consists of four million tables. Following the official TabPFN v2 implementation and our own setup, we select the datasets with at least 1,250 samples (1,000 for training, using an 80–20 split), at most 10 classes (selecting the 10 most frequent), and a 100-feature limit (chosen randomly). We also convert regression tasks into 10-class classifications based on quartiles to maintain consistency in training and evaluation. We appreciate your feedback and plan to detail our dataset choice and experimental methodology both in the main text and appendix.
>
> > **Weakness 3.** ... propositions and theorems are unclear and could be better formulated
>
> **Response:** We are planning to do a pass over the statements and relocate Remark B.5. into the main text for clarity and completeness of the approximations in the theorem statements. We welcome any additional suggestions to clarify our statements.

---

### Official Review · Reviewer_yFX9 · 2025-03-17

**Overall Recommendation:** 3

**Summary:**

In-context learning and Test-time Training (TTT) are two ways of enhancing the predictive power of pretrained models on new tasks at test time. In-context learning involves incorporating demonstrations from the task into the prompt context. TTT involves light finetuning of the model on data related to the test task. Recent work has shown that TTT can naturally integrate with in-context learning: one can lightly finetune the model on provided demonstrations and subsequently use them as an in-context prompt with the finetuned model.

This paper develops a theoretical understanding of TTT to enhance the in-context learning ability of a model for a given task. Specifically, given a model trained for in-context learning of linear functions, the model is test-time trained on in-context prompts from a target linear function. The analysis focuses on single-layer linear attention models where a single gradient step is used for TTT. The paper characterizes the performance of the test-time trained model as a function of context length, the number of TTT samples, and the alignment between the pretrained model and the target task. High-level takeaways from the analysis include:

- With TTT, one can reduce the number of in-context learning examples needed to perform well on a given task. Of course, this comes at the cost of additional training samples and extra time during the TTT phase.

- Comparison of TTT from scratch vs. TTT on a model pretrained for in-context learning: (i) The relative advantage of TTT on top of a pretrained in-context learning model depends on how "aligned" the target task is with the tasks the model was trained on. (ii) As the number of TTT samples increases, the relative advantage of TTT on a pretrained model diminishes.

Simulations demonstrate the theory, and some findings are also empirically validated on multi-layer non-linear Transformers.

Finally, the paper takes a model pretrained for in-context learning of tabular data and shows that with TTT, one can substantially reduce the number of in-context examples needed to perform well on given tasks. This leads to efficient inference, especially given the quadratically growing cost of softmax attention.

### update after rebuttal

I thank the authors for their detailed response and answering my clarifying questions. My general view of strengths and weaknesses of the paper remains the same. Hence, I would stick to my original rating.

**Claims And Evidence:**

Yes, the claims made in the submission are supported by clear evidence.

**Essential References Not Discussed:**

Prior work has been adequately discussed.

**Experimental Designs Or Analyses:**

The experimental design and analysis seems sound.

**Methods And Evaluation Criteria:**

Yes, the proposed methods and evaluation criteria make sense for the problem.

**Other Comments Or Suggestions:**

Minor typo: The gradient seems to be missing from the gradient descent update equation in line 134.

**Other Strengths And Weaknesses:**

Strengths:

The intersection of TTT and in-context learning provides a valuable framework for understanding the interplay between in-context learning and in-weights learning. This paper advances this understanding by formally exploring the role of factors such as alignment between target and pretraining tasks, the number of TTT samples, and context length.

I also liked the experiment with tabular data. Recently, Transformer models pretrained for in-context learning of tabular data have proven highly effective. However, inference with these models can be expensive for large tabular datasets due to the cost increasing qudratically with the sequence length. This work demonstrates that such costs can be significantly reduced using TTT, though at the expense of additional training time.

Overall, I enjoyed reading the paper. It is clearly written, with simulations accompanying most theoretical results, making it easy to absorb the main claims.

Weaknesses:

While the theory presented concretely highlights the effect of various factors on TTT, most results do not seem particularly surprising. A possible criticism is that the theory does not provide new insights that significantly change existing perspectives on TTT or in-context learning.

**Questions For Authors:**

Q1. What is the intuition behind non-monotonic trends in Figure 1a? This was probably the most non-intuitive result for me. It would be great if the authors could discuss the intuion and practical relevance of this result.

Q2. Are there any insights from your theoretical analysis that you think would hold generally and add to existing perspectives on TTT or in-context learning? I know this question is a bit subjective but I would love to hear authors' thoughts on this.

Q3. The experiment with GPT-2 is not on top of a pretrained GPT-2 model but using a randomly initialized GPT-2 style architecture? It would be good to clarify this to avoid confusion.

**Relation To Broader Scientific Literature:**

Discussed in summary and strengths and weaknesses sections.

**Theoretical Claims:**

I did not verify the proofs but the claims made intuitive sense and were also corroborated with simulations.

---

> ### Author Rebuttal · Authors · 2025-04-01
>
> We appreciate the reviewer’s positive assessment of our work and their detailed feedback. We now address each point below.
>
> > **Weakness.** While the theory presented concretely highlights the effect of various factors on TTT, most results do not seem particularly surprising. ... theory does not provide new insights that significantly change existing perspectives on TTT or in-context learning.
>
> **Response**: The main goal of our paper is to provide a rigorous theoretical framework that precisely quantifies the improvement by single-step gradient descent with TTT for in-context learning tasks. While it may seem intuitive that TTT would help or that transition points might emerge, to our knowledge, our work is the first formal analysis revealing why and how these phenomena occur. Achieving this required overcoming significant technical challenges, including computing high-order matrix moments using Gaussian approximations with a rigorous error analysis.
>
> Particularly, we characterize the improvement by TTT in terms of context length $(n)$, embedding dimension $(d)$, test-time training set size $(k)$, and the alignment between the pre-trained model and target task. This characterization allows us to show how the transition points depend on different regimes of test-time training set size $(k)$ and alignment between the model and target task, and we reveal the resulting non-monotonicities after TTT. Moreover, our theory and experiments show that one step of TTT is sufficient, which complements the existing practical research (Akyürek et al., 2024) stating that a few gradient steps offer significant test-time improvements. We hope that our results will serve as a foundation for future research on the interaction of TTT and ICL methods. We've provided a further discussion of insights in Q2.
>
> > Minor typo: The gradient seems to be missing from the gradient descent update equation in line 134.
>
> **Response:** Thanks; we will edit the revision accordingly.
>
> > **Q1.** What is the intuition behind non-monotonic trends in Figure 1a?
>
> **Response**: The non-monotonic behavior is observed as the result of two opposing effects, which are the initial loss and the improvement by TTT.
>
> * As $n$ grows against $d$ (i.e. $\alpha$ increases), the pre-trained model does better initially and already has a lower loss before TTT, which makes it harder to be further reduced by TTT as the rank-1 update is unable to correct all directions.
> * On the other hand, when $d$ grows against $n$  ($\alpha$ decreases), the initial loss is high, and there's more room (error) to be corrected by rank-1 update, and thus, the improvement by TTT is larger. This intuitively aligns with Theorem 4.2, which states that the TTT improvement scales as $(\frac{d}{n+d})^3$.
>
> Together, these two trends cause the non-monotonic behavior. Practically, this tells us that TTT through single-step gradient descent is more likely to be useful when $d = \Omega (n)$.
>
> > **Q2.** Are there any insights from your theoretical analysis that you think would hold generally and add to existing perspectives on TTT or in-context learning?
>
> **Response:** We thank the reviewer for the valuable question. A key insight from our paper is that test-time training can serve as a lightweight, supervised complement to ICL. When TTT is combined with ICL, one might benefit from the examples at test time in a supervised fashion and do a quick parameter update by treating a subset of them as training data instead of simply using them all as context. Accordingly, we analyze the single-step gradient descent as it has minimal test-time compute overhead (which makes it close to realistic settings). This way, the TTT update can be done only in a single forward-backward pass with appropriate masking, as discussed in the Remark in Section 3. Through our experiments, we also observe that a single gradient step of TTT captures a large part of the benefit compared to multiple steps, and might be an appealing method for practical scenarios. This observation also holds in practice as Akyürek et al. (2024) find that only a small number of gradient steps on test-time training data is enough to achieve strong performance. We believe that our work might inspire future integrations of TTT with ICL.
>
> > **Q3.** The experiment with GPT-2 is not on top of a pretrained GPT-2 model but using a randomly initialized GPT-2 style architecture?
>
> **Response:** We thank the reviewer for bringing this detail. Between lines 428-431, we discuss how we obtain the pre-trained and scratch models. The scratch model is just initialized randomly with the GPT-2 architecture before the TTT process. For the pre-trained model, we obtain it by training it until convergence with the tasks sampled from the task covariance $\boldsymbol{\Sigma}_{\boldsymbol{\beta}}$, and before the training, it's also initialized *randomly* using the GPT-2 style architecture. We will add this missing detail and clarify the related paragraph further.

---

### Decision · Program_Chairs · 2025-05-01

**Decision:**

Accept (poster)

**Comment:**

This paper investigates the impact of test-time training on in-context learning for linear models. Within a rigorous theoretical framework, the authors quantify its benefits, such as reducing the number of in-context examples required for downstream tasks. A nice balance of theoretical analysis with experimental observation distinguishes this work from other studies on in-context learning which often focus on either theoretical analyses or experimental observations.